# An updated end-to-end ecosystem model of the Northern California Current reflecting ecosystem changes due to recent marine heatwaves

Dylan G. E. Gomes[1,2¤]*, James J. Ruzicka[3], Lisa G. Crozier[4], David D. Huff[5], Elizabeth M. Phillips[6], Pierre-Yves Hernvann[7,8], Cheryl A. Morgan[2], Richard D. Brodeur[5], Jen E. Zamon[9], Elizabeth A. Daly[2], Joseph J. Bizzarro[10,11], Jennifer L. Fisher[5], Toby D. Auth[12]

1 National Academy of Sciences NRC Research Associateship Program, Northwest Fisheries Science Center, National Marine Fisheries Service, National Oceanic and Atmospheric Administration, Seattle, WA, United States of America, 2 Cooperative Institute for Marine Ecosystem and Resources Studies, Hatfield Marine Science Center, Oregon State University, Newport, OR, United States of America, 3 Ecosystem Sciences Division, Pacific Islands Fisheries Science Center, National Marine Fisheries Service, National Oceanic and Atmospheric Administration, Honolulu, HI, United States of America, 4 Fish Ecology Division, Northwest Fisheries Science Center, National Marine Fisheries Service, National Oceanic and Atmospheric Administration, Seattle, WA, United States of America, 5 Fish Ecology Division, Northwest Fisheries Science Center, National Marine Fisheries Service, National Oceanic and Atmospheric Administration, Newport, OR, United States of America, 6 Fishery Resource Analysis and Monitoring Division, Northwest Fisheries Science Center, National Marine Fisheries Service, National Oceanic and Atmospheric Administration, Seattle, WA, United States of America, 7 Conservation Biology Division, Northwest Fisheries Science Center, National Marine Fisheries Service, National Oceanic and Atmospheric Administration, Newport, OR, United States of America, 8 Institute of Marine Sciences, University of California, Santa Cruz, Santa Cruz, CA, United States of America, 9 Fish Ecology Division, Point Adams Research Station, Northwest Fisheries Science Center, National Marine Fisheries Service, National Oceanic and Atmospheric Administration, Hammond, OR, United States of America, 10 Fisheries Ecology Division, Southwest Fisheries Science Center, National Marine Fisheries Service, National Oceanic and Atmospheric Administration, Santa Cruz, CA, United States of America, 11 Fisheries Collaborative Program, University of Santa Cruz, Santa Cruz, CA, United States of America, 12 Pacific States Marine Fisheries Commission, Newport, OR, United States of America

¤ Current address: Forest and Rangeland Ecosystem Science Center, United States Geological Survey, Seattle, WA, United States of America

* dylan.ge.gomes@gmail.com

**Data Availability Statement:** Ecosystem model files and scripts, including the balanced and unbalanced diet matrices, biomass estimates,

## Abstract

The Northern California Current is a highly productive marine upwelling ecosystem that is economically and ecologically important. It is home to both commercially harvested species and those that are federally listed under the U.S. Endangered Species Act. Recently, there has been a global shift from single-species fisheries management to ecosystem-based fisheries management, which acknowledges that more complex dynamics can reverberate through a food web. Here, we have integrated new research into an end-to-end ecosystem model (i.e., physics to fisheries) using data from long-term ocean surveys, phytoplankton satellite imagery paired with a vertically generalized production model, a recently assembled diet database, fishery catch information, species distribution models, and existing literature. This spatially-explicit model includes 90 living and detrital functional groups ranging from phytoplankton, krill, and forage fish to salmon, seabirds, and marine mammals, and nine fisheries that occur off the coast of Washington, Oregon, and Northern California. This

various cleaning scripts, readme files, and all files mentioned in the text can be found at https://doi.org/10.5281/zenodo.7079777.

**Funding:** This research was performed while DGEG held a National Academy of Science National Research Council (NRC) Research Associateship award at the National Oceanic and Atmospheric Administration's National Marine Fisheries Service (NOAA Fisheries; NWFSC). JJB was supported by the Cooperative Institute for Marine, Earth, and Atmospheric Systems. The funders had no role in study design, data collection and analysis, decision to publish, or preparation of the manuscript.

**Competing interests:** The authors have declared that no competing interests exist.

model was updated from previous regional models to account for more recent changes in the Northern California Current (e.g., increases in market squid and some gelatinous zooplankton such as pyrosomes and salps), to expand the previous domain to increase the spatial resolution, to include data from previously unincorporated surveys, and to add improved characterization of endangered species, such as Chinook salmon (*Oncorhynchus tshawytscha*) and southern resident killer whales (*Orcinus orca*). Our model is mass-balanced, ecologically plausible, without extinctions, and stable over 150-year simulations. Ammonium and nitrate availability, total primary production rates, and model-derived phytoplankton time series are within realistic ranges. As we move towards holistic ecosystem-based fisheries management, we must continue to openly and collaboratively integrate our disparate datasets and collective knowledge to solve the intricate problems we face. As a tool for future research, we provide the data and code to use our ecosystem model.

## Introduction

The Northern California Current (NCC) marine ecosystem extends from Vancouver Island, British Columbia to Cape Mendocino, California and is a highly productive upwelling ecosystem that is economically and ecologically important [1, 2]. This ecosystem has recently experienced multiple biophysical stressors: an increase in water temperatures [3], seasonally low oxygen [4, 5], decreased pH and calcium carbonate saturation state (i.e., ocean acidification) [6, 7], increased magnitude and frequency of marine heatwaves (MHWs) (i.e., the 2014–2016 and 2019–2020 MHWs) [8–10], and coastwide harmful algal blooms [11, 12] that cumulatively have resulted in dramatic changes to the ecosystem [13–16]. Climate change is expected to continue to exacerbate these issues within the NCC in complex ways [17, 18], which is a cause for concern as the NCC is home to many commercially and recreationally important species as well as taxa listed under the Endangered Species Act (ESA) [19–21].

Over the last 30 years, there has been a growing recognition of the importance of holistic, ecosystem-based management [22–24]. Additionally, improved availability of long-term datasets, increased computing power, and advances in quantitative tools have allowed a heightened focus on multi-species and ecosystem-based management approaches that consider complex trophic interactions, incorporate physical oceanographic processes, and integrate multiple disparate data sources and their uncertainties [23, 25–30]. Ecosystem models have been used to address fisheries management questions because they track energy flow through modeled ecosystems, thereby evaluating sensitivities to perturbations of predator-prey interactions and furthering our understanding of poorly studied species, and allowing for the evaluation of long-term management scenarios [31]. In end-to-end ecosystem models, full ecosystems are parameterized from the physical oceanographic drivers to the trophic interactions within food webs (including fisheries). Advances in end-to-end ecosystem modeling efforts have leveraged multiple ongoing data collection efforts in the NCC to understand how complicated interactions shape ecosystem response to environmental perturbations [7, 32–36].

Ecosystems and predator-prey interactions are ever-changing in time and space in concert with fluctuations in climate [37, 38]. Thus, for a successful understanding of changing ecosystem states, we must continually update, adapt, or further develop ecosystem models with the most recently available data. Here we update and expand upon previous ecosystem models of the NCC described by Ruzicka et al. [32, 33] and present an end-to-end ecosystem model using the

EcoTran platform [27], which builds upon the widely-used Ecopath framework [25]. We used long-term NCC ship-based surveys and marine mammal stock assessments to derive biomass inputs for 77 of 90 functional groups (from phytoplankton, zooplankton, and micronekton to coastal pelagic fish species and salmon to seabirds, pinnipeds, and killer whales). We focused on data collected primarily during and after recent MHWs (2014–2021) to more accurately reflect the current conditions within the NCC, as there is evidence that the ecosystem has entered a novel state [15, 16, 39]. We updated the representation of food web interactions for 26 functional groups, including diet data from the MHW period for 19 groups, with published and unpublished datasets and reports to reflect potential reshuffling of some trophic links. We incorporated updated landings data from nine fisheries. We provide detailed descriptions, tables, model code, and visualizations of the ecosystem model. We also demonstrate our ability to represent ecosystem states using model validation metrics and visualizations [30, 40]. This new model has potential to become a tool for scientists and managers interested in the California Current marine ecosystem and will continue to evolve as data sources and information are gathered.

## Methods

### Model background

For the purposes of this model, we define the Northern California Current (NCC) domain (Fig 1) to extend latitudinally from Cape Flattery, Washington (48.34N) to just north of Cape Mendocino, in Eureka, California (40.80N) and longitudinally across the shelf from the 1 m to 1280 m isobaths. We define 15 subregions of the model both latitudinally and bathymetrically. That is, the model domain is divided into three bathymetric bins: inner shelf (1–100 m depths), middle shelf (101–200 m), and outer shelf (201–1280 m), and five latitudinal bins: northern California (40.8–42˚N), southern Oregon (42–44.4˚N), northern Oregon (44.4–46˚N), the Columbia River region (46–46.7˚N), and the Washington coast (46.7–48.34˚N). The ecosystem model is parameterized with inputs from satellite data, pelagic survey data, fishery data, local diet studies, marine mammal stock assessments, acoustic-trawl and bottom-trawl survey reports from the National Oceanic and Atmospheric Administration's (NOAA) National Marine Fisheries Service (NMFS)–from both the Northwest Fisheries Science Center (NWFSC) and the Southwest Fisheries Science Center (SWFSC), Oregon State University, and various other sources (see below and S1 Table). Where possible, we preferentially selected recent years (2014 –present) to parameterize initial conditions. In some cases that was not possible as new data is not readily available, so we relied on a previous version of the NCC food web (see below and S1 Table).

We chose to model the functioning of the NCC ecosystem using the EcoTran framework (from Ecopath-Transpose; [27, 32, 41]. EcoTran is a mass-balanced food web model that represents the energy flow across ecosystem components and expands upon the well-known Ecopath model [25]. Ecopath solves for the rates of live-weight biomass transfer along each trophic linkage by calculating the consumption demand of consumer groups (grazer or predator) upon each prey group given group biomasses, weight-specific consumption and production rates, and diet compositions. The Ecopath solution for the food web is a matrix describing each consumer group's consumption rate of each producer group. The Ecopath consumption matrix was re-expressed as a trophic network, mapping the fate of all consumed biomass by each functional group among all living consumers, detritus pools, and nitrogenous waste pools via EcoTran techniques (A.K.A. Ecopath-Transpose) [27, 32, 41]:

$$A_{ji} = \frac{D_{ij}c_j}{\sum_j (D_{ij}c_j)} \qquad \text{Eq1}$$

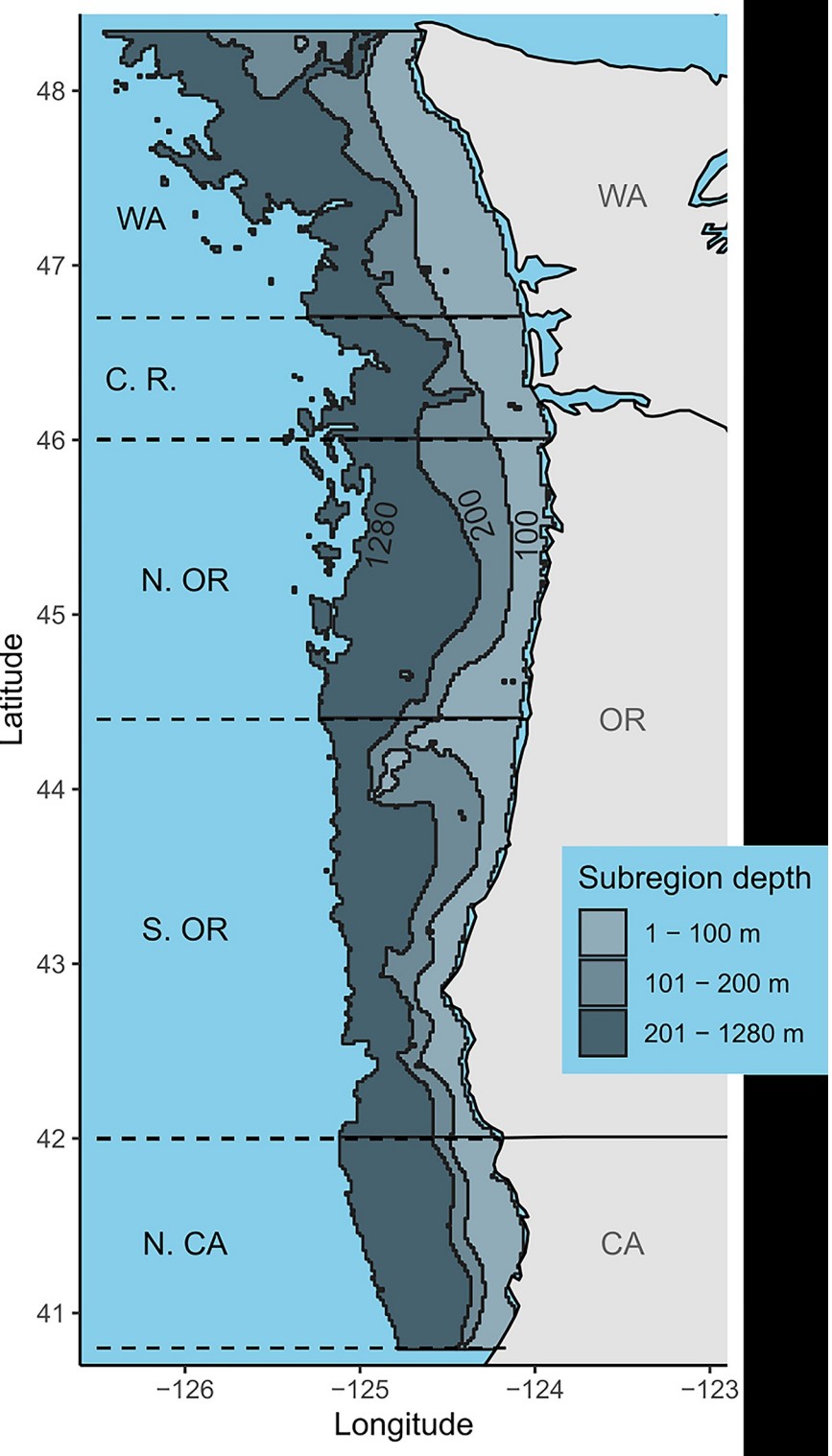

**Fig 1. Map of Northern California Current.** Extent of end-to-end ecosystem model in the Northern California Current marine ecosystem. Shaded gray bins indicate the 15 ecosystem model subregions. The model domain is broken down into three bathymetric bins (inner shelf: 1–100 m; mid shelf: 101–200 m; and outer shelf: 201–1280 m) and five latitudinal bins (northern California: 40.8–42˚N; southern Oregon: 42–44.4˚N; northern Oregon: 44.4–46˚N; Columbia River zone: 46–46.7˚N; and Washington coast: 46.7–48.34˚N). State outline data comes from US Department of Commerce, Census Bureau, Cartographic Boundary Files.

where $A_{ji}$ = the trophic network matrix (the fraction of total production of each producer $i$ consumed by each consumer $j$), $D_{ij}$ = the diet matrix (the fraction of each producer $i$ in the diet of each consumer $j$), and $c_j$ = consumption rate of consumer $j$. The trophic network matrix $A_{ji}$ was expanded to include detritus and nitrogenous waste (nitrate and ammonium) pools as distinct functional groups.

## Physical model structure and model drivers

EcoTran is a spatially-explicit end-to-end model and allows direct linking of physical oceanographic forcings to the food web model, which drives primary production and the transport of plankton, detritus, and nutrients across model domain boundaries. We use the 2-dimensional model structure of Ruzicka et al. [33, 41]. The cross-shelf physical model domain is divided into five sub-regions (Fig 2): box I inner shelf zone of coastal upwelling, boxes II and III middle shelf zone, boxes IV and V outer shelf zone. The middle and outer shelf zones are divided into surface (boxes II and IV) and subsurface layers (boxes III and V) defined by an annual mean mixed layer depth of 15 m [33, 41]. Boxes I, II, and IV each contain individual, vertically-integrated food webs representing the complete set of trophic interactions among pelagic and benthic functional groups. Sub-surface boxes III and V are used to account for the physical transport of nutrients, plankton, and detritus particles across the shelf at depth and the loss of sinking detritus from surface boxes (S2 and S3 Tables). Trophic interactions in boxes III and V are limited to the transfer of phytoplankton to detritus via senescence, the metabolism of detritus by bacteria, and the nitrification of ammonium. See Ruzicka et al. [33, 41] for details.

The currency of a time-dynamic EcoTran model is nitrogen input to the system as nitrate and ammonium at the base of the food web via upwelling and detritus remineralization by bacteria [33]. Nutrients are input to the system via advection flux across the ocean-shelf model domain boundary (boxes IV and V, Fig 2) that are defined by the daily coastal upwelling transport index (CUTI) [42], and by the monthly climatological nitrate and ammonium concentrations observed by the Newport Hydrological Line survey across the central Oregon shelf over years 1998–2008 (44.64˚ N) [33, 43]. Nutrient input drives primary production which, in turn,

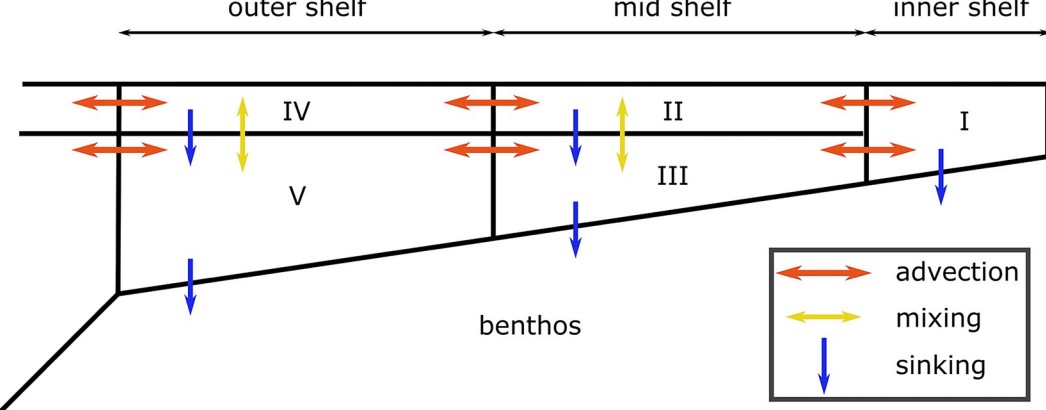

**Fig 2. Cross-shelf physical model.** The EcoTran ecosystem model allows direct linking of physical oceanographic forcings to the food web model, which drives primary production and the transport of plankton, detritus, and nutrients across model domain boundaries. The currency of a time-dynamic EcoTran model (see Figs 9 and 10 for examples) is nitrogen input to the system as nitrate and ammonium at the base of the food web via upwelling and detritus remineralization by bacteria. The ecosystem model is driven by nutrient flux that is important for bottom-up food web processes using the coastal upwelling transport index (CUTI) [42]. The CUTI time series (in daily time steps) drives advection (red arrows) of nutrients across the shelf. Primary production is supported by nutrient uptake and, in turn, supports grazing and predation by higher trophic level groups and catch by fishing fleets. Adapted from Ruzicka et al., 2016 [33].

supports grazing and predation by higher trophic level groups and catch by fishing fleets. Boundary conditions for non-nutrient functional groups are assumed to be identical on both sides of the surface and deep oceanic boundaries (reflective boundary conditions). There is no net input of non-nutrient biomass into the model domain nor dilution via physical flux of oceanic waters, though non-nutrient biomass can be exported from the model domain to the ocean.

## Mass-balancing

After parameterizing functional group biomasses (see below), we compared the model-wide (aggregated across all subregions, hereafter 'aggregated') average biomass values with those from Ruzicka et al. [32, 33]. When values differed by an order of magnitude, we revisited those datasets and data providers in search of errors in the data cleaning process or in the metadata (e.g., units of measurement). Once obvious errors were fixed or removed, then we attempted to mass-balance the aggregated model. During mass-balancing, imbalance may occur and highlight an insufficient biomass or excess production of a given functional group to sustain predation and/or fishing mortality, resulting in a loss or gain of energy in the system. Such imbalance could result from incorrect assumptions, or inaccurate values of the main trophic flow parameters or biomass inputs. The relative reliability of multiple datasets may be assessed by examining whether or not estimates from each source are unbalanced (i.e., a prey group's energy or biomass store is consumed more than the group produces itself).

The physiological rate parameters used by Ecopath to mass-balance the aggregated model are P/B = Biomass-specific Production rate, C/B = Biomass-specific Consumption rate, P/Q = Production efficiency, which can be defined as (P/B) / (C/B), AE = Assimilation Efficiency, and EE = Ecotrophic Efficiency (defined or estimated; Eq 2; Table 1). Most physiological rate parameters (P/B, C/B, P/Q, and AE) were taken from a previous NCC ecosystem model [32] or borrowed from other trophic models of the northeast-Pacific [31, 44, 45].

During mass-balancing, the ecotrophic efficiency (EE) is estimated by the Ecopath master equation (Eq 2 below) for most groups [excepting groups where biomass was estimated and EE was fixed (see 'Biomass density' section below) based on common Ecopath assumptions [25, 31]] as,

$$B_i \left(\frac{P}{B}\right)_i \times EE_i + I_i + BA_i = \Sigma \left[B_j \times \left(\frac{C}{B}\right)_j \times D_{ij}\right] + E_i + F_i \qquad \text{Eq2}$$

where $B_i$ is the biomass density of prey group i, $(P/B)_i$ is the biomass-specific production rate, $EE_i$ is the Ecotrophic Efficiency for group i, $I_i$ is the immigration rate into the model domain, $BA_i$ is a user-defined biomass accumulation term if biomass is known to be increasing or decreasing over time (for simplicity, we account for no known accumulation over time). Everything on the left side of the equation (i.e., the total production of group i in the system) must be balanced by the loss terms on the right, where $B_j$ is the biomass density of predator group j, $(C/B)_j$ is the biomass-specific consumption rate of predator group j, $D_{ij}$ represents the fraction of prey i consumed by predator j as defined by the diet matrix, $E_i$ is the emigration rate for prey group i, and $F_i$ is the fisheries catch of group i. $\Sigma$ indicates that everything in the brackets is summed across each predator j for the consumed group i. All of these terms are input parameters of the model, except either $B_i$ or $EE_i$ for each group. That is, only one of these terms is provided and the other is estimated during the mass-balancing process [25, 31].

The Ecopath EE is the fraction of a given functional group's production that is used in the system (i.e., not transferred to the detritus), hence between 0 and 1; that is, production is (i) consumed by another functional group or fishing, (ii) moved across model domain

**Table 1. NCC EcoTran parameters.**

| # | Functional group | TL | Biomass | P/B | C/B | EE |
|---|---|---|---|---|---|---|
| 1 | Large phytoplankton | 1.00 | 45.753 | 215.00 | – | 0.980 |
| 2 | Small phytoplankton | 1.00 | 6.230 | 215.00 | – | 0.945 |
| 3 | Micro-zooplankton | 2.00 | 16.219 | 150.00 | 428.57 | **0.900** |
| 4 | Large copepods | 2.25 | 6.552 | 15.00 | 60.00 | 0.744 |
| 5 | Small copepods | 2.35 | 26.443 | 37.00 | 148.00 | 0.419 |
| 6 | Small invertebrate larvae | 2.35 | 4.813 | 37.00 | 148.00 | 0.573 |
| 7 | Pteropods | 2.53 | 0.272 | 15.00 | 50.00 | 0.759 |
| 8 | Pelagic amphipods | 2.65 | 0.662 | 14.00 | 56.00 | 0.734 |
| 9 | Pelagic shrimp | 3.27 | 18.939 | 3.00 | 12.00 | 0.885 |
| 10 | Other macro-zooplankton | 2.88 | 6.309 | 10.00 | 40.00 | 0.726 |
| 11 | Small jellyfish (net-feeders) | 2.40 | 1.671 | 45.00 | 150.00 | 0.021 |
| 12 | Small jellyfish (carnivores) | 3.48 | 0.054 | 20.00 | 66.67 | 0.023 |
| 13 | Large jellyfish | 3.08 | 0.101 | 15.00 | 60.00 | 0.079 |
| 14 | Pyrosomes | 2.08 | 15.976 | 45.00 | 150.00 | 0.001 |
| 15 | *E. Pacifica* | 2.30 | 32.294 | 6.00 | 24.00 | 0.932 |
| 16 | *T. Spinifera* | 2.30 | 10.234 | 7.00 | 28.00 | 0.799 |
| 17 | Small cephalopod aggregate | 3.63 | 2.640 | 3.00 | 12.00 | **0.850** |
| 18 | Cephalopod humboldt | 4.36 | 0.005 | 2.75 | 11.00 | 0.969 |
| 19 | Smelt aggregate | 3.65 | 7.383 | 1.80 | 7.20 | 0.821 |
| 20 | Shad | 3.34 | 2.081 | 1.13 | 4.53 | **0.900** |
| 21 | Sardine | 3.11 | 1.841 | 1.13 | 4.53 | 0.966 |
| 22 | Herring | 3.17 | 4.099 | 1.80 | 7.20 | 0.943 |
| 23 | Anchovy | 3.19 | 3.038 | 1.80 | 7.20 | 0.860 |
| 24 | Saury | 3.75 | 0.130 | 1.13 | 4.53 | 0.775 |
| 25 | Coho yearling | 4.28 | 0.222 | 1.80 | 7.20 | 0.933 |
| 26 | Chinook yearling spring-run | 4.20 | 0.110 | 1.13 | 4.53 | 0.881 |
| 27 | Chinook yearling fall-run | 4.24 | 0.080 | 1.13 | 4.53 | 0.834 |
| 28 | Chinook subyearling fall-run early | 3.94 | 0.015 | 1.80 | 7.20 | 0.841 |
| 29 | Chinook subyearling fall-run late | 4.19 | 0.094 | 1.80 | 7.20 | 0.820 |
| 30 | Other Chinook yearling | 4.31 | 0.014 | 1.13 | 4.53 | 0.956 |
| 31 | Other Chinook subyearling | 4.07 | 0.057 | 1.80 | 7.20 | 0.961 |
| 32 | Other juvenile salmon | 3.46 | 0.028 | 1.80 | 7.20 | 0.855 |
| 33 | Mesopelagic fish aggregate | 3.36 | 1.245 | 1.75 | 7.00 | **0.850** |
| 34 | Planktivorous rockfish | 3.76 | 6.420 | 0.13 | 1.25 | 0.952 |
| 35 | Coho | 4.17 | 0.230 | 1.80 | 10.59 | 0.739 |
| 36 | Chinook | 4.07 | 0.112 | 0.75 | 4.41 | 0.892 |
| 37 | Other salmon aggregate | 3.98 | 0.019 | 1.90 | 11.18 | 0.756 |
| 38 | Shark aggregate | 4.73 | 0.017 | 0.20 | 3.33 | 0.788 |
| 39 | Jack mackerel | 3.64 | 21.395 | 0.23 | 2.30 | 0.103 |
| 40 | Pacific mackerel | 3.49 | 0.857 | 0.76 | 7.60 | 0.848 |
| 41 | Piscivorous rockfish | 3.88 | 3.072 | 0.17 | 1.72 | 0.980 |
| 42 | Dogfish aggregate | 4.21 | 3.475 | 0.20 | 2.50 | 0.272 |
| 43 | Hake | 3.65 | 18.500 | 0.35 | 3.54 | 0.987 |
| 44 | Tuna aggregate | 4.29 | 0.200 | 0.30 | 3.00 | 0.893 |
| 45 | Sablefish | 4.16 | 1.787 | 0.23 | 2.30 | 0.952 |
| 46 | Hexagrammidae (lingcod greenling) | 4.43 | 0.722 | 0.30 | 3.00 | 0.905 |
| 47 | Flatfish (water-column feeders) | 4.29 | 3.797 | 0.28 | 1.38 | 0.817 |

(*Continued*)

**Table 1.** (Continued)

| # | Functional group | TL | Biomass | P/B | C/B | EE |
|---|---|---|---|---|---|---|
| 48 | Skates & rays | 3.71 | 2.769 | 0.23 | 2.30 | 0.266 |
| 49 | Misc. Small benthic fishes | 3.33 | 8.900 | 0.40 | 4.00 | **0.900** |
| 50 | Benthivorous rockfish | 3.67 | 7.987 | 0.07 | 0.70 | 0.811 |
| 51 | Gadidae (cod haddock pollock) | 3.48 | 0.120 | 0.35 | 3.50 | 0.838 |
| 52 | Flatfish (benthic feeders) | 3.16 | 11.878 | 0.30 | 3.00 | 0.684 |
| 53 | Flatfish (small) | 3.45 | 7.968 | 0.38 | 1.90 | 0.919 |
| 54 | Grenadier | 3.62 | 1.206 | 0.20 | 1.00 | 0.066 |
| 55 | Juvenile rockfish | 3.51 | 1.202 | 2.70 | 10.80 | 0.930 |
| 56 | Juvenile fish (other) | 3.26 | 5.991 | 2.70 | 10.80 | 0.735 |
| 57 | Juvenile fish (chondrichthyes) | 3.44 | 0.535 | 2.70 | 10.80 | **0.850** |
| 58 | Infauna | 2.00 | 80.000 | 4.50 | 18.00 | 0.973 |
| 59 | *Pandalus spp.* | 2.91 | 14.257 | 3.00 | 12.00 | 0.903 |
| 60 | Other epibenthic shrimp (Caridea) | 2.81 | 12.950 | 4.20 | 16.80 | **0.850** |
| 61 | Mysids | 2.83 | 2.637 | 22.00 | 110.00 | **0.850** |
| 62 | Echinoderms | 2.07 | 21.361 | 1.21 | 6.05 | **0.850** |
| 63 | Benthic amphipods isopods and cumaceans | 2.05 | 7.158 | 21.50 | 107.50 | **0.850** |
| 64 | Bivalves | 2.03 | 64.450 | 1.30 | 6.50 | **0.850** |
| 65 | Misc. Epifauna (suspension feeders) | 2.14 | 3.828 | 7.40 | 37.00 | **0.850** |
| 66 | Dungeness crab | 3.27 | 5.109 | 1.50 | 6.00 | 0.952 |
| 67 | Tanner crab | 2.99 | 0.869 | 1.00 | 4.00 | 0.939 |
| 68 | Misc. Epifauna (carnivorous) | 2.67 | 31.550 | 3.00 | 15.00 | **0.850** |
| 69 | Sooty shearwaters | 4.31 | 0.017 | 0.10 | 73.00 | 0.014 |
| 70 | Common murre | 4.37 | 0.015 | 0.17 | 72.00 | 0.210 |
| 71 | Gulls & terns | 3.79 | 0.001 | 0.17 | 73.00 | 0.774 |
| 72 | Alcids | 3.90 | 0.001 | 0.17 | 110.00 | 0.084 |
| 73 | Large pelagic seabirds | 4.08 | 0.001 | 0.07 | 75.00 | 0.128 |
| 74 | Other pelagic seabirds | 4.29 | 0.001 | 0.10 | 73.00 | 0.046 |
| 75 | Coastal seabirds (divers) | 4.29 | 0.001 | 0.16 | 73.00 | 0.218 |
| 76 | Storm-petrels | 3.86 | 0.0001 | 0.12 | 144.00 | 0.072 |
| 77 | Gray whales | 3.72 | 0.146 | 0.06 | 8.90 | 0.002 |
| 78 | Baleen whales | 3.69 | 0.572 | 0.04 | 7.60 | 0.002 |
| 79 | Small pinnipeds | 4.55 | 0.024 | 0.08 | 8.30 | 0.034 |
| 80 | Sea lions | 4.64 | 0.049 | 0.07 | 24.00 | 0.279 |
| 81 | Northern elephant seals | 4.49 | 0.062 | 0.07 | 24.00 | 0.219 |
| 82 | Small toothed whales | 4.44 | 0.072 | 0.10 | 25.80 | 0.112 |
| 83 | Large toothed whales | 4.57 | 0.067 | 0.05 | 6.61 | 0.003 |
| 84 | Other killer whales | 4.94 | 0.0004 | 0.03 | 11.16 | 0.000 |
| 85 | Southern resident killer whales | 5.13 | 0.005 | 0.03 | 11.16 | 0.000 |
| 86 | Invertebrate eggs | 1.00 | 0.00002 | 0.00 | 0.00 | 0.225 |
| 87 | Fish eggs | 1.00 | 1.906 | 0.00 | 0.00 | 0.421 |
| 88 | Pelagic detritus | 1.00 | 10.000 | 0.00 | 0.00 | 0.525 |
| 89 | Fishery offal | 1.00 | 5.000 | 0.00 | 0.00 | 0.684 |
| 90 | Benthic detritus | 1.00 | 10.000 | 0.00 | 0.00 | 0.885 |
| 91 | Dredge | 3.04 | – | – | – | – |
| 92 | Hook & line | 5.16 | – | – | – | – |
| 93 | Other gear | 4.57 | – | – | – | – |
| 94 | Net | 4.34 | – | – | – | – |

(*Continued*)

**Table 1.** (Continued)

| # | Functional group | TL | Biomass | P/B | C/B | EE |
|---|---|---|---|---|---|---|
| 95 | Pot & trap | 4.53 | – | – | – | – |
| 96 | Trolling | 5.23 | – | – | – | – |
| 97 | Trawl (non-shrimp) | 4.67 | – | – | – | – |
| 98 | Shrimp trawls | 3.97 | – | – | – | – |
| 99 | Recreational fishery | 5.07 | – | – | – | – |

Ecopath parameterization of the model. TL = estimated trophic level, Biomass = estimated average biomass density in ecosystem (mt/km$^2$), P/B = weight-specific production rate, C/B = weight-specific consumption rate, EE = ecotrophic efficiency. EE values are estimated by the Ecopath master equation (see Eq 1 in main text), except those in which biomass needed to be estimated (EE in bold). Assimilation Efficiency is 0.8 for all living consumers. Production Efficiency (P/Q) can be calculated as P/B divided by C/B. See https://doi.org/10.5281/zenodo.7079777 for a csv version of this table (which includes a calculated P/Q column) and see S2–S5 Tables for other ecosystem model parameters.

boundaries, or (iii) integrated into the functional group's growth as additional biomass. According to the first law of thermodynamics, EE should be between 0 and 1 so that mass-balance is reached. When EE values greater than one were encountered, we balanced the model by either applying scaling factors to survey biomass or by adjusting the diet matrix. Survey biomasses are notoriously underestimated since many animals can avoid sampling gear, and sampling is unlikely to occur at peak activity in the water column for all species surveyed. Thus, relatively large scaling factors are used for small animals (e.g., invertebrate eggs) that can pass through the net more easily or large animals (e.g., adult salmon) that can avoid nets more easily (i.e., estimates are low) and scaling factors near, or precisely, a value of one were used for acoustic-trawl surveys that use targeted trawls to validate acoustic biomass estimates (i.e., estimated biomass is likely more accurate and precise; see "BiomassScalers.csv" in supplement for scaling factors).

Survey biomasses and the diet matrix are imperfect because they are only a snapshot in time and space. Mass-balancing is inherently subjective as we adjust input parameters based on our understanding of survey/study limitations and the data quality (in time, space, methodology, and sampling effort). We were willing to adjust diets that were parameterized with older data (or with fewer samples) more so than diets with newer and more spatially relevant data (or with higher sample sizes). Most diet adjustments during mass-balancing were caused by a precipitous decline in sardine between this model and the previous version (sardine biomass is lower by a factor of more than 8), which was not reflected in the outdated diet studies (see "MassBalancingDetails.csv" in supplementary data and code for more details).

Since gelatinous zooplankton have a greater water content relative to other functional groups [46], their importance to the trophic network may be overestimated in our model (which is expressed in wet-weight biomass during Ecopath model balancing). Thus, the biomass of gelatinous functional groups and diet contributions of these groups as prey to the next trophic level were scaled such that each unit of small gelatinous zooplankton biomass and large carnivorous jellyfish biomass equaled the water content of crustacean zooplankton and pelagic fishes, respectively [34]. Pyrosome data were collected during three different times of year [47]. Thus, we used assumptions of seasonal changes in abundance and linear interpolation to generate a yearly average pyrosome density, in combination with the above gelatinous conversion factors, to scale down our overall biomass densities for pyrosomes which were originally calculated from the Pre-recruit survey (see "BiomassScalers.csv", "PyrosomeScaling.csv", "FlowChart.pdf", and "MassBalancingDetails.csv" for more details).

## Biomass density

Model parameterization and community composition of the model includes 90 functional groups (2 primary producers, 83 consumers, 5 detritus) and 9 fisheries (Table 2). Biomass densities (Table 1) were estimated from survey data for 65 of these groups, values for another 12 groups were borrowed from a previous NCC ecosystem model [32, 33], and biomasses of the remaining 13 (of the 90 living and detritus groups) were estimated by the model itself (see Eq 2). Below we describe the data sources for the ecosystem model [48] and our procedure for estimating biomass density from the data provided. S1 Table includes a comprehensive list of biomass density data sources for each functional group.

All survey data were quality controlled with input from the collectors and maintainers of the datasets. Failing to include absence data would severely overestimate biomass. If absence data were missing (i.e., if data provided only included positive occurrences), we added zero values for all sampled stations that did not include a positive value for any species that was recorded by the survey at another time and location.

Each survey dataset was used to calculate volumetric biomass densities as weight in metric tons per volume of water sampled in cubic kilometers (mt/km$^3$; details for individual datasets are described below). Volumetric densities were converted to areal densities (mt/km$^2$; i.e., vertically-integrated densities) by multiplying each volumetric density by assumed vertical distributions for each functional group (see Ruzicka et al., 2012, 2016 [32, 33]; depth range assumptions in supplemental data) within each 15 model subregions (Fig 3). We average across individual sampling events within each of the 15 model subregions to obtain values for mass-balancing and as starting points for time dynamic scenarios (Table 1; Fig 4). Because the spatial distribution of many marine organisms is patchy (with infrequent catches of very high abundance or biomass), data can be highly skewed and of high variance. Thus, estimating mean biomass values can be more accurate when based on a log-normal distribution [49]. The delta distribution is a probability distribution used to estimate the mean and variance for a dataset that has both zero and non-zero values; in this case, only the non-zero values follow a log-normal distribution [49]. To derive fish biomass from bottom trawl surveys, we used the delta distribution to account for infrequent catches of large biomass values (highly skewed and of high variance). In contrast, we used the arithmetic mean to derive fish biomass from acoustic-trawl data (i.e., hake and coastal pelagic species, see below) and mid-water tows for zooplankton and other small organisms, as their distribution in the water column is thought to be more uniform (see "RegionalBiomassCalculator.m" in supplemental code).

## Functional group data processing

**Phytoplankton.** Net primary productivity rates were estimated from satellite images and a vertically generalized production model (VGPM) product (SeaWIFS Chl data; [50, 51]). Monthly VGPM products for 2014–2021 were downloaded on March 8, 2022 (http://sites.science.oregonstate.edu/ocean.productivity/index.php). Phytoplankton production rate data were trimmed to our model domain using a shapefile of the domain extent (see supplement) and the R package 'sp'[52]. Units provided for phytoplankton production rates were mg C / m$^2$ day$^{-1}$ and were converted to mt/km$^2$ year$^{-1}$ with phytoplankton-specific carbon–weight conversion factors [53] and standard unit conversions for weight, area, and time (see "BiomassWork/" in the supplemental data and code). Production was then converted to areal biomass density (mt/km$^2$) by dividing by the biomass-specific production rates of phytoplankton used in the model (yearly P/B rates in Table 1). We used average phytoplankton biomass density from April–September to parameterize model initial conditions to match the high productivity months during which most biological surveys occurred

**Table 2. Functional group definitions.**

| Functional group | Group composition |
|---|---|
| Large phytoplankton | >10 um (large chain and centric diatoms) |
| Small phytoplankton | ≤ 10 um (cyanobacteria, dinoflagellates, small diatoms) |
| Micro-zooplankton | Ciliates, flagellate grazers |
| Large copepods | Copepods ≥ 0.025 mg C |
| Small copepods | Copepods < 0.025 mg C |
| Small invertebrate larvae | Copepods (nauplii), small crustacean larvae (zoea, cypids), euphausiid (larvae), mollusk larvae (veligers), echinoderm larvae (pluteus), other invert larvae |
| Pteropods | Order: Pteropoda |
| Pelagic amphipods | Hyperiidae, Gammaridae |
| Pelagic shrimp | Sergestidae, Penaeidae |
| Other macro-zooplankton | Chaetognaths, large crustacean larvae (megalopae), ichthyoplankton, other macro-zooplankton (pelagic polychaetes, heteropods, ostracods, cladocerans) |
| Small jellyfish (net-feeders) | Urochordates (larvaceans, salps) |
| Small jellyfish (carnivores) | Ctenophores, misc. Small medusae |
| Large jellyfish | Sea nettle (*Chrysaora fuscescens*), moon jelly (*Aurelia labiata*), egg yolk jelly (*Phacellophora camtschatica*), water jelly (*Aequorea spp.*), lion's mane jelly (*Cyanea capillata*) |
| Pyrosomes | *Pyrosoma atlanticum* |
| *E. pacifica* | *Euphausia pacifica* (adult & juveniles) |
| *T. spinifera* | *Thysanoessa spinifera* (adult & juveniles) |
| Small cephalopods | Market squid (*Doryteuthis opalescens*) |
| Humboldt squid | Humboldt squid (*Dosidicus gigas*) |
| Smelt aggregate | Pacific sand lance (*Ammodytes hexapterus*), jacksmelt/silversides (*Atherinopsis californiensis*), eulachon (*Thaleichthys pacificus*), night smelt (*Spirinchus starksi*), longfin smelt (*Spirinchus thaleichthys*), surf smelt (*Hypomesus pretiosus*), whitebait smelt (*Allosmerus elongates*), popeye blacksmelt (*Bathylagus ochotensis*) |
| Shad | American shad *(Alosa sapidissima)* |
| Sardine | Pacific sardine *(Sardinops sagax)* |
| Herring | Pacific herring *(Clupea pallasii)* |
| Anchovy | Northern anchovy *(Engraulis mordax)* |
| Saury | Pacific saury *(Cololabis saira)* |
| Juvenile coho | Juvenile Coho salmon (*Oncorhynchus kisutch*) yearling |
| Juvenile Chinook Y spring | Juvenile Chinook salmon (*Oncorhynchus tshawytscha*) spring-run yearlings (Columbia River + Washington coast stocks) |
| Juvenile Chinook Y fall | Juvenile Chinook salmon *(O. tshawytscha)* fall-run yearlings (Columbia River + Washington coast stocks) |
| Juvenile Chinook SY fall early | Juvenile Chinook salmon *(O. tshawytscha)* fall-run subyearlings; early ocean migrants (May and June JSOES surveys; Columbia River + Washington coast stocks) |
| Juvenile Chinook SY fall late | Juvenile Chinook salmon *(O. tshawytscha)* fall-run subyearlings; late ocean migrants (September JSOES surveys; Columbia River + Washington coast stocks) |
| Other juv. Chinook Y | All other juvenile Chinook salmon *(O. tshawytscha)* yearlings |
| Other juv. Chinook SY | All other juvenile Chinook salmon *(O. tshawytscha)* subyearlings |
| Other juvenile salmon | All juvenile salmon not described above: pink *(Oncorhynchus gorbuscha)*, chum *(O. keta)*, sockeye *(O. nerka)*, steelhead *(O. mykiss)* |
| Mesopelagic fish aggregate | Myctophidae, Bathylagidae, Lophotidae (Crestfishes), Ophidiidae (cusk eel), Paralepididae (barracudina), Stomiidae (dragonfish), Trachipteridae (ribbonfishes), Nemichthyidae (snipe eels) |

*(Continued)*

**Table 2.** (Continued)

| Functional group | Group composition |
|---|---|
| Planktivorous rockfish | Aurora *(Sebastes aurora)*, bank *(S. Rufus)*, blue *(S. Mystinus)*, darkblotched *(S. Crameri)*, greenstriped *(S. Elongates)*, harlequin *(S. Variegatus)*, Pacific Ocean perch *(S. Alutus)*, Puget Sound *(S. Emphaeus)*, pygmy *(S. Wilsoni)*, redstripe *(S. Proriger)*, rosy *(S. Rosaceus)*, sharpchin *(S. Zacentrus)*, shortbelly *(S. Jordani)*, splitnose *(S. Diploproa)*, stripetail *(S. Saxicola)*, widow *(S. Entomelas)*, yellowmouth *(S. Reedi)* |
| Coho | Adults: *(Oncorhynchus kisutch)* |
| Chinook | Adults: *(Oncorhynchus tshawytscha)* |
| Other salmon aggregate | Adults: pink *(Oncorhynchus gorbuscha)*, chum *(O. keta)*, sockeye *(O. nerka)*, steelhead *(O. mykiss)*, cutthroat trout *(O. mykiss)* |
| Shark aggregate | Tope (a.k.a soupfin; *Galeorhinus galeus)*, blue *(Prionace glauca)*, thresher *(Alopias vulpinus)*, salmon *(Lamna ditropis)*, shortfin mako *(Isurus oxyrinchus)* |
| Jack mackerel | Jack mackerel *(Trachurus symmetricus)* |
| Pacific mackerel | Pacific chub mackerel *(Scomber japonicus)* |
| Piscivorous rockfish | Black *(Sebastes melanops)*, blackgill *(S. Melanostomus)*, bocaccio *(S. Paucispinis)*, canary *(S. Pinniger)*, chilipepper *(S. Goodie)*, yelloweye *(S. Ruberrimus)*, yellowtail *(S. Flavidus)* |
| Dogfish aggregate | Spiny dogfish *(Squalus acanthias)*, brown catshark *(Apristurus brunneus)*, filetail catshark *(Parmaturus xaniurus)*, Pacific sleeper shark *(Somniosus pacificus)* |
| Hake | Pacific hake *(Merluccius productus)* |
| Tuna aggregate | Albacore *(Thunnus alalunga)*, Pacific barracuda *(Sphyraena argentea)*, bigeye tuna *(T. obesus)*, bluefin tuna *(T. thynnus)*, Bramidae (pomfret), Carangidae (jacks, pompanos), yellowtail tuna *(T. albacares)*, Pacific bonito *(Sarda chiliensis)* |
| Sablefish | Sablefish *(Anoplopoma fimbria)* |
| Hexagrammidae (lingcod greenling) | Lingcod *(Ophiodon elongates)*, greenling *(Hexagrammos decagrammus)* |
| Flatfish (water-column feeders) | Pacific halibut *(Hippoglossus stenolepis)*, arrowtooth flounder *(Atheresthes stomias)*, petrale sole *(Eopsetta jordani)*, California halibut *(Paralichthys californicus)* |
| Skates & rays | Bat ray *(Myliobatis californica)*, big skate *(Raja binoculata)*, black skate *(Bathyraja trachura)*, Pacific electric ray *(Torpedo californica)*, longnose skate *(Raja rhina)*, Pacific angelshark *(Squatina californica)*, spotted ratfish *(Hydrolagus colliei)* |
| Misc. Small benthic fishes | Agonidae (poachers), Bathymasteridae (ronquils), Batrachoididae (Toadfishes), Blenniidae (blennies), Cottidae (sculpins), Cyclopteridae (lumpfish), Embiotocidae (surfperch), Gasterosteidae (sticklebacks), Gobiidae (gobies), hagfish, Kyphosidae (sea chubs), lamprey eels, Liparidae (snailfish), Moronidae (striped bass), Pholidae (gunnels), prowfish, Sciaenidae (drums, croakers), Stichaeidae (prickleback), Syngnathidae (pipefishes), Triglidae (Searobins), Zoarcidae (eelpout) |
| Benthivorous rockfish | Cabezon *(Scorpaenichthys marmoratus)*, longspine thornyhead *(Sebastolobus altivelis)*, shortspine thornyhead *(Sebastolobus alascanus)*, copper *(Sebastes caurinus)*, quillback *(S. maliger)*, redbanded *(S. babcocki)*, rosethorn *(S. helvomaculatus)*, rougheye *(S. aleutianus)*, shortraker *(S. borealis)*, silvergray *(S. brevispinis)*, tiger *(S. nigrocinctus)* |
| Gadidae (cod haddock pollock) | Pacific cod *(Gadus macrocephalus)*, walleye pollock *(Theragra chalcogramma)*, Pacific tomcod *(Microgadus proximus)* |
| Flatfish (benthic feeders) | English sole *(Parophrys vetulus)*, Dover sole *(Microstomus pacificus)*, rex sole *(Glyptocephalus zachirus)* |
| Flatfish (small) | Butter sole *(Isopsetta isolepis)*, curlfin sole *(Pleuronichthys decurrens)*, deepsea sole *(Embassichthys bathybius)*, flathead sole *(Hippoglossoides elassodon)*, rock sole *(Lepidopsetta bilineata)*, sand sole *(Trulla capensis)*, sanddabs *(Citharichthys spp.)*, slender sole *(Lyopsetta exilis)*, starry flounder *(Platichthys stellatus)*, Pleuronectidae (turbot) |

*(Continued)*

**Table 2.** (Continued)

| Functional group | Group composition |
|---|---|
| Grenadier | Giant grenadier *(Albatrossia pectoralis)*, Pacific grenadier *(Coryphaenoides acrolepis)* |
| Juvenile rockfish | Thornyheads, Greenlings, Painted greenling, Lingcod, Sculpins, Irish lords, Red Irish lord, Brown Irish lord, Cabezon, All rockfish (Sebastidae) |
| Juvenile fish (other) | American shad, Pacific herring, Pacific sardine, Northern anchovy, Pacific hake, Pacific tomcod, Sablefish, Wolf eel, Eel leptocephalus, Jack mackerel, Pacific chub mackerel, Pacific sanddab, Speckled sanddab, Righteye flounders, Arrowtooth flounder, Deepsea sole, Rex sole, Flathead sole, Butter sole, Rock sole, Slender sole, Dover sole, English sole, Soles/Turbits/Flounders, Curlfin turbot, Sand sole |
| Juvenile fish (chondrichthys) | Juvenile sharks, skates, and rays |
| Infauna | Misc. Benthic organisms living in ocean floor substrate |
| *Pandalus* spp. | Pink shrimp *(Pandalus jordani)* |
| Other epibenthic shrimp (caridea) | *Crangon spp.*, *Callianassa spp.*, *Pasiphaea pacifica* |
| Mysids | Mysids & cumaceans |
| Echinoderms | Red sea urchin *(Mesocentrotus franciscanus)*, purple sea urchin *(Strongylocentrotus purpuratus)*, misc. brittle stars, misc. Sea cucumbers, (NOTE: does not include starfish) |
| Benthic amphipods isopods and cumaceans | Benthic amphipods, isopods, cumaceans |
| Bivalves | Basket cockle *(Clinocardium nuttallii)*, butter clam *(Saxidomus gigantean)*, California mussel *(Mytilus californianus)*, gaper clam *(Tresus capax)*, Manila clam *(Venerupis philippinarum)*, native littleneck clam *(Leukoma staminea)*, rock scallop *(Crassadoma gigantean)*, Weathervane scallops *(Patinopecten caurinus)*, Pacific oyster *(Crassostrea gigas)*, razor clam *(Siliqua patula)*, soft-shelled clam *(Mya arenaria)*, purple varnish clam *(Nuttallia obscurata)*, rough paddock *(Zirfaea pilsbryi)*, flat tipped piddock *(Penitella penita)* |
| Misc. Epifauna (suspension feeders) | Barnacles, bryozoans, sea anemones |
| Dungeness crab | Dungeness crab *(Cancer magister)* |
| Tanner crab | Bairds tanner crab *(Chionoecetes bairdi)*, grooved tanner crab *(Chionoecetes tanneri)*, triangle tanner crab *(Chionoecetes angulatus)* |
| Misc. Epifauna (carnivorous) | Misc. Small crabs, misc. Gastropods, starfishes |
| Sooty shearwaters | Sooty shearwaters *(Puffinus griseus)* |
| Common murre | Common murre *(Uria aalge)* |
| Gulls & terns | California gull *(Larus californicus)*, glaucous-winged gull *(L. Glaucescens)*, Heermann's gull *(L. heermanni)*, herring gull *(L. Argentatus)*, ring-billed gull *(L. Delawarensis)*, Sabine's gull *(Xema sabini)*, western gull *(L. Occidentalis)*, hybrid gulls, arctic tern *(Sterna paradisaea)*, Caspian tern *(S. caspia)*, common tern *(S. Hirundo)* |
| Alcids | Cassin's auklet *(Ptychoramphus aleuticus)*, rhinoceros auklet *(Cerorhinca monocerata)*, pigeon guillemot *(Cepphus columba)*, marbled murrelet *(Brachyramphus marmoratus)*, ancient murrelet *(Synthliboramphus antiquus)*, tufted puffin *(Fratercula cirrhata)*, horned puffin *(F. Corniculata)* |
| Large pelagic seabirds | Black-footed albatross *(Phoebastria nigripes)*, Laysan albatross *(Phoebastria immutabilis)*, parasitic jaeger *(Stercorarius parasiticus)*, northern fulmar *(Fulmarus glacialis)*, skuas, petrels |
| Other pelagic seabirds | Buller's shearwater *(Puffinus bulleri)*, flesh-footed shearwater *(Puffinus carneipes)*, pink-footed shearwater *(Puffinus creatopus)*, red-necked Phalarope *(Phalaropus lobatus)*, other murres |
| Coastal seabirds (divers) | Brown pelican *(Pelecanus occidentalis)*, white pelican *(Pelecanus erythrorhynchos)*, Brandt's cormorant *(Phalacrocorax penicillatus)*, double-crested cormorant *(Phalacrocorax auritus)*, pelagic cormorant *(Phalacrocorax pelagicus)*, western grebe *(Aechmophorus occidentalis)*, Clark's grebe *(Aechmophorus clarkii)* |

*(Continued)*

**Table 2.** (Continued)

| Functional group | Group composition |
|---|---|
| Storm-petrels | Fork-tailed storm petrel *(Oceanodroma furcata)*, Leach's storm-petrel *(Oceanodroma leucorhoa)* |
| Gray whales | Gray whales *(Eschrichtius robustus)* |
| Baleen whales | Minke whale *(Balaenoptera acutorostrata)*, humpback whale *(Megaptera novaeangliae)*, sei whale *(Balaenoptera borealis)*, fin whale *(Balaenoptera physalus)*, blue whale *(Balaenoptera musculus)* |
| Small pinnipeds | Harbor seal *(Phoca vitulina richardsi)*, Northern fur seal *(Callorhinus ursinus)* |
| Sea lions | California sea lion *(Zalophus californicus)*, Steller's sea lion *(Eumetopias jubatus)* |
| N. elephant seals | Northern elephant seal *(Mirounga angustirostris)* |
| Small toothed whales | Harbor porpoise *(Phocoena phocoena)*, Dall's porpoise *(Phocoenoides dalli)*, Pacific white-sided dolphin *(Lagenorhynchus obliquidens)*, Risso's dolphin *(Grampus griseus)*, northern right whale dolphin *(Lissodelphis borealis)* |
| Large toothed whales | Sperm whale *(Physeter catadon)*, pilot whale *(Globicephala macrorhynchus)*, Baird's beaked whale *(Berardius bairdii)*, mesoplodon beaked whale *(Mesoplodon spp.)*, Cuvier's beaked whale *(Ziphius cavirostris)* |
| Other killer whales | Killer whales *(Orcinus orca)* |
| SRKW | Southern resident killer whales *(Orcinus orca)* |
| Invertebrate eggs | Invertebrate eggs |
| Fish eggs | Fish eggs |
| Pelagic detritus | Pelagic detritus |
| Fishery offal | Fishery offal |
| Benthic detritus | Benthic detritus |
| Dredge | Dredge fishery |
| Hook & line | Hook and line fishery |
| Other gear | Other gear / miscellaneous |
| Net | Non-trawl nets (e.g., Gill-net, etc.) |
| Pot & trap | Pots and traps |
| Trolling | Trolling fishery |
| Trawl (non-shrimp) | Non-shrimp trawls |
| Shrimp trawls | Shrimp trawls |
| Recreational fishery | Recreational fishing |

Functional group names and description of composition. See S1 Table for a description of the data sources. See supplemental data for csv version of table.

(see S1 Appendix: Biomass sources). The phytoplankton size composition–large (diatom; >10 μm) and small (flagellate; ≤ 10 μm)–were estimated from the total phytoplankton concentration using a relation between the fraction of small phytoplankton and total Chlorophyll a from observations of the Newport Hydrographic Line (see S1 Appendix: Newport Hydrographic Line for a description of the survey; fraction $P_{small} = 0.30821 \times [\text{Chl a (mg m}^{-3})]^{-0.82351}$; as described in [32].

**Zooplankton, jellies, and pyrosomes.** Zooplankton were parameterized with the Juvenile Salmon and Ocean Ecosystem Survey (JSOES), Newport Hydrographic Line, and Pre-recruit surveys (see S1 Appendix for each survey description). When individual species data came from multiple surveys (see below), calculated biomass densities (for each subregion; Fig 3) were first averaged across surveys (where sampling was spatially overlapping) and then summed within functional groups as average biomass densities for the entire functional group (Tables 1 and 2).

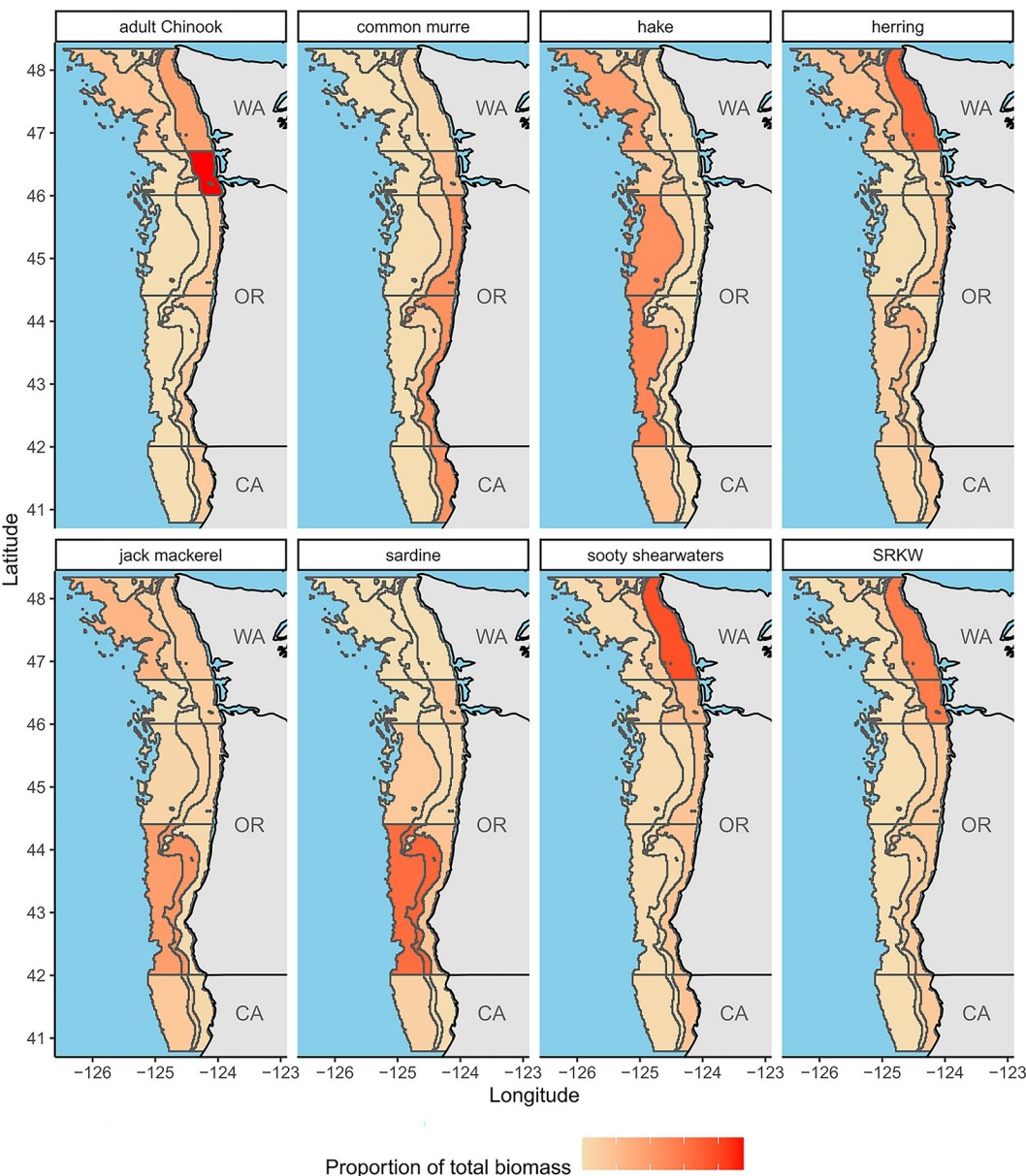

**Fig 3. Subregional heatmaps of functional group spatial distributions.** Each functional group was broken out into the 15 subregions via survey data, fisheries landings data, or species distribution models (see Methods), or distributional assumptions when there was a lack of available information. The color of each subregional cell is a gradient denoting the proportion of biomass (for each functional group) that is within each subregional cell (with red being the highest and pale yellow being the lowest proportions, respectively). The proportion of biomass in each subregion sums to 1 across all subregions. See "SubRegions/" in supplemental data and code to reproduce this plot, and for plots of all functional groups). Adult Chinook, common murre, and sooty shearwater distributions are based off of the juvenile salmon and ocean ecosystem survey (JSOES); hake distributions are from the hake acoustic trawl survey; herring, jack mackerel, and sardine distributions are from the coastal pelagic species (CPS) acoustic trawl survey; and Southern resident killer whale (SRKW) distributions are based off of movement data from satellite-tagged Southern resident killer whales [130]. State outline data comes from US Department of Commerce, Census Bureau, Cartographic Boundary Files.

Crustacean larvae, euphausiid larvae, pelagic (hyperiid and gammarid) amphipods, chaetognaths, ichthyoplankton, and fish eggs were sampled with JSOES bongo nets; copepods, pteropods, ostracods, cladocerans, polychaetes, urochordates, ctenophores and small medusae,

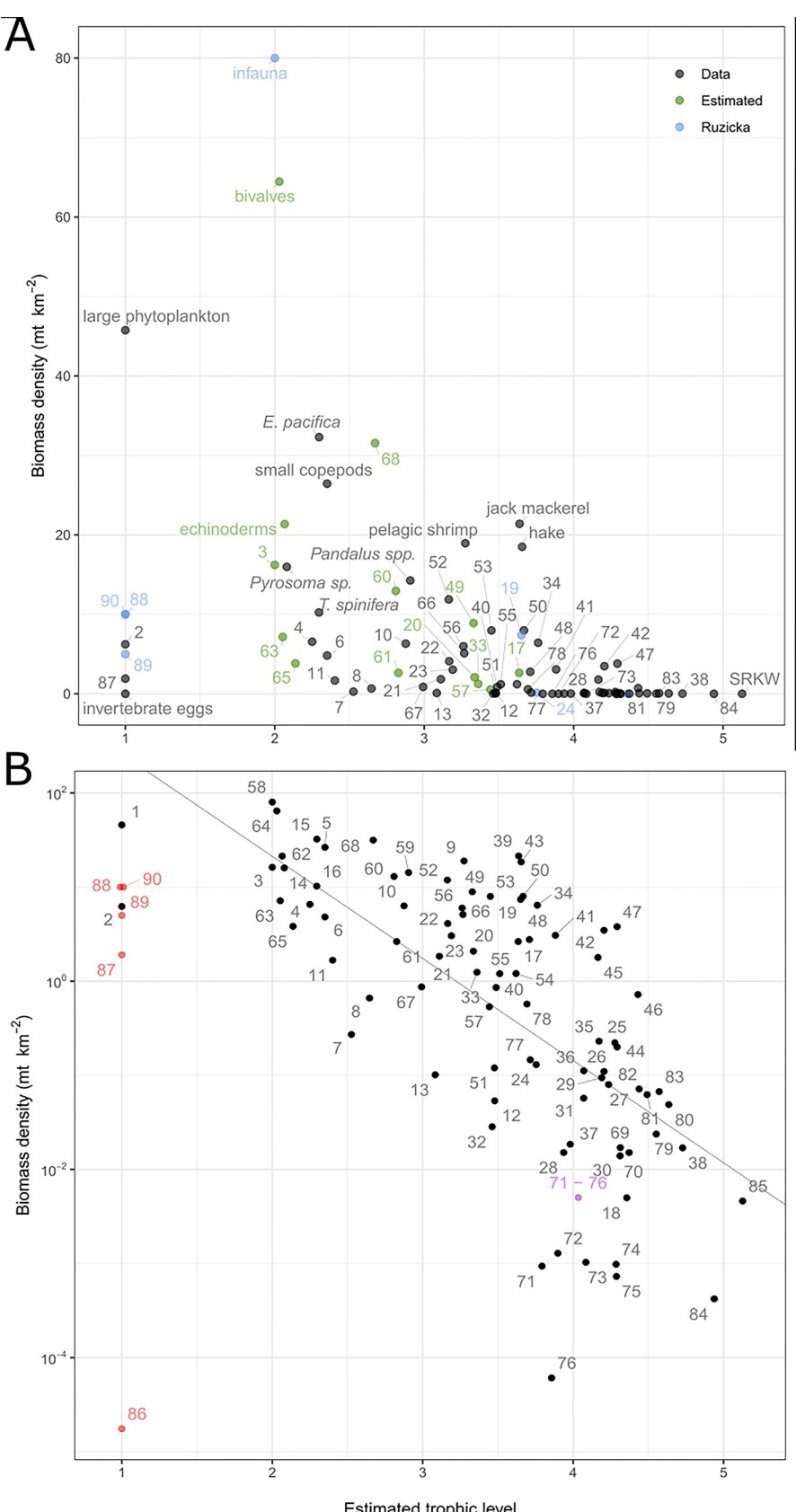

**Fig 4. Biomass density and trophic level of model functional groups.** A) Ecosystem-wide biomass density input values (y-axis) for the aggregated ecosystem model (no subregion-specific values) are based on survey data / stock assessments (black points), were borrowed from a previous ecosystem model (Ruzicka et al. 2012; blue points), or are estimated by the model during mass-balancing (green points; see methods). The trophic level of each functional group is estimated by the model and is based on the diet matrix. Numbers indicate functional groups, identified in Table 1. B) Same as A, but on a log (base 10) axis scale. Red points indicate detritus groups, which are not used in estimation of the regression line (Link 2010). Purple point is aggregated seabird groups 71–76 to display how the choice to aggregate groups affects how far away from the line the point falls. Equation for slope is estimated only with black points and is defined as log(y) = 3.5–1.085 x.

mollusk larvae, echinoderm larvae, other small invertebrate larvae and eggs, and other macro-zooplankton were sampled with JSOES vertical nets; and large jellies came from the JSOES fish trawl nets (all data from 2014–2021).

Additional data on small invertebrate larvae and invertebrate eggs from the Newport Hydrographic Line (2014–2020; May–July) were used to calculate average biomass density values (along with the JSOES data) for all transects that fell within overlapping subregions (Figs 1 and 3). Biomass density values for both surveys were provided in units of mg C/m$^3$ and converted into mt/km$^3$ using group-specific carbon-to-dry-weight and dry-weight-to-wet-weight conversion factors (Ruzicka et al., 2016 [33]; see supplemental code at https://doi.org/10.5281/zenodo.7079777 under directory: Reproducibility/BiomassWork).

Additional data on pelagic amphipods, ctenophores, small medusae, and large jellies from the Pre-recruit survey (2014–2019) were used to calculate average biomass density values (along with the JSOES data) for all transects that fell within overlapping subregions. The Pre-recruit survey (2017–2018) data were also used to parameterize pyrosome biomass. Pre-recruit data were provided as counts per trawl with corresponding length data for a subset of up to 30 specimens per trawl. Counts per trawl were converted to volumetric biomass densities by first converting average lengths (or bell diameter; provided by survey) to average weight with standard species-specific length-weight relationships [54] (see "CountsToBiomass.R" in supplemental data and code) and then multiplying by counts to get total biomass (for each species) from each trawl. Biomass was converted to density by dividing by the volume of water sampled during each trawl.

**Krill and shrimp.** Areal krill biomass densities (mt/km$^2$) were provided by the hake survey (2015, 2017, and 2019; see S1 Appendix: Pacific Hake Ecosystem Acoustic Trawl Survey for survey description). Because krill biomass estimates from the hake survey are not resolved to species level (due to the overlap in the frequency response of *Thysanoessa spinifera* and *Euphausia pacifica*), we used krill species data from the Pre-recruit survey (2019) along with the Newport Hydrographic Line (2014–2016) to apportion krill biomass from the hake survey into two functional groups (*E. pacifica* and *T. spinifera*; Table 2) based on the average relative proportion of each species observed (see "KrillBreakout.csv" in supplemental data and code). We used data from the Pre-recruit survey (2014–2019) to parameterize biomass for pink shrimp (*Pandalus* spp.) and other pelagic shrimp (Sergestidae, Penaeidae), which were provided as counts and lengths and converted to biomass following the description under the "Zooplankton, jellies, and pyrosomes" subheading.

**Juvenile fish.** We used data from the Pre-recruit survey (2014–2019) to parameterize biomasses for all juvenile fish groups (except salmonids; see below), which were provided as counts and lengths and converted to biomass following the description under the "Zooplankton, jellies, and pyrosomes" subheading above. Individual species data were then summed within functional groups as total group volumetric biomass densities (Tables 1 and 2; see "Biomass_Aggregation/" in Zenodo repository).

**Coastal pelagic fish.** Data for Pacific sardine, Pacific herring, Pacific chub mackerel, Pacific jack mackerel, and northern anchovy were provided by the SWFSC coastal pelagic species acoustic-trawl survey [55, 56] as areal biomass densities (in mt/km$^2$; from 2015–2019; see S1 Appendix: Coastal Pelagic Species).

**Salmonids.** We used 2014–2019 data from the JSOES to parameterize juvenile salmonids into eight distinct functional groups and adult salmonids into three groups based on species and life history characteristics (Table 2). The EcoTran platform does not currently support linked life stages, so juvenile salmonid groups do not recruit into adult salmonid groups. Juvenile coho salmon comprise their own group, juvenile Chinook salmon were split into six groups based on life history and ocean entry timing (see below), and all other juvenile salmonids species that individually constitute a small percent of the total catch (<5%; e.g., steelhead, sockeye, chum) made up the final juvenile salmonid functional group (Table 2). Juvenile Chinook salmon were distinguished as spring versus fall run-type Chinook salmon based on genetic stock information [57, 58] and broken out by life history (i.e., sub-yearlings and yearlings; based on fork-length and time of year; [59]), and ocean entry timing (i.e., May/June and September; [58, 60]), which influences their migration behaviors and residence time in the NCC (and thus their trophic interactions with other functional groups). JSOES data on salmonids were provided as counts per trawl with length and weight information for individuals. Counts were multiplied by weight to get a biomass value for each trawl and then divided by the total volume sampled to calculate biomass density (mt/km$^3$) values for each functional group.

**Hake.** Areal biomass densities (mt/km$^2$) were provided (2015, 2017, and 2019) for age 2 + hake from the Pacific Hake Ecosystem Acoustic Trawl Survey [61, 62]. Juvenile hake were provided by the Pre-recruit survey (2014–2019) as counts and lengths and converted to biomass following the description under the "Zooplankton" subheading above. Calculated biomass densities (within each model subregion) from the hake survey and Pre-recruit survey were then summed.

**Groundfish and crabs.** On 28 July 2021, we accessed the groundfish survey (see S1 Appendix: West Coast Groundfish Bottom Trawl Survey) database via the R package 'nwfscSurvey'[63] and downloaded data from 2014–2019 on demersal species of interest (Table 2, S1 Table). Data were provided as areal biomass densities (kg/km$^2$), which were divided by 1000 to convert to consistent units for the model (mt/km$^2$). Individual species data were then summed within functional groups as total group areal biomass densities (Tables 1 and 2).

**Seabirds.** We used seabird species data from JSOES during 2015–2019 to parameterize this model. Data were provided as count densities, in individuals per square kilometer, and were converted to areal biomass densities (mt/km$^2$) by multiplying by the average weight of individual birds, which were obtained from the Sibley Guide to Birds [64] and the Cornell Lab of Ornithology's All About Birds webpage [65]. Individual species data were then summed within functional groups as total group biomass densities (Tables 1 and 2).

**Marine mammals.** Coastwide or stock-specific marine mammal biomasses were obtained from stock assessments [66, 67] and were used to scale regional biomass values calculated in a previous NCC ecosystem model [32, 33]. In cases where population stock assessments have not been updated, or populations have not drastically changed, the values were used directly from the previous ecosystem model.

**Model-estimated groups.** While biomass for most groups was estimated with data, there were a few groups for which we lacked survey data. Additionally, in a couple of cases where we did have some data, surveys were inadequate samplers for particular functional groups (see below)–we instead either allowed the model to estimate biomass by fixing EE values (0.85–0.9;

see Table 1 for a list of model-estimated groups) or by using biomass values from the previous NCC ecosystem model [32, 33]. EE values describe the proportion of a functional group's production that is used by higher trophic levels (see 'Mass balancing' above).

Two groups sampled by the groundfish survey (see S1 Appendix: West Coast Groundfish Bottom Trawl Survey for survey descriptions) were estimated by the model–the mesopelagic fish aggregate was likely above (in the water column) the bottom-trawl net, and the miscellaneous small benthic fishes were likely too small and avoidant to be adequately sampled by the survey. Epibenthic shrimp (Caridea) were similarly not well sampled by the Pre-recruit survey, which relies on a nighttime midwater net trawl that uncommonly captures small shrimp in the benthos; thus, this group was also estimated by the model. Two groups of medium pelagic fish (smelt and saury) were not well sampled by the JSOES, so we relied on biomass estimates from a previous NCC model [33]. Similarly, market squid are caught by the Pre-recruit survey and JSOES, but neither survey provides an adequate absolute estimate of biomass. These cases were not surprising as none of the listed groups were designed to be sampled by the respective surveys. Some benthic groups (such as bivalves; echinoderms; and other carnivorous epifauna such as small crabs, sea stars, and gastropods) have not been well-sampled recently [68, 69]. The remaining groups are not sampled (to our knowledge) by any survey and were estimated by the model: juvenile chondrichthyes and other benthic groups (mysids; suspension feeders such as barnacles, bryozoans, and sea anemones; benthic amphipods, isopods, and cumaceans).

## Diet matrix

The diet matrix partitions each predator's consumption rate amongst its prey (Fig 5). Together with the biomass densities informed in the model and the other trophic flow parameters [also referred to as physiology parameters (e.g., P/B, C/B, etc.); see Eq 2, Table 1], it expresses the biomass flows among functional groups (Figs 6 and 7).

Diets were compiled as follows (see "DietWork/" in supplementary code). For each data source, unidentified material was removed from analyses, and observations were summed (across all individuals) by weight of prey (or % frequency of occurrence (FO) for marine mammals and kelp greenling) and then divided by the total weight (or % FO) of identified prey that was consumed to get a relative (standardized) proportion of each prey in the diet for a given predator species.

If multiple sources were used for one predator species, the resulting predator species' diet was computed as an average of diets calculated from each source, weighting the latter by the sample size (number of predator individuals) within each source. Then predators within the same functional group were combined, with each predator species contributing to a given functional group relative to (or weighted by) the proportion of biomass that a given species contributed to said functional group. For example, northern fur seals and harbor seals make up one functional group (small pinnipeds), but harbor seals account for 98.9% of the small pinniped functional group biomass in our model, whereas northern fur seals only account for 1.1%. Thus, the small pinniped functional group diet consists of harbor seal diet proportions multiplied by 0.989 and northern fur seal diets multiplied by 0.011 summed together.

If a prey grouping from a source was broader than our functional groups (e.g., "cephalopod" includes squid and octopuses, which are in separate functional groups in EcoTran), the relative proportion of prey going into each group was allocated proportionally to the biomass of the groups if they have been observed at least once in the diet of the predator. This procedure assumes that the more abundant prey are more readily available for consumption [31] (see "Build_Diet_Matrix.R" in supplemental data and code).

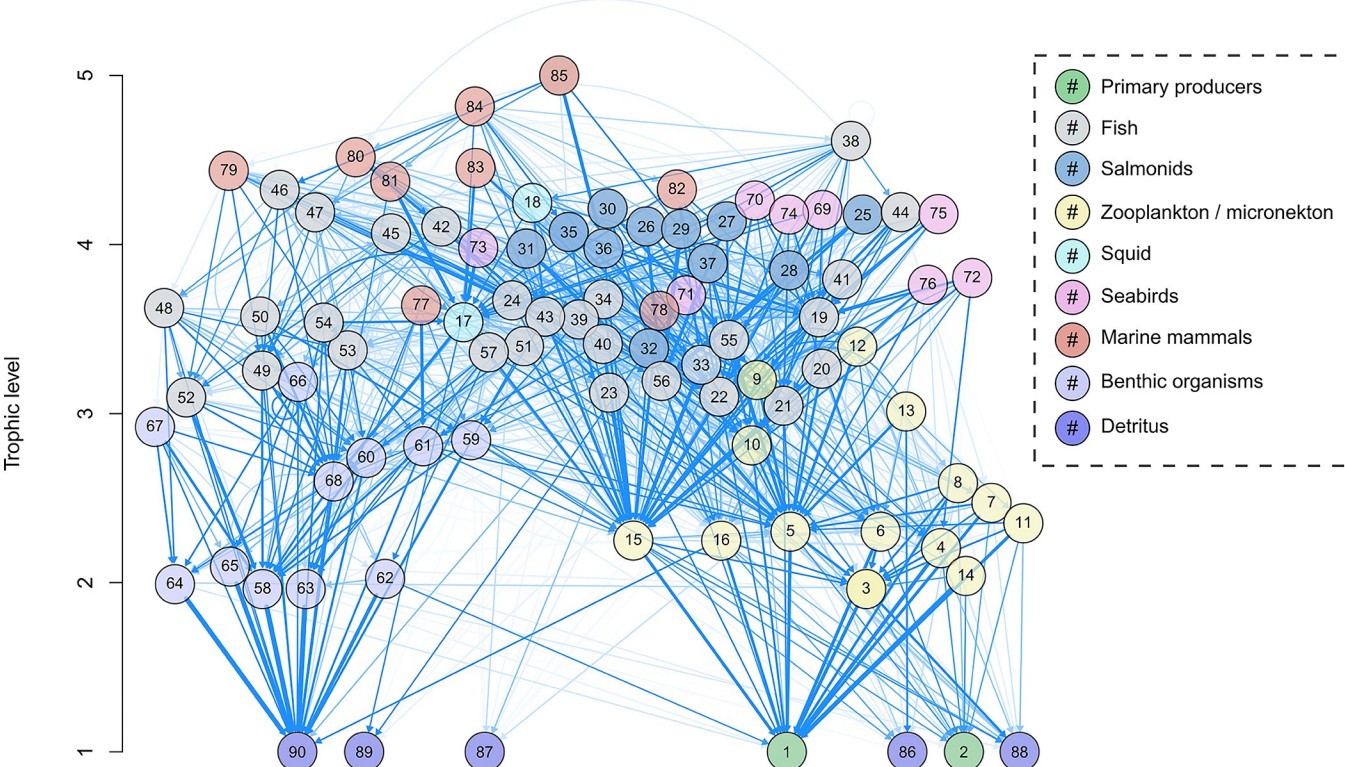

**Fig 5. Diet matrix of ecosystem model by trophic level.** The diet matrix is visualized here as a weighted, directed graph. Numbered nodes are functional groups (see Table 1 for functional group numbers), whereas arrows indicate directed edges from predator group towards prey group. The shade of blue indicates strength of interaction (higher diet preference results in darker blue network edges) up to a value of 0.10, at which point network edges get thicker with higher values. This aesthetic choice was made to not overly clutter the diagram and to make visualization of strong interactions more apparent. The y-axis values are the estimated trophic level of each functional group, and the x-axis is value-less and only used to help visualize multiple groups.

Diet information was obtained from local NCC studies to the extent possible. Data from small pinnipeds (Northern fur seals and some harbor seal diets), sharks (blue shark, common thresher shark, shortfin mako shark), Pacific spiny dogfish, California market squid, Northern anchovy, Pacific herring, Pacific jack mackerel, Pacific chub mackerel, Pacific sardine, Pacific saury, sablefish and hexagrammids (kelp greenling and lingcod) were taken from the California Current Trophic Database (hereafter 'trophic database'; [70]; https://oceanview.pfeg.noaa.gov/cctd/; see 'Diet sources' in the S1 Appendix). Other harbor seal diets [71, 72] and other Pacific spiny dogfish diets [73] came from the literature and were combined with data from the trophic database as described above (see S1 Table).

Juvenile salmonid diet data comes from the JSOES, which contains genetic stock information [74], and allows the diets to be broken out by specific functional groups (see the S1 Appendix: Juvenile salmon diets and Table 2). Diet data for jellyfish, zooplankton, invertebrates, seabirds, cetaceans, and many fish groups were taken from an early version of the NCC food web [32]. See the S1 Appendix: Diet sources for more information and S1 Table for a complete list of data sources for each functional group.

In addition, for all groups mentioned above, diets obtained from the trophic database and the literature were arithmetically averaged with an early version of the food web of Ruzicka et al. [32]. The inclusion of more sources of information helps ensure that diets (which are always a product of imperfect sampling in time and space) are as diverse as possible, which more likely reflects the possibility that predators are consuming a diverse prey field, depending

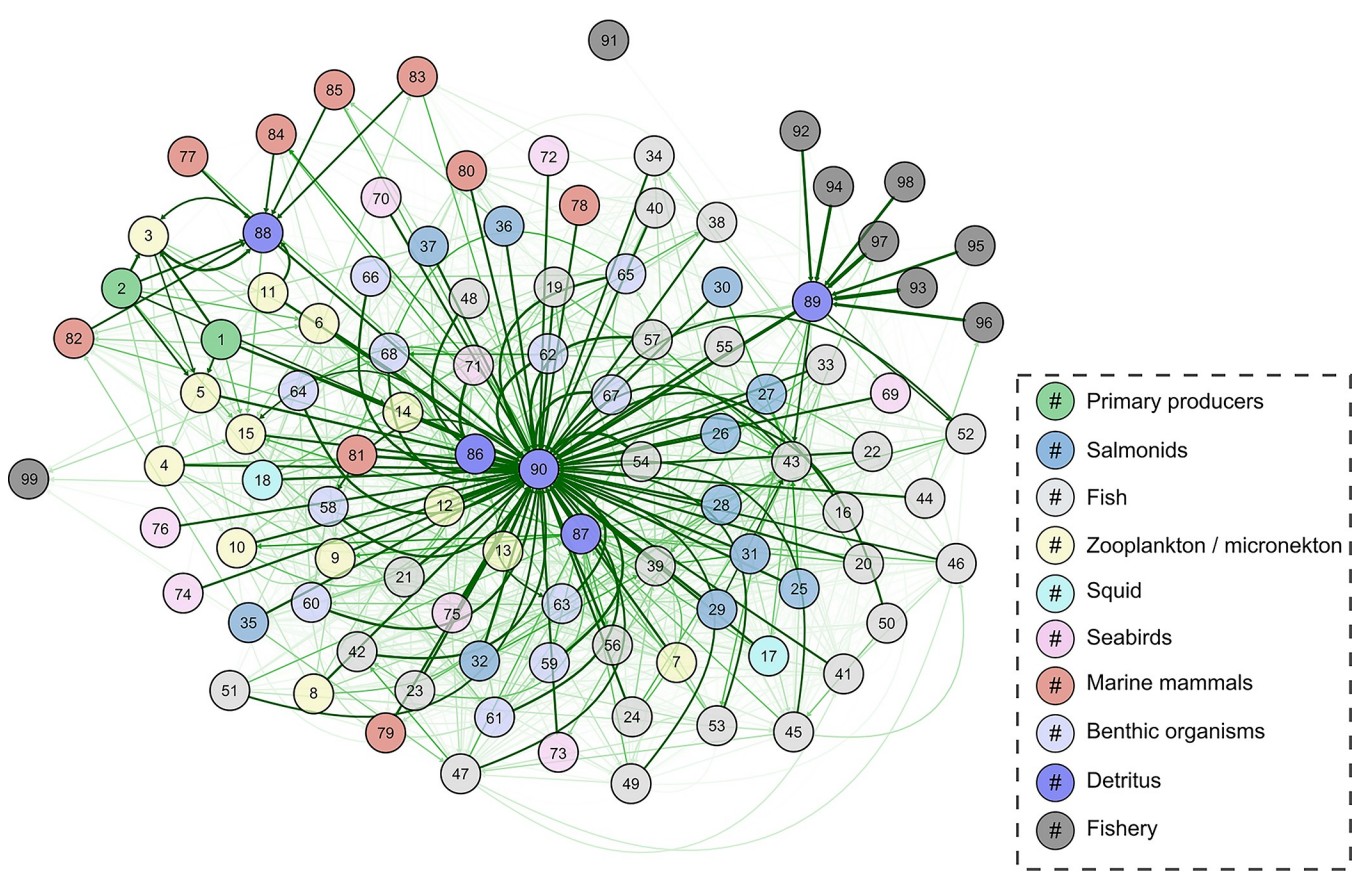

**Fig 6. EcoTran trophic network of ecosystem model.** The EcoTran trophic network is visualized here as a weighted, directed graph. Numbered nodes are functional groups (see Table 1 for numbers), while arrows indicate directed edges (energy flows from producer groups towards consumer groups). The shade of green indicates strength of interaction (higher diet preference and prey biomass results in darker green network edges) up to a value of 0.10, at which point network edges get thicker with higher values, as in Fig 5. This graph includes detritus groups (86–90), which dominate the network.

on what is available to them. Thus, it is important to note that the diet matrix does not fully represent changes since the onset of MHWs. We were able to update the diets of 26 functional groups, 19 of which included samples from the MHW years (2014 and later; see S1 Table). The addition of diets collected during the MHW period will reflect some shifts in diet trophic dynamics, even if averaged with older data. However, it will be more conservative in how far the diets have shifted since we are averaging them with their original diets. We think this conservative approach is beneficial because diet studies are stochastic (depending completely on when and where individuals are sampled) and have high degrees of uncertainty. In an ideal world, this updated model would include solely diet data from 2014 and onwards, yet decisions were ultimately driven by the fact that so few diet studies have been made available since the onset of MHWs.

**Marine mammal diets.** In addition to the trophic database (https://oceanview.pfeg.noaa.gov/cctd/), harbor seal diets were extracted from the literature [71, 72]. We followed Ruzicka et al. [32] in the use of Wright et al. (2007) [72] to parameterize harbor seal diets for their ecosystem model. That is, the proportion of salmon in harbor seal diets was scaled down from 36% to 20% to account for greater consumption of salmonids by river seals than by their coast-wide counterparts, which are included in the model. To use river seal diets for the entire coast would greatly overestimate the total abundance of salmon consumed by harbor seals. As

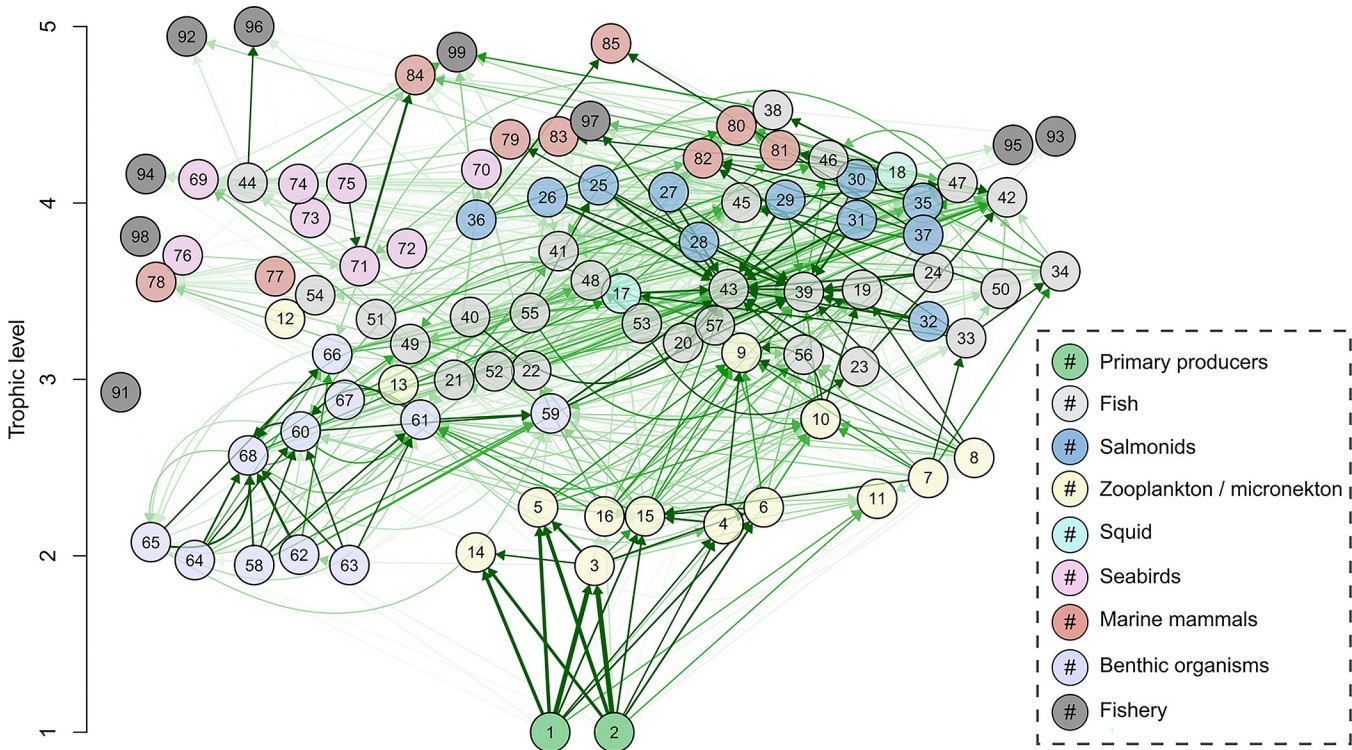

**Fig 7. EcoTran trophic network of non-detritus groups by trophic level.** The EcoTran trophic network is visualized here as a weighted, directed graph with detritus groups (86–90) removed (see Fig 6). Numbered nodes are functional groups (see Table 1 for numbers); arrows indicate directed edges (energy flows from producer groups towards consumer groups). The shade of green indicates strength of interaction (higher diet preference and prey biomass results in darker green network edges) up to a value of 0.025, at which point network edges get thicker with higher values.

discussed above, resulting harbor seal diets were a weighted (by sample size) average of these three sources, with 94% of harbor seal diet data coming from the trophic database, which means Wright et al. (2007) does not strongly influence the overall model but remains important in order to represent a broader range of functional groups in harbor seal diets. California and Steller's sea lion diet data were taken from the literature [71, 75, 76] (see "MarineMammal-Diets.R" in supplemental data and code), as were diet data from Northern elephant seals [77], and Southern resident killer whales [21].

## Fisheries landings

Many of the larger commercial fisheries were updated (data from 2014–2021) directly from the Pacific Coast Fisheries Information Network (PacFIN; http://pacfin.psmfc.org/; accessed 16 March 2022) for the following functional groups (see Table 2 for scientific names): the small cephalopod aggregate, smelt aggregate, shad, sardine, herring, anchovy, coho salmon, Chinook salmon, other salmon aggregate, jack mackerel, Pacific chub mackerel, hake, sablefish, Hexagrammidae (lingcod, greenlings), and Gadidae (cod, haddock, walleye pollock). Smaller commercial fisheries not mentioned above and recreational fisheries take were borrowed from a previous NCC ecosystem model [32, 33], which also obtained the data from PacFIN as well as the Pacific States Marine Recreational Fisheries Information Network (RecFIN; http://www.recfin.org/).

Fisheries landings in the PacFIN database were provided in large spatial domains that did not perfectly correspond to our model subregions (Fig 1). Thus, we redistributed the fisheries

landings north-south to match our model subregions by scaling to the proportion of overlap in latitude between PacFIN and EcoTran (see "PACFIN/" in supplemental data and code). Then, to allocate landings across the shelf (i.e., bathymetric breaks; Fig 1), we considered that fisheries landings were roughly proportional to where the species occurred in the ocean (reasonable assumption since our model domains do not represent costly or intractable harvesting areas). We used species distribution models for commercial forage fish (i.e., sardine, herring, anchovy, jack mackerel, Pacific chub mackerel, and market squid; Barbara Muhling, *personal communication*), fisheries bycatch information for adult salmon [78], the acoustic-trawl survey for hake, species count data from the groundfish survey, and assumptions for other groups (e.g., bivalves were taken in the nearshore region; see "PACFIN" in supplemental data and code). Fisheries landings data by gear (also from PacFIN) were then used to proportionally allocate model subregion landings to individual fishery fleets, which are tracked separately in EcoTran (Table 2, S4 Table). Discard rates were borrowed from a previous NCC food web (S5 Table) [32, 41, 79].

### Tuning detritus recycling

Rates of detritus recycling were tuned following Ruzicka et al. [41] so that the average total primary production rate across the whole shelf and the ratio of new production to total production [f-ratio = $NO_3$ uptake/($NO_3$+$NH_4$ uptake)] of the model system were comparable to independently obtained estimates. The f-ratio was tuned to approximate a cross-shelf range of 0.3–0.8 [80]. Total primary production was evaluated against the satellite-derived VGPM [50].

Pelagic detritus and benthic detritus are recycled in the model via bacterial remineralization and by direct consumption by metazoan consumers. The two major pathways of production loss from the EcoTran model are through advective transport of nutrients, plankton, and detritus offshore and through the sequestration of benthic detritus (Fig 2). Three recycling parameters, rates of pelagic and benthic detritus remineralization and rates of benthic detritus sequestration, were systematically varied in 0.1 unit increments from 0 to 1 for all possible parameter combinations in 20-year simulation runs. Tuning simulations were visualized as f-ratio and total primary productivity surface plots of two parameters at a time (see "Tuning_Detritus_Parameters/" in supplement). We also assessed the number of functional groups whose production rates fell below 1% of their original values at any point in the simulation–i.e., are going extinct in the model [30].

We chose the detritus remineralization and sequestration values that generated f-ratios and total primary productivity values closest to realistic values while preventing functional groups going extinct. With benthic detritus sequestration rates above 0.2 (>20% of benthic detritus production becomes "lost" to the system at every time step), production rates of many groups were falling to unreasonably low values as too much detritus production was being eliminated from the system to support detritivory. We set benthic detritus sequestration at 0.1. Once the benthic detritus sequestration rate was fixed to 0.1, changes to pelagic and benthic detritus remineralization did not substantially change the f-ratio, primary productivity, or extinction rates (see "Tuning_Detritus_Parameters/" in supplemental data and code). Pelagic and benthic detritus remineralization were each set to 0.1 (see results).

Further model details can be found in previous EcoTran articles [27, 32–34, 79]. To be more open, reliable, transparent, and reproducible [81], we have provided data and code to reproduce and use this ecosystem model at https://doi.org/10.5281/zenodo.7079777.

### Model validation

We used the mass-balancing steps described above [25, 31] and model validation diagnostics (described in Link and Heymans et al. [40, 82]) to visualize and assess the plausibility of the

food web parameterization of the pre- and post-balance aggregated model. Food web evaluation criteria guidance from Link [40] states that i) biomass density values of all functional groups should span 5–7 orders of magnitude, ii) there should be a 5–10% decrease in biomass density (on the log scale) for every unit increase in trophic level, iii) biomass-specific production values (P/B) should never exceed biomass-specific consumption (C/B) values, and iv) ecotrophic efficiency (EE) for each group should be below 1 [40].

Within the coupled physical-biological model system, we assessed the persistence of functional groups over 150-year time dynamic simulations within the aggregated regional model. Within such short timeframes, functional groups should not go extinct when driven by a constant upwelling time series [30]. We drove a 150-year simulation with an average upwelling (CUTI) time series, which was averaged across 1988–2021 for each day of the year. This single year average timeseries was repeated in each 150 years of the simulation. This average upwelling time series ensures that interannual variation is not adding unnecessary noise to our assessment of model stability and equilibrium over time (S1 Fig).

We assessed whether or not the ecosystem model reached equilibrium states in such dynamic simulations by assessing extinction rates and population dynamic trends in the final 20 years of the 150-year, constant-upwelling simulation, as suggested by Kaplan and Marshall (2016) [30]. That is, biomass values for most groups should not change significantly in the last 20 years of the simulation. We visualized the functional group trends within the simulation to assess how many groups were changing by more than 5% in the last 20 years.

In order to assess whether the ecosystem model can track the actual direction of change that we observe in the ocean over short (i.e., daily, monthly, and yearly) timescales, we drove nutrient availability in the model with a real (not averaged), 33-year upwelling time series (1988–2021 CUTI; [42]). We visualized ecosystem model-generated time series outputs of primary production, large jellyfish, market squid, Dungeness crab, Pacific sardine, Northern anchovy, Pacific jack mackerel, Pacific (chub) mackerel, common murre, sooty shearwater, baleen whale, and Southern-resident killer whales and compared these outputs with independently-derived time series estimates of abundance or biomass. For this comparison, we used time series of a vertically generalized production model (2002–2021) for primary production estimates aggregated across our entire spatial domain [50, 51]; the JSOES (1998–2019; [15]) for large jellyfish (sea nettles); coastwide commercial fisheries landings (2000–2020) for the small cephalopod aggregate (market squid; PacFIN: http://pacfin.psmfc.org/); a pre-season abundance model for legal-size male Dungeness crabs (1988–2016; N. CA, OR, and WA; [83]); stock assessments for the northern sub-population of Pacific sardine (2004–2019; [84]), the central stock of Northern anchovy (1995–2018; [56, 85]), Pacific jack mackerel (2006–2018; [55, 56, 86]), and Pacific chub mackerel (2008–2020; [87]), the JSOES (2003–2019; [15]) for common murre and sooty shearwaters; a humpback whale mark-recapture study (baleen whales; 1995–2018; [88]); and whole-population counts of Southern-resident killer whales (1988–2021; https://www.whaleresearch.com/orca-population).

## Results

Our constructed EcoTran model was consistent with ecological energetics, as suggested by the "PREBAL" criteria of Link [40]. Biomass densities span 6 orders of magnitude (within the 5–7 suggested range; Fig 4) and the slope on the log scale is about an 8.5% change each trophic level [between the 5–10% suggested range [40]; see Fig 4B]. Biomass-specific production values (P/B) never exceed biomass-specific consumption (C/B) values, and ecotrophic efficiency (EE) values are all below 1 (Table 1). In general, P/B and C/B should decrease as trophic level (TL) increases, except homeotherms should be lower (for P/B) and higher (for C/B) than expected

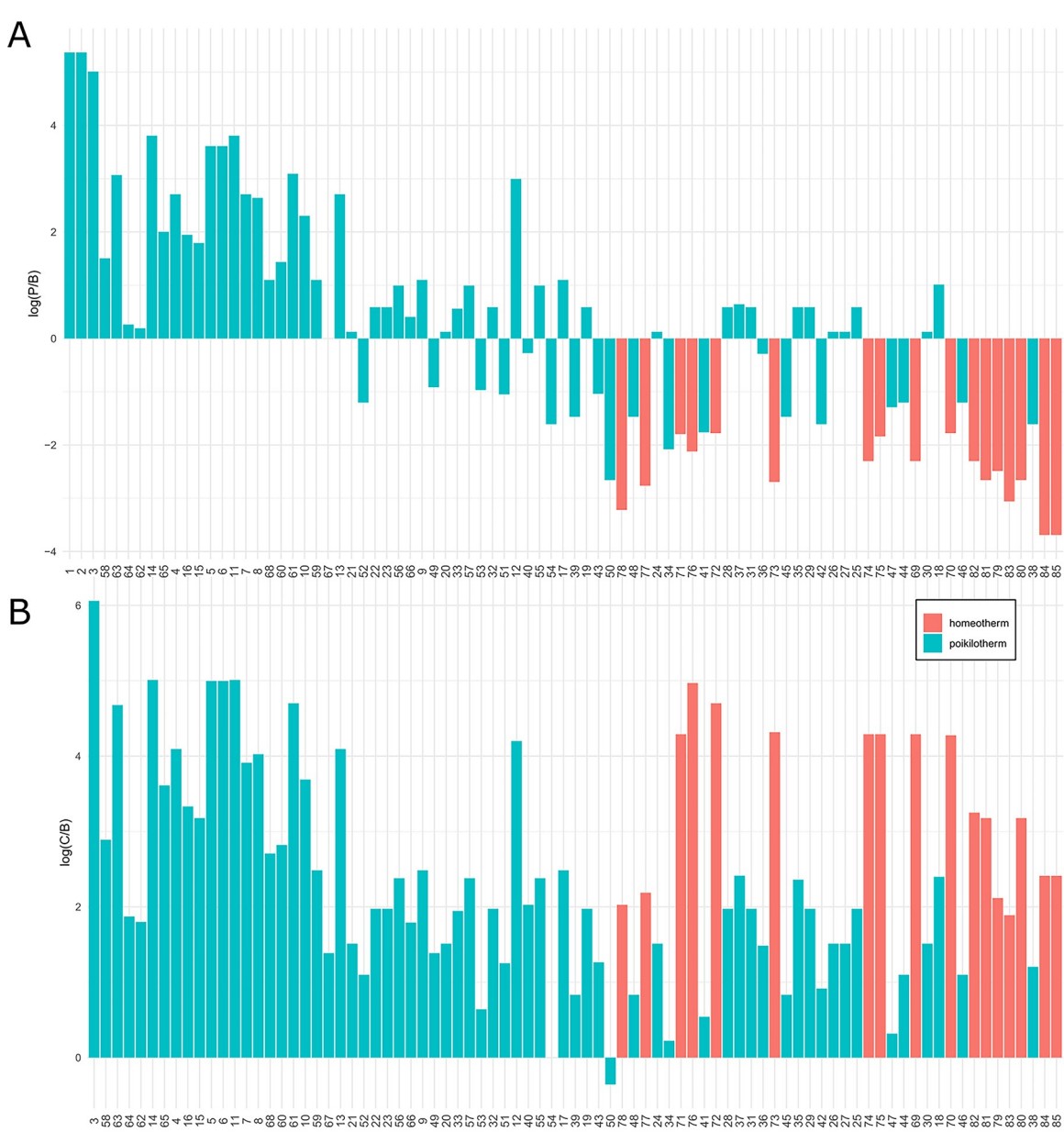

**Fig 8. Validation; biomass-specific production and consumption values by trophic level.** (A) The y-axis is the log-scaled production to biomass ratio (biomass-specific production) and the x-axis numbers indicate the different functional groups (see Table 1) sorted by increasing trophic level (far left is low trophic levels, e.g., phytoplankton, and far right is high trophic levels, e.g., marine mammals). (B) The y-axis is the log-scaled consumption to biomass ratio (biomass-specific consumption) and the x-axis is the same as in panel A. The y axes are on the log scale, so negative values indicate P/B and C/B values (for A and B, respectively) less than 1 on the original scale.

for other groups [40]. The model presented here appears to be consistent with these criteria (Fig 8), as a few deviations (or exceptions) are not a cause for concern [40]. Biomass estimates of primary productivity are on the same order as biomass estimates for detritus (Table 1; Fig 4B), which is also a sign that model parameterization is not unreasonable [40].

Some groups are consumed at relatively low levels (low EE values in Table 1 correspond roughly to the proportion consumed). Some notable groups are small copepods, with less than half of their substantial biomass (~26 mt/km$^2$) consumed (EE = 0.419); jack mackerel (~21 mt/

km$^2$; only consumed at ~10% of potential, EE = 0.1); pyrosomes, which are nearly as dense (~16 mt/km$^2$) but are hardly consumed at all (EE = 0.001); and three groups of jellies, none of which are as dense (1.7, 0.05, and 0.1 mt/km$^2$) but are similarly unconsumed (EE = 0.02, 0.02, and 0.08 respectively; Table 1).

Over a 150-year simulation, driven by constant time series and an average upwelling time series, zero functional groups go extinct from the model (Fig 9). Of 83 living consumer groups, none changed by more than 5% in the final 20 years, suggesting that all groups appear to have reached a mostly stable equilibrium. Longer-lived functional groups are expected to have very slow population dynamics, and so it is expected that some groups take longer than others to reach an equilibrium (Fig 9).

The f-ratio [NO$_3$ uptake/(NO$_3$+NH$_4$ uptake)] inner shelf is 0.68, mid shelf is 0.52, and outer shelf is 0.43, which are reasonably comparable to the cross-shelf range and pattern (0.3–0.8) for the NCC [80] and near values (0.75) previously used in NCC ecosystem models [41].

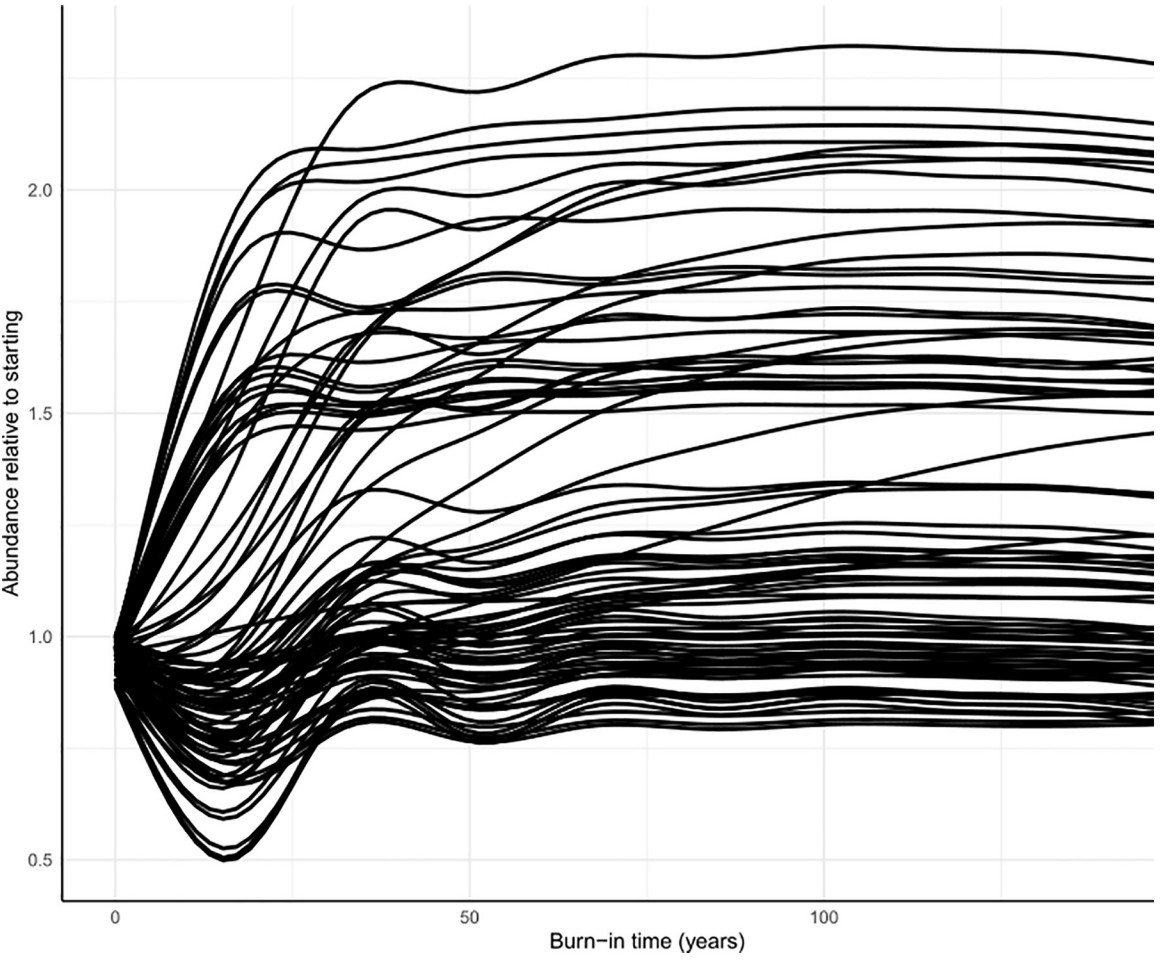

**Fig 9. Assessing stability: 150-year model simulation driven by average upwelling time series.** Each living consumer functional group is pictured here as a smoothed individual line. The proportion of abundance relative to starting values are plotted on the y-axis and the daily simulation timestep is plotted on the x-axis (axis in yearly units). The simulation here is driven by the average (day of year average for 1988–2021, see Methods) Coastal Upwelling Transport Index (CUTI) to reduce interannual variation for assessment of equilibrium and model stability. No functional groups are changing by more than 5% in the final 20 years of simulation (see S6 Table). That is, all groups have reached equilibrium (i.e., are within this 5% threshold). No functional groups go extinct over the 150-year simulation.

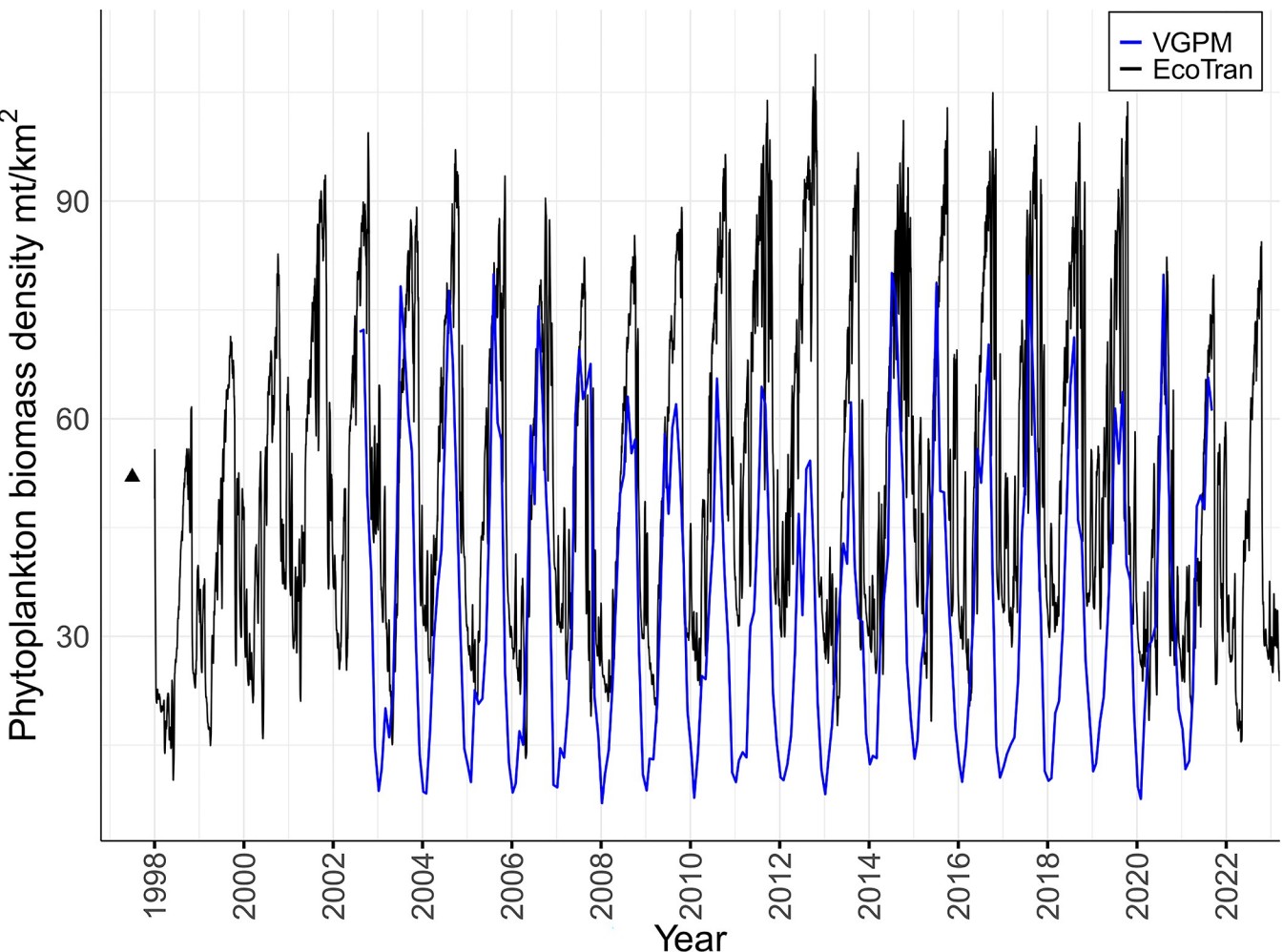

**Fig 10. Primary production time series as both ecosystem model output and vertically generalized production model.** Satellite imagery-derived estimates of daily primary productivity via a vertically generalized production model (VGPM, blue lines) [50, 51] are plotted against ecosystem model-derived estimates of primary production driven by an upwelling time series (black lines are aggregated large and small phytoplankton functional groups). The average values of the final eight years of the VGPM (2014–2021) are used to inform the starting conditions (values) for the ecosystem model. After this point, the ecosystem model is driven entirely by nutrient inputs to the system as determined by the coastal upwelling transport index (CUTI) [42], and any resemblance to the VGPM time series is an indication that the ecosystem model is capturing the appropriate dynamics in primary productivity.

Total primary productivity averaged across the ecosystem model is 3.25 mmol N m$^{-3}$ d$^{-1}$, which is reasonably close to the 1.99 mmol N m$^{-3}$ d$^{-1}$ vertically generalized production model (VGPM) estimate from satellite data [50] (see supplement, "Fratio_PP_check.m"). Our model output matches the seasonal timing observed in a phytoplankton primary productivity time series estimated from the VGPM (2002–2021; Fig 10) [50, 51]. The time dynamic simulations also match model output quite well for large jellyfish and Pacific sardine and reasonably well for Dungeness crab and Northern anchovy (Figs 11 and 12).

## Discussion

Here we have presented an end-to-end ecosystem model of the Northern California Current (NCC) marine ecosystem. Our model was built using many long-term West Coast ocean surveys [48], databases, and literature and updates previous ecosystem models of the region [32, 33, 45, 89]. The ecosystem model is in thermodynamic balance, meaning that no functional

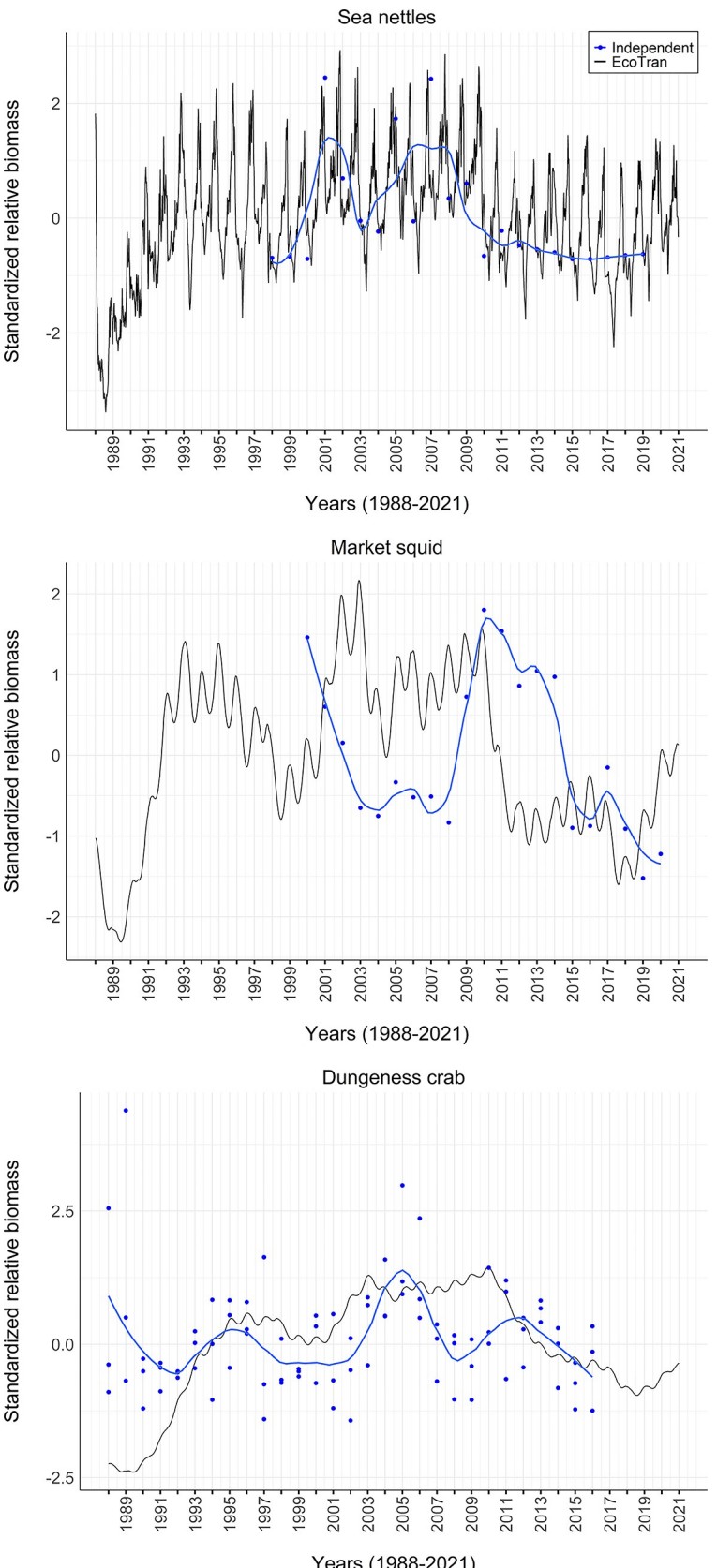

**Fig 11. Invertebrate functional group time series as both ecosystem model output and independent estimates.**
Independently derived estimates (blue points; blue lines = locally estimated scatterplot smoothing lines) of relative biomass via a Juvenile Salmon and Ocean Ecosystem Survey (JSOES; large jellies), fisheries landings (small cephalopod aggregate = market squid), and a pre-season abundance model (Dungeness crabs [83]) are plotted against ecosystem model-derived estimates of matching functional groups (black lines). The ecosystem model is driven entirely by nutrient inputs to the system as determined by the coastal upwelling transport index (CUTI) [42] and trophic relationships, and any resemblance of the two time series is an indication that the ecosystem model is matching independently-observed dynamics.

groups produce more than they consume, and model validation suggests that our model parameterization is energetically feasible [40]. This model serves as a useful accounting tool for understanding the fates of energy flow through the highly connected food web at broad spatial and temporal scales and allows for nearly unlimited simulation scenarios that we would be unlikely to be able to perform experimentally in the marine ecosystem.

In time-dynamic simulations, no functional groups go extinct and functional group biomasses are stable after 150-year simulations (Fig 9), suggesting that the model is stable and

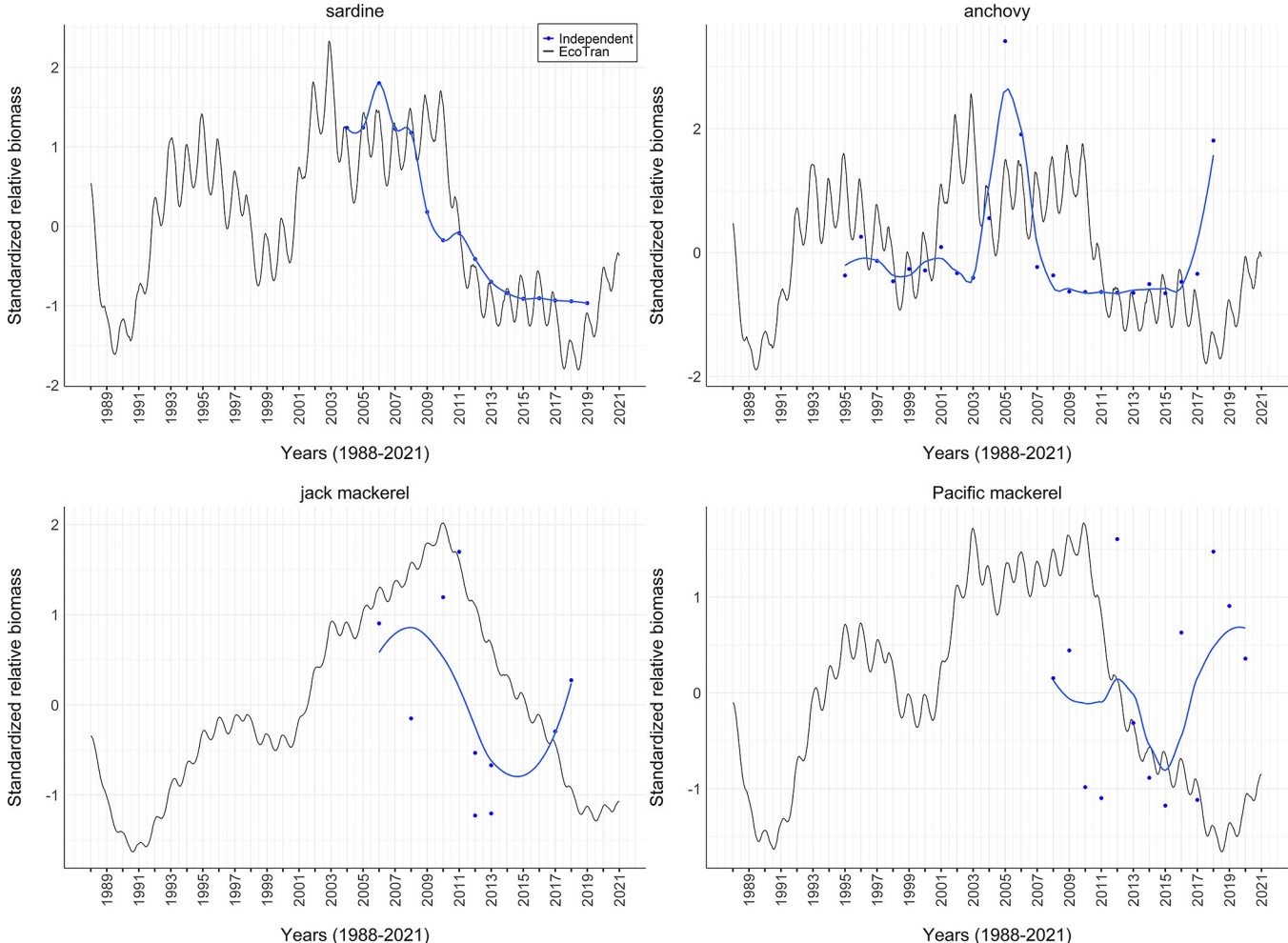

**Fig 12. Coastal pelagic fish functional group time series as both ecosystem model output and independent estimates.** Independently derived estimates (blue points; blue lines = locally estimated scatterplot smoothing lines) of relative biomass of sardine, anchovy, jack mackerel, and Pacific chub mackerel via stock assessments [56, 84, 86, 87] are plotted against ecosystem model-derived estimates of matching functional groups (black lines).

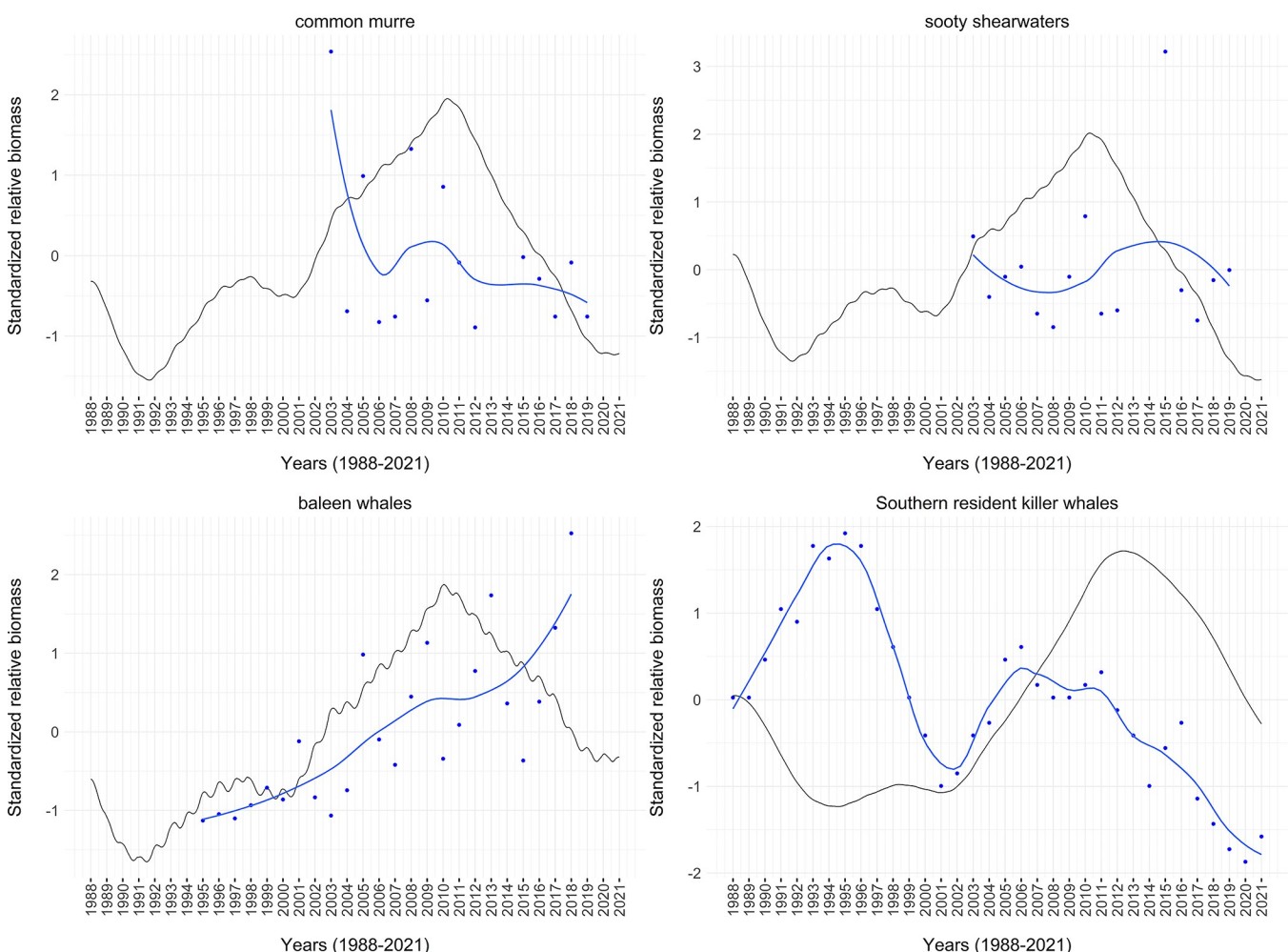

**Fig 13. Seabird and mammal functional group time series as both ecosystem model output and independent estimates.** Independently derived estimates (blue lines and points) of relative biomass via a Juvenile Salmon and Ocean Ecosystem Survey (JSOES; common murre and sooty shearwaters), a humpback whale mark-recapture study (baleen whales [88]), and counts of the well-monitored Southern resident killer whale population [https://www.whaleresearch.com/orca-population] are plotted against ecosystem model-derived estimates of matching functional groups (black lines).

reasonably parameterized [30]. Our upwelling-driven ecosystem model-derived estimates of primary production tracks the satellite imagery-derived production model estimates of primary productivity (Fig 10) [50, 51]. Primary productivity in the ecosystem model is determined by nutrient inputs to the system which are driven by the coastal upwelling transport index (CUTI; S1 Fig) [42], and the remarkably close cyclical resemblance to the VGPM time series is an indication that the ecosystem model is capturing the appropriate primary productivity dynamics well-enough to reproduce such dynamics (Fig 10). This last model validation check is promising, as primary production is what determines the bottom-up energy flux to higher trophic levels and drives the production of the entire food web (Fig 7).

Independent timeseries of large jellies and Pacific sardine were also tracked quite well by the ecosystem model (Figs 11 and 12). Higher trophic levels (i.e., seabirds and mammals; Fig 13) were not matched well, temporally, by the ecosystem model. This is unsurprising given that the higher trophic levels are further removed from primary production (and thus the upwelling driver). Additionally, and perhaps more importantly, complex behaviors that are often employed by higher trophic level species, such as migration and exposure to conditions

outside the NCC domain boundaries, are not captured by this ecosystem model. In these cases, static snapshots and analyses averaged over larger timescales are likely more appropriate and can be quite useful [32, 34, 35, 39, 41, 79]. It is important to note that some of the independent time series are based on a short period of time or are limited in space, which themselves might not always be accurate representations of reality; it is not always clear whether the independent timeseries or the ecosystem model outputs better reflect reality. However, any alignment of the ecosystem model output and independent time series suggest that at least some temporal dynamics have been captured, since the probability of both randomly aligning is quite low. Continued support for ongoing and new ocean ecosystem survey efforts (and their validation) will aid in accurately modeling such complex systems mathematically.

Our model is also congruent with model guidelines that are based on an understanding of trophic ecology [40]. Some seabirds (groups 71–76) fall below the log scale biomass density–trophic level line (Fig 4B), which may indicate that biomass densities for some of these higher trophic levels are actually under-estimated (inadequacy of the onboard observations for assessing abundance), which may be possible if some seabirds actively avoid survey vessels. Perhaps a more likely explanation is that each of these groups are more taxonomically resolved compared to other groups in the model (Tables 1 and 2). When these groups are combined for visualization (purple point in Fig 4B), they fall much closer to the trend line. Thus, it is worth noting that the choice of functional group aggregation can impact the interpretation of some of these validation diagnostics, but it does not mean that this will influence the model or that they should necessarily be combined. Combining all seabirds into one functional group should lead to energetically similar food webs, while it would severely muddy our ability to understand any individual group's impact on the ecosystem, if that ability were of scientific interest.

Some groups in our model appear to be bottom-up energetic pathway 'dead-ends' in that they are not consumed nearly as much as they could be. Small copepods for example appear to be able to handle double the predation that they currently do (see calculated EE in Table 1). It seems unlikely that this functional group is so abundant that it swamps predators, since euphausiids (*E. pacifica* and *T. spinifera*) are both more available and more consumed than the small copepod group. It is possible, however, that diets of copepod predators are not described well enough to parameterize the real consumption of this group. The size distinction between small and large copepods here is captured in the surveys that sample them, but this level of detail is often lost in diet analyses, especially due to breakdown during digestion (which may lead to an underestimation of consumed invertebrates in general). Here, we assume predators with copepods in their diets consumed these two groups based on the relative abundance of each. If small copepods are actually consumed more than our diet data indicate, we may be underestimating their contribution to the next trophic level.

Other energetic dead-ends such as the gelatinous zooplankton (i.e., jellies and pyrosomes) are not surprising given the low energy content of such prey [90]. However, jellies and other soft-bodied organisms are often so easily digested that they are often underrepresented in laboratory-based visual diet analyses [91]. Alternative methods, such as DNA analysis, stable isotopes, and animal-borne cameras, suggest that gelatinous zooplankton are consumed far more than we used to think [90, 92], which means we likely underestimate their role in transferring energy to higher trophic levels here. Pyrosomes have exploded in the ecosystem since the onset of the MHWs, potentially supplanting jellies [39, 47, 93–95]. The nutritional and energetic content of pyrosomes compared to more typically consumed crustaceans and fishes, as well as their likelihood to end up as prey in the NCC, remains unclear. This could have major implications for energy flux throughout the ecosystem [39], especially if jellies are indeed more important to higher trophic level predators than is presently assumed [92]. Indeed, analysis of diets of multiple pelagic fishes in 2015 and 2016 showed dramatically increased reliance on

gelatinous zooplankton compared to crustacean prey in normal or cool years [96], reflecting a similar shift seen in trawl survey catches [95].

Jack mackerel are under-consumed in the system. This species is currently more abundant in this system than they have been in the past 10 years [56], but it is not typically a species targeted by US fisheries, and fisheries catch reported by PacFIN is relatively low. The diets of some potential pelagic predators, such as sharks, tunas, and toothed whales overlap more strongly with the pelagic distribution of jack mackerel than other predators in the model. Yet, diet data from these predators are more limited in space and time, which means we may be underestimating the contribution of jack mackerel to predators in recent years. Additionally, it is possible that marine mammal diet studies that report "unidentified bony fish" due to the difficulty of identifying hard parts might be leading us to underestimate the total predation on jack mackerel. It is also possible that jack mackerel biomass is somewhat overestimated in the ecosystem, but this does not seem like an adequate explanation to fully account for the lack of consumption of the group. While this requires further research, this may be a relatively under-consumed group that commercial fisheries have yet to exploit.

Our hope is that the model presented here can aid in making management decisions surrounding the sustainable use of our oceans and the fate of imperiled species. We have, for example, expanded the functional groups within previous versions of the model to include endangered species act (ESA)-listed populations such as southern resident killer whales and their preferred prey, Chinook salmon [21] (Fig 14). Pacific salmon (*Oncorhynchus* spp.) are a group of general interest within the NCC due to their economic [97, 98], ecological [99–102], and cultural [103–106] benefits and their large scale declines across much of their range [107, 108]. Some populations of Chinook salmon (*O. tshawytscha*) are listed as threatened or endangered [109] while others continue to be commercially and recreationally fished [110, 111]. The most vulnerable life stage of the Chinook salmon is thought to be when salmon smolts enter the ocean and begin growing and transitioning into adults [112–116]. However, it is notoriously difficult to assess what occurs in the ocean because sampling is spatially and temporally incomplete and the food web is large, complex, and understudied.

Here we expand subyearling and yearling Chinook salmon from the previous NCC Eco-Tran model into six functional groups (see Table 2, Fig 14), which will allow future work to understand the role that the ocean stage plays in juvenile Chinook salmon persistence. These groups were chosen specifically to inform recovery of protected species, hydrosystem operations, and hatchery management. Most of the spring-run yearlings in the NCC are listed as threatened under the U.S. Endangered Species Act, and the factors affecting their marine survival are a primary concern of recovery managers. Snake River fall-run Chinook salmon are also threatened, but these fish responded to impoundment of the Snake River by developing a mixed life history, in which they might either migrate as subyearlings or yearlings. The difference in the marine survival between these two groups has important implications for recovery. The distinction between early and late ocean entry subyearlings is most relevant for hatchery management. Hatcheries can choose whether to release yearlings or subyearlings, and when to release them. The expected marine survival of each group is an important contributor to their decision. We further separate smolts that are migrating north from southerly locations, which are not affected by the Columbia River hydrosystem and its management (i.e., the 'other' juvenile Chinook salmon groups).

While our model presented here contains 90 functional groups, there are notable groups that are not well-parameterized and/or are missing from the model entirely. Many of the benthic organisms have not recently been sampled in any surveys that we are aware of. Yet, benthic groups are likely important to the ecosystem as they comprise a large portion of the total biomass (Table 1; Fig 4) and recycle nutrients by consuming benthic detritus (Figs 5 and 6)

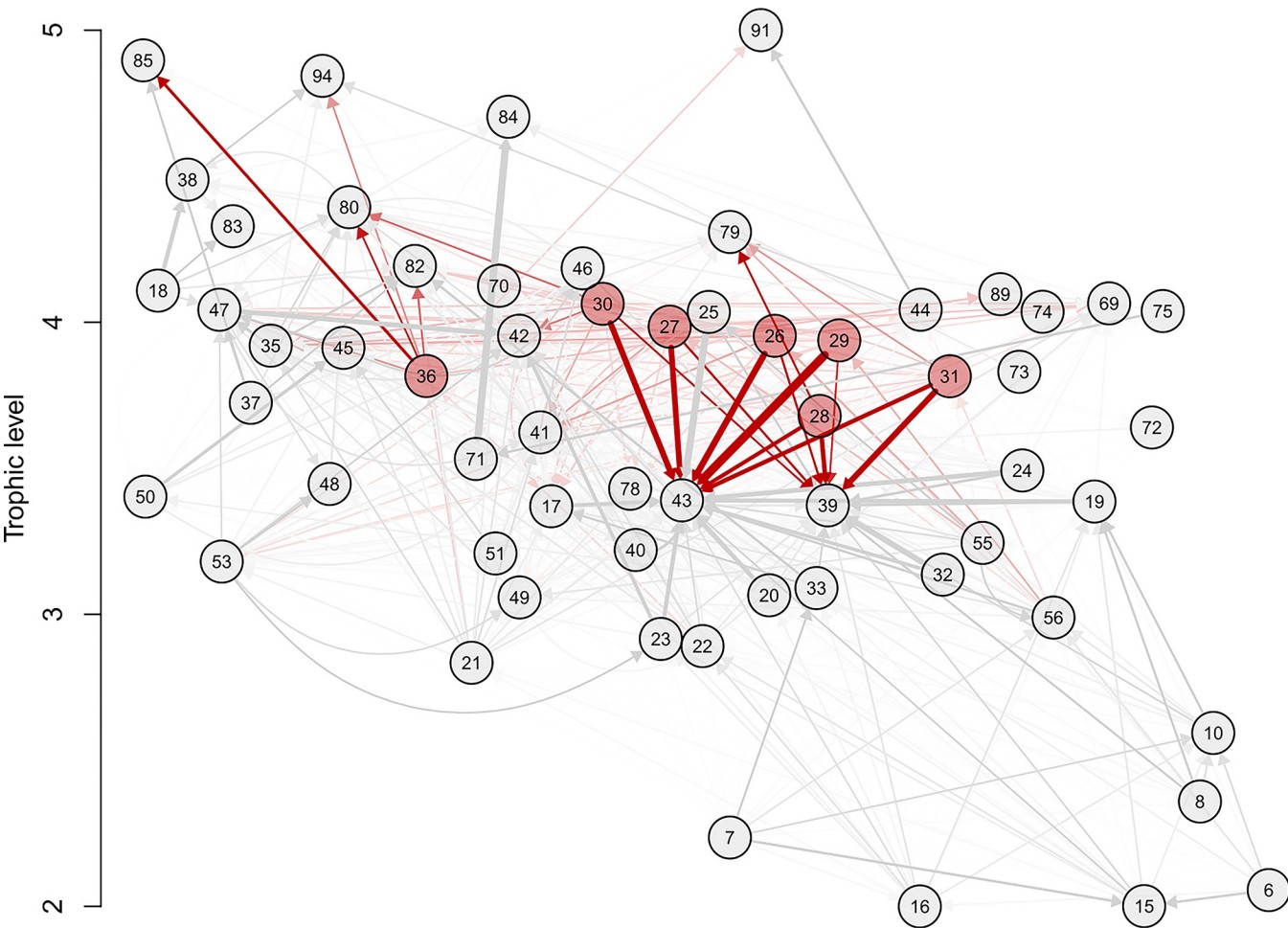

**Fig 14. Simplified Chinook salmon centric EcoTran trophic network.** The EcoTran trophic network is visualized here as a weighted, directed graph with detritus groups (86–90) and those without direct energy flow to or from Chinook salmon groups removed. That is, this is a simplified version of Fig 7, which allows for a focused perspective on Chinook salmon. Numbered nodes are functional groups (see Table 1 for numbers); arrows indicate directed edges (energy flows from producer groups towards consumer groups). The color intensity and line thickness indicates strength of interaction. Higher diet preference and prey biomass results in darker network edges up to a value of 0.025, at which point network edges get thicker with higher values (direct salmon connections are in red, all other connections in grey).

[32, 89]. Marine mammal biomass estimates are based solely on coastwide or stock-specific marine mammal population assessments, some of which have not been recently updated due to the stability of such populations or the absence of substantial threats to those species [66]. While these biomass estimates appear well within reason (Fig 4), more recent and spatially-resolved marine mammal data would reduce our uncertainty in these values. Additionally, many of the marine mammal and seabird groups, other than pinnipeds, lack thorough (i.e., in time, space, and sampling effort) diet analyses. An increased sampling effort would be valuable given the unknown importance of some understudied groups as predators (e.g., harbor porpoises on endangered salmon) [117, 118].

Some groups have been omitted from the model. One example, kelp, is known to be a crucial component of this ecosystem [119], as kelp forests provide habitat and food for various species of fish, including commercially important species and those listed under the Endangered Species Act, such as populations of Pacific salmon [120–123]. Others such as abalone

and urchins are lumped into very general functional groups, where a lack of available empirical data has forced a reliance on allowing our ecosystem model to estimate total biomass. The California Current is currently experiencing urchin barrens where kelp forests once existed [124–126]. Recent MHWs in combination with booming populations of urchins has taken such a toll on red abalone that Northern California and Oregon recreational fisheries (N. CA fishery valued at $44M year[-1]) have been completely closed [125, 127]. As increased attention is placed on the importance of ecosystem-based fisheries management, more data on these understudied functional groups will likely become available. The ecosystem model presented here would be a useful framework with which to assess various future scenarios and management strategies pertaining to such important functional groups.

Our model comprises a highly interconnected, albeit simplified, view of the NCC ecosystem and provides a flexible framework for understanding complex food web dynamics, management actions, and future states such as various climate change futures. This model represents significant shifts to the NCC ecosystem since the onset of MHWs nearly every year since 2014 [15, 39, 95]. The EcoTran ecosystem modeling framework can be used to estimate the pressure that various consumer groups exert on lower trophic levels and the rest of the ecosystem, identify important food web nodes and how energy is transferred between them, compare ecosystem states during periods of low and high predator or competitor biomass, and to conduct simulation analyses to estimate the impact of events such as northward expansions of fish, jellyfish blooms, fishing pressure, climate change, and other events upon the ecosystem [32–34, 39, 79, 128]. It is important to note, however, that this view is based on an incomplete snapshot in time and space based on limited available data [129]. Ecosystem modeling efforts would benefit from additional surveys and, importantly, more readily available data [81] (see [63] for an exemplary example) as is being done more effectively in the southern extent of the California Current [https://calcofi.org/data/]. As we move beyond single-species models towards holistic ecosystem-based fisheries management, we must openly and collaboratively integrate our disparate datasets and collective knowledge to solve the intricate problems we currently face in a changing world.

## Supporting information

**S1 Fig. Coastal Upwelling Transport Index (CUTI).** The Coastal Upwelling Transport Index (CUTI) is plotted on the y-axis against the day of year (x-axis). Each year (1988–2021) is plotted as an individual line in the blue gradient. The full timeseries (1988–2021) is used to drive the model for comparisons to the vertically generalized production model (VGPM) in Fig 10. The red line in the middle is the average CUTI time series (averaged by day of year across all years), which is used to drive the validation plot in Fig 9.
(PDF)

**S1 Table. Data sources and years included.** The table contains information about the sources of both the biomass and diet data for each functional group (rows of table) in the model. See https://doi.org/10.5281/zenodo.7079777 for a csv version of this table.
(CSV)

**S2 Table. Fates of detritus for each functional group.** The table contains information about the fates of detritus (eggs, pelagic detritus, fishery offal, benthic detritus, or export from the system) for each functional group (rows of table) in the model. See https://doi.org/10.5281/zenodo.7079777 for a csv version of this table and see Table 1, S3–S5 Tables for other ecosystem model parameters.
(CSV)

**S3 Table. Additional EcoTran parameters.** EcoTran parameterization of the model. BA = biomass accumulation and EM = emigration. Detritus fates are listed for feces, senescence and excretion to 2D surface and sub-surface boxes (see Fig 2). Retention scaler indicates the ability of advection to move various functional groups. Advection values of 0 means that groups are physically driven by cross-shelf advection (upwelling and downwelling), while values of 1 means that groups can resist the advection forces. See https://doi.org/10.5281/zenodo.7079777 for a csv version of this table and see Table 1, S2–S5 Tables for other ecosystem model parameters.
(CSV)

**S4 Table. Fishery landings for each functional group.** The table contains information about the yearly landings (mt/km$^2$) of each fishery in the model (columns in table; see Table 2 for descriptions) for each living functional group (rows of table) in the model. See https://doi.org/10.5281/zenodo.7079777 for a csv version of this table and see Table 1, S2–S5 Tables for other ecosystem model parameters.
(CSV)

**S5 Table. Fishery discards for each functional group.** The table contains information about the yearly discards (mt/km$^2$) of each fishery in the model (columns in table; see Table 2 for descriptions) for each living functional group (rows of table) in the model. See https://doi.org/10.5281/zenodo.7079777 for a csv version of this table and see Table 1, S2–S5 Tables for other ecosystem model parameters.
(CSV)

**S6 Table. Stability in 150-year simulations.** The table contains percent change in the last 20 years of a 150-year simulation with an average CUTI upwelling timeseries (see Fig 9) for each functional group (rows of table) in the model. See https://doi.org/10.5281/zenodo.7079777 for a csv version of this table.
(CSV)

**S1 Appendix.**
(DOCX)

## Acknowledgments

We thank Isaac Kaplan for many conversations about survey data, ecosystem modeling, and model validation; Isaac Kaplan and Brian Wells for reviewing an earlier version of this manuscript; Douglas Draper for information on the proportion of pyrosomes in the diets of several bottomfish; Anne Thompson for data on pyrosome diets; Sheanna Steingass and Casey Clark for sharing their knowledge of pinnipeds with us; Barbara Muhling for sharing her species distribution models for coastal pelagic species and market squid with us; Kevin Stierhoff for sharing the coastal pelagic species survey data with us; and Chantel Wetzel and her collaborators for creating a user-friendly R package to download West Coast Groundfish survey data from.

## Author Contributions

**Conceptualization:** Dylan G. E. Gomes, James J. Ruzicka, Lisa G. Crozier, David D. Huff.

**Data curation:** Dylan G. E. Gomes, James J. Ruzicka, Elizabeth M. Phillips, Pierre-Yves Hernvann, Cheryl A. Morgan, Richard D. Brodeur, Jen E. Zamon, Elizabeth A. Daly, Joseph J. Bizzarro, Jennifer L. Fisher, Toby D. Auth.

**Formal analysis:** Dylan G. E. Gomes, James J. Ruzicka.

**Funding acquisition:** Dylan G. E. Gomes, James J. Ruzicka, Lisa G. Crozier, David D. Huff.

**Investigation:** Dylan G. E. Gomes.

**Methodology:** Dylan G. E. Gomes, James J. Ruzicka.

**Project administration:** Dylan G. E. Gomes, James J. Ruzicka, Lisa G. Crozier.

**Resources:** James J. Ruzicka, Lisa G. Crozier, David D. Huff.

**Software:** Dylan G. E. Gomes, James J. Ruzicka.

**Supervision:** Dylan G. E. Gomes, James J. Ruzicka, Lisa G. Crozier.

**Validation:** Dylan G. E. Gomes, James J. Ruzicka, Pierre-Yves Hernvann.

**Visualization:** Dylan G. E. Gomes.

**Writing – original draft:** Dylan G. E. Gomes.

**Writing – review & editing:** Dylan G. E. Gomes, James J. Ruzicka, Lisa G. Crozier, David D. Huff, Elizabeth M. Phillips, Pierre-Yves Hernvann, Cheryl A. Morgan, Richard D. Brodeur, Jen E. Zamon, Elizabeth A. Daly, Joseph J. Bizzarro, Jennifer L. Fisher.

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
