## [Decision Letter · Decision Letter 0]

23 Mar 2023

PONE-D-22-35465

An updated end-to-end ecosystem model of the Northern California Current reflecting ecosystem changes due to recent marine heat waves

PLOS ONE

Dear Dr. Gomes,

Thank you for submitting your manuscript to PLOS ONE. After careful consideration, we have decided that your manuscript does not meet our criteria for publication and must therefore be rejected, In fact, one of the reviewers advice the manuscript rejection based on solid arguments that deserve my support and agreement.

I am sorry that we cannot be more positive on this occasion, but hope that you appreciate the reasons for this decision.

Kind regards,

João Miguel Dias, Ph.D.

Academic Editor

PLOS ONE

Reviewers' comments:

Reviewer's Responses to Questions

**Comments to the Author**

1. Is the manuscript technically sound, and do the data support the conclusions?

Reviewer #1: Yes

Reviewer #2: Yes

2. Has the statistical analysis been performed appropriately and rigorously? 

Reviewer #1: N/A

Reviewer #2: Yes

3. Have the authors made all data underlying the findings in their manuscript fully available?

Reviewer #1: Yes

Reviewer #2: Yes

4. Is the manuscript presented in an intelligible fashion and written in standard English?

Reviewer #1: Yes

Reviewer #2: Yes

5. Review Comments to the Author

Reviewer #1: An updated end-to-end ecosystem model of the Northern California Current reflecting ecosystem changes due to recent marine heat waves

For consideration in PLOS One

Authors: Gomes et al.

Manuscript ID: PONE-D-22-35465

This manuscript presents what is essentially an updated version of an existing end-to-end model (Ruzicka et al.) of the Northern California Current ecosystem, with several technical modifications/improvements. Apparently, the impetus for the update (at this time) is the need to account for anomalous oceanographic conditions that occurred throughout the past decade, to facilitate better understanding of how those conditions may impact the ecosystem, including several vulnerable/endangered species (e.g., Pacific salmon).

It seems the intent of this manuscript was not to report the responses of the ecosystem and these vulnerable species to the changing ocean conditions, but instead to validate the ecosystem model by presenting several general, pseudo-quantitative metrics of model performance (relative to expectations), in addition to diagnostic criterion associated with large end-to-end models, as well as the data inputs. From this standpoint, the manuscript has succeeded and I have not found any deficiencies or methods that appear to be incorrect. Additionally, the documentation of the methods and input data was very clear, and everything was easily accessible, per the references in the text.

I will say that the predicted increase in southern resident killer whale abundance was a bit counterintuitive, given that it was one of only two groups to show a significant increase, and that the population has been is such a precarious state the past two decades. I suppose this could be the topic for further research using this updated model? All told, I only have several very minor corrections for the authors to address, which are listed below.

Major comments/issues

(None)

Minor comments/issues

Line 479: References here are 64 (which is bird-related) and 65; shouldn’t this be 65 and 66 instead?

Line 565: Again, references here appear to be off, by one. I’m assuming you’re referring to 69 and 70 here, not 68 and 69?

Lines 641-646: This sentence reads awkwardly in its current form. I recommend breaking it up, or using alternate punctuation.

Fig. 9: The red against black color theme of this plot may be difficult for colorblind readers to interpret—I suggest replacing red with purple, green, or yellow to go against the black.

Reviewer #2: A Review of “An updated end-to-end ecosystem model of the Northern California Current reflecting ecosystem changes due to recent marine heat waves”

Here the authors present an updated and expanded ecosystem model that focusses on food web dynamics of the Northern California Current. The model is updated from a previously published version to include greater spatial domain, additional functional groups and greater resolution of some species groups (with a focus on endangered Chinook salmon), and to represent a more current time period (2014-2019) that was characterized by marine heatwave conditions. The model is driven by upwelling time series that drives nutrient input and thus primary production in the model. The model is presented as a tool for future analyses of food web dynamics, and specifically Chinook salmon trophic dynamics, under climate change scenarios.

This manuscripts describes the development of a tool (a model) and fits under the PLOS ONE submission category of “new methods, software, databases, or tools“. The Journal guidelines refer to acceptance criteria for this type of submission as (1) Utility, (2) Validation, and (3) Availability. I reviewed the manuscript accordingly.

I have reservations about recommending this manuscript for publication. I believe the model is sound and will support ecosystem analyses that will be submitted for future publication. However, the manuscript does not adequately fulfill the three categories to be published independently as a new tool. I would welcome its publication accompanied by analyses using the model (with most model details in the appendix), or conversely, additional development (and resubmission) with more of a focus of making the model a usable and available tool for other scientists and managers.

Utility

1. Use as a tool:

a. This model will be useful in examine trophic dynamics in the NCC, with a focus on Chinook salmon. The relatively large number of Chinook functional groups makes this model well-suited to salmon related ecosystem analyses.

b. The model was updated to represent the 2014-2021 marine heatwave period and potential changes in ecosystem state. I am not sure how the model will be used to show these potential changes, as there is no equivalent model in a previous time period for comparison. The previously published model would have a reduced footprint and functional groups, and no separation of Chinook salmon, potentially challenging the interpretation of differences in model outputs. Please note my point #2 under the Validation section for further discussion. I acknowledge the difficulty in assessing the ‘utility’ of this model without a detailed discussion of how the authors plan to use it.

c. In considering the ‘Utility’ of the model beyond the immediate author group, I believe it will be challenging. In my opinion, it is not well enough presented and supported to have utility beyond people with intimate knowledge of the model. While I applaud the large effort to document and make publicly available the data files and code to produce this model (Zenodo repository), I do not believe it is easily and readily available ‘Tool’ for use by other scientists and managers, limiting its utility. I will note this level of publicly available data files exceeds the documentation and availability of most models used to support published analyses of trophic dynamics. An advanced level of model utility and availability is not a requirement in those scenarios. But if the model is being published independently as a new ‘Tool’ then I believe more could be done to allow for its use by a broader audience. Some examples to support this comment:

i. The “README.txt” file states how there is no workflow to follow from start to finish, some data are not provided due to data ownership criteria, and some files may be missing.

ii. The main model file (“NCC2_08032022.csv”) is available and could be run but if other users wanted to modify the model for different applications, it would be challenging to work through the underlying files and code.

iii. The calculations span more than one programming software

iv. The “FlowChart.pdf” on Zenodo describes the workflow at a high level but not enough to easily recreate/modify the model.

2. Diet updates:

a. I suggest the authors are more specific in how the model was updated and can be interpreted. For example in Line 161: “We focused on data collected primarily during and after recent marine heatwaves (2014 – 2021) to more accurately reflect the current conditions within the NCC, as there is evidence that the ecosystem has entered a novel state [15,16]. We updated the representation of food web interactions with published and unpublished datasets and reports to reflect potential reshuffling of trophic links.” The model more accurately reflects the biomass and catch of the MHW period, but less so of “reshuffling of trophic links” since the diets do not specifically reflect that time period (more on this point below). It would be interesting to include diet data from this time period if available (which is challenging to have available data), especially for Chinook salmon since they are the focus of the model.

b. While diet data were updated, it is difficult to determine which groups were updated and what time period they represent from Table S1. Following the links to the online diet database (https://oceanview.pfeg.noaa.gov/cctd/ ; source of diet data for some groundfish), it lists the most current data as 2016 for Pacific herring and sardine, and 2011 for sablefish, as examples. It could be stated more clearly what time frame the diet data represent (newest surveys plus what older data were averaged together) to determine, for example, if the marine heatwave period is represented in the average or not.

c. Diet averaging methods: The authors describe their method of updating diet data as an average across all available diet data (including newer data and the original model data) to broaden the sources of information and reflect the diverse prey field (Line 549). I don’t disagree with this method but would suggest the authors highlight that ‘updating’ the diets does not reflect the MHW period specifically and would not reflect potential shifts in diet trophic dynamics.

Validation

1. The authors followed published and well-regarded prebalance validation methods described in Link (2010). These are legitimate steps in validating a model’s stability and basic functioning but do not speak to how well the model captures the dynamics of the specific system (Northern California Current) and trophic dynamics it is intended to represent. It is helpful to see the comparison of primary production estimates (Fig 10) as one step of this validation but the rest of the model’s trophic dynamics/biomass trajectories are not validated in that way. Comparisons to biomass time series is a common practice for models starting at an earlier time period, but given the more recent time period the model, I can see how that would be challenging. Heymans et al. (2016) describe “Best Practices for validation of Ecosim models”, which could also be useful.

2. I question why the expanded model was not initiated in the time period of the original published model (Ruzinka et al. 2010) so that it could be (1) compared to the original model with reduced functional groups and spatial footprint to determine how the expansion of the model changed the trophic dynamics and interpretation, (2) allow for a comparison of the MHW ecosystem structure with that portrayed in the earlier model (a stated purpose of this update) and (3) run forward in time and fit to time series of biomass and catch. In this way, it could be determined if the MHW conditions were captured by the model, it could be validated using the historic trends in biomass, and then run into the future based on that understanding.

3. A major focus of suggested analyses using this model will be on Chinook salmon. As such, it would be helpful to show some validation to determine how well salmon dynamics are represented in the model. At a minimum, highlighting the Chinook part of the network (e.g., Figures 5 and 6) could help qualitatively/visually validate and interpret the model.

Availability

1. The main model file, code, and a large portion of the model development files are in a public Zenodo repository.

2. The model is more available than many research models that support ecosystem analyses published in peer-review journals. However, I do not believe it meets the criteria of ‘Available’ if it is being published independently as a tool to be used by other researchers and managers. The supporting examples below are the same as listed under Untility_1c:

a. The README.txt file states how there is no workflow to follow from start to finish, some data are not provided due to data ownership criteria, and some files may be missing.

b. The main model file (NCC2_08032022.csv) is available and could be run but if other users wanted to modify the model for different applications, it would be challenging to work through the underlying files and code.

c. The calculations span more than one programming software

d. The “FlowChart.pdf” on Zenodo describes the workflow at a high level but not enough to easily recreate/modify the model.

Minor Comments

1. Trophic network figures 5&6 are difficult to interpret. A suggestion is to highlight (labels, colored lines, reduced diagrams) some key trophic pathways such as Chinook salmon, killer whales, or other species of commercial/ecological interest.

2. DietDataSources.csv on Zenodo – many cells reference the “XXXX diet database”; suggest replacing “XXXX” with the name of the database

3. Figure 8: Suggest removing the gray grid lines and rotate the axis labels on the x-axis

4. Figure 2 (Cross Shelf Physical Model) is extremely similar to Figure 2 published in Ruzinka et al., 2016. Check for copyright rules and appropriate acknowledgment.

6. PLOS authors have the option to publish the peer review history of their article (what does this mean?). If published, this will include your full peer review and any attached files.

Reviewer #1: No

Reviewer #2: No

- - - - -

---

## [Author Response · Author response to Decision Letter 0]

18 May 2023

To whom it may concern,

We appreciate that both reviewers have done a great job assessing the details of the 80-page manuscript and the 1,500 files of the data/code repository that was included. We believe that both reviewers see the value in this submission and offer suggestions for improvement, and that these concerns have been fully addressed here and have certainly improved the clarity of our manuscript and data/code repository. 

Thank you for your time and consideration and we look forward to your reply.

On behalf of all co-authors,

Dylan Gomes

(also see Word version of this reply as "Response to reviewers.docx")

Reviewers' comments:

Reviewer's Responses to Questions

Comments to the Author

1. Is the manuscript technically sound, and do the data support the conclusions?

Reviewer #1: Yes

Reviewer #2: Yes

2. Has the statistical analysis been performed appropriately and rigorously?

Reviewer #1: N/A

Reviewer #2: Yes

3. Have the authors made all data underlying the findings in their manuscript fully available?

Reviewer #1: Yes

Reviewer #2: Yes

4. Is the manuscript presented in an intelligible fashion and written in standard English?

Reviewer #1: Yes

Reviewer #2: Yes

5. Review Comments to the Author

Reviewer #1: An updated end-to-end ecosystem model of the Northern California Current reflecting ecosystem changes due to recent marine heat waves

For consideration in PLOS One

Authors: Gomes et al.

Manuscript ID: PONE-D-22-35465

This manuscript presents what is essentially an updated version of an existing end-to-end model (Ruzicka et al.) of the Northern California Current ecosystem, with several technical modifications/improvements. Apparently, the impetus for the update (at this time) is the need to account for anomalous oceanographic conditions that occurred throughout the past decade, to facilitate better understanding of how those conditions may impact the ecosystem, including several vulnerable/endangered species (e.g., Pacific salmon).

It seems the intent of this manuscript was not to report the responses of the ecosystem and these vulnerable species to the changing ocean conditions, but instead to validate the ecosystem model by presenting several general, pseudo-quantitative metrics of model performance (relative to expectations), in addition to diagnostic criterion associated with large end-to-end models, as well as the data inputs. From this standpoint, the manuscript has succeeded and I have not found any deficiencies or methods that appear to be incorrect. Additionally, the documentation of the methods and input data was very clear, and everything was easily accessible, per the references in the text.

Thank you for your comments and positive feedback. 

I will say that the predicted increase in southern resident killer whale abundance was a bit counterintuitive, given that it was one of only two groups to show a significant increase, and that the population has been is such a precarious state the past two decades. I suppose this could be the topic for further research using this updated model? All told, I only have several very minor corrections for the authors to address, which are listed below.

Thank you for this comment. We do not actually predict an increase in SRKW over time, instead we are just looking at stability in the production of those groups over time. The fact that most groups asymptote when run with this constant average upwelling timeseries only suggests that the model is stable, not that we would predict all of these groups to remain the same over a real timeseries with strong inter-annual variation. Marine mammals are so long-lived that this asymptote can take a very long time to reach, which is what is occurring for SRKWs in this figure. This is perhaps confusing and is clarified in the manuscript on lines 664-678 and the Fig 9 caption. Additionally, we have extended the simulation to 150 years (instead of 100), to show that all groups reach stability, which should aid in preventing other readers from being confused by the highlighted increase in SRKWs.

Major comments/issues

(None)

Minor comments/issues

Line 479: References here are 64 (which is bird-related) and 65; shouldn’t this be 65 and 66 instead?

Line 565: Again, references here appear to be off, by one. I’m assuming you’re referring to 69 and 70 here, not 68 and 69?

Thank you for catching these mistakes. Our citation manager for some reason had not updated the numbers in the references prior to submission, and made some the references off by a value of 1. We have now fixed this and appreciate your attention to detail.

Lines 641-646: This sentence reads awkwardly in its current form. I recommend breaking it up, or using alternate punctuation.

Thank you for this point. We agree this sentence was clunky and confusing, and have clarified this in the revised manuscript, now on lines 657-662.

Fig. 9: The red against black color theme of this plot may be difficult for colorblind readers to interpret—I suggest replacing red with purple, green, or yellow to go against the black.

Thank you for this point. I was only aware of the red-green distinction being difficult for colorblind readers. We have now changed this in our updated manuscript.

Reviewer #2: A Review of “An updated end-to-end ecosystem model of the Northern California Current reflecting ecosystem changes due to recent marine heat waves”

Here the authors present an updated and expanded ecosystem model that focusses on food web dynamics of the Northern California Current. The model is updated from a previously published version to include greater spatial domain, additional functional groups and greater resolution of some species groups (with a focus on endangered Chinook salmon), and to represent a more current time period (2014-2019) that was characterized by marine heatwave conditions. The model is driven by upwelling time series that drives nutrient input and thus primary production in the model. The model is presented as a tool for future analyses of food web dynamics, and specifically Chinook salmon trophic dynamics, under climate change scenarios.

This manuscripts describes the development of a tool (a model) and fits under the PLOS ONE submission category of “new methods, software, databases, or tools“. The Journal guidelines refer to acceptance criteria for this type of submission as (1) Utility, (2) Validation, and (3) Availability. I reviewed the manuscript accordingly.

I have reservations about recommending this manuscript for publication. I believe the model is sound and will support ecosystem analyses that will be submitted for future publication. However, the manuscript does not adequately fulfill the three categories to be published independently as a new tool. I would welcome its publication accompanied by analyses using the model (with most model details in the appendix), or conversely, additional development (and resubmission) with more of a focus of making the model a usable and available tool for other scientists and managers.

Thank you for your positive feedback. We have prepared a resubmission that focuses more on making the model a “usable and available tool”, by adding additional details and clarification to the data/code repository (your second suggested option). However, we also believe that criteria within your first suggested option have already been met; the manuscript already does include quite some analyses using the model (Figs 4-10 are model outputs, and additional new Figs 11-13 are as well), and we want to be careful not to make the 80-page manuscript even more cumbersome than it already is by including too many additional analyses.

Utility

1. Use as a tool:

a. This model will be useful in examine trophic dynamics in the NCC, with a focus on Chinook salmon. The relatively large number of Chinook functional groups makes this model well-suited to salmon related ecosystem analyses.

Thank you for your positive feedback.

b. The model was updated to represent the 2014-2021 marine heatwave period and potential changes in ecosystem state. I am not sure how the model will be used to show these potential changes, as there is no equivalent model in a previous time period for comparison. The previously published model would have a reduced footprint and functional groups, and no separation of Chinook salmon, potentially challenging the interpretation of differences in model outputs. Please note my point #2 under the Validation section for further discussion. I acknowledge the difficulty in assessing the ‘utility’ of this model without a detailed discussion of how the authors plan to use it.

Thank you for your comments. We do not need a second “equivalent” model to assess changes in the ecosystem state. Previous work has been published on the ecosystem that we can directly compare to this ecosystem model and its outputs (e.g., many of these values are expressed per unit area, or grouped by functional group aggregates that were combined for analysis). Additionally, most of the functional groups are identical across models. As far as functional group changes go, only minor modifications have been made to improve species resolution around Chinook salmon and marine mammals to make modelling scenarios more useful for endangered and commercially fished Chinook salmon. This doesn’t mean that the two models cannot be directly compared for all other 90+ functional groups (as well as an aggregated Chinook or killer whale group). We believe we have suggested possible uses of the model on lines 828-834 (now lines 866-872): “The EcoTran ecosystem modeling framework can be used to estimate the pressure that various consumer groups exert on lower trophic levels and the rest of the ecosystem, identify important food web nodes and how energy is transferred between them, compare ecosystem states during periods of low and high predator or competitor biomass, and to conduct simulation analyses to estimate the impact of events such as northward expansions of fish, jellyfish blooms, fishing pressure, climate change, and other events upon the ecosystem [32,32–34,77,115].”, but agree that to be more specific than this would be to clarify exactly how we plan to use it in the near future, which would have little utility in this manuscript.

c. In considering the ‘Utility’ of the model beyond the immediate author group, I believe it will be challenging. In my opinion, it is not well enough presented and supported to have utility beyond people with intimate knowledge of the model. While I applaud the large effort to document and make publicly available the data files and code to produce this model (Zenodo repository), I do not believe it is easily and readily available ‘Tool’ for use by other scientists and managers, limiting its utility. 

Thank you for your comments and positive feedback about the large effort to document and make available the files and code. We agree that as the repository was, it was quite confusing as to how to use the model. We have now separated the repository into “reproducibility”, which describes how we built the model (not necessary for future uses of the model), and “ECOTRAN_Code”, which describes the code and final model file (which completely allows the user to build their own model scenarios and simulations relatively easily). We have now also included a 22-page README about the model code that we accidentally omitted originally. Future users will only have to edit the first few lines of one MATLAB script in order to use the model for an infinite number of model scenarios and simulations – this has now been clarified in the data and code repository.

I will note this level of publicly available data files exceeds the documentation and availability of most models used to support published analyses of trophic dynamics. An advanced level of model utility and availability is not a requirement in those scenarios. But if the model is being published independently as a new ‘Tool’ then I believe more could be done to allow for its use by a broader audience. 

Thank you for your comments. This submission isn’t solely a new ‘Tool’, but also includes many model outputs, many of which are already usable to others working in the ecosystem (Figs 4-10, and recently added Figs 11-13, all Tables, supplementary data, etc.). We have now further clarified the use of the model and distinguished the many files (data & code) that belong to the reproducibility part of the submission (which are not required to use the tool). See comments above about improvements to the documentation, including the 22-page readme for the model code. While we have made clarifying changes to the new version of the repository, we also recognize that more can always be done to improve ease of use. The manuscript itself is already 80 pages and the data/code repository 1500 files. We have added additional clarifying details now to the supplemental data/code repository, which we agree will make it more useful. 

Some examples to support this comment:

i. The “README.txt” file states how there is no workflow to follow from start to finish, some data are not provided due to data ownership criteria, and some files may be missing.

This isn’t exactly true. It states “there isn't a completely linear workflow to get from start to finish”. This is an important distinction, as the workflow is there, but it is somewhat iterative. Importantly, this is the workflow for the reproducibility part of the data/code submission, not the use of the model, which has now been more thoroughly detailed in a revised data/code submission. We have also more properly highlighted the two distinct parts (reproducibility and model use) of the data/code repository, which should lead to less confusion. Some raw data that were used to build the model are only available here as a header with a few rows. This is due to data contributors being unwilling to openly share data. This, however, does not change the usability or availability of the model, as these data are only necessary to reproduce the model from “scratch”. Thus, the files that call these header data files are included in the spirit of transparency and open science, which does not detract from anyone’s ability to use the tool.

ii. The main model file (“NCC2_08032022.csv”) is available and could be run but if other users wanted to modify the model for different applications, it would be challenging to work through the underlying files and code.

We agree that this would be challenging as it took us nearly 2 years to update this model for different applications than were previously available. This tool is built to be specifically for the Northern California Current. Any adaptation to other ecosystems will require substantial work, which is not limited by the data and code we’ve provided, but rather future user’s own time. Many of the underlying files and code are simply for the sake of reproducibility, and are not needed to use the model. This has now been clarified in the repository (see responses above). As it stands, the main model file you reference, and the code to run scenarios are immediately available and relatively easy to use, as the user only needs to open one MATLAB file and edit the first few lines of annotated code to run simulations – this has also been clarified in the newest version of the data and code repository. We have improved the language around this within our initial readme file to make this more clear, and appreciate you pointing this out to us. 

iii. The calculations span more than one programming software

While this isn’t exactly merit for disqualification, again, this isn’t entirely true for the “tool” part of the model. The model was built using R and MATLAB code. So indeed, to reproduce our calculations, you would have to use both languages. However, the model is run purely on MATLAB code, so users can indeed use this tool with only one programming software.

iv. The “FlowChart.pdf” on Zenodo describes the workflow at a high level but not enough to easily recreate/modify the model.

We agree that the FlowChart.pdf doesn’t describe the workflow enough to recreate or modify the model, but this wasn’t the point of this document. The document was intended to do exactly as you say, show a “high level” description of what is going on – i.e., only to orient the user. The highly annotated code itself and individual readme files within each subdirectory are what provides the detail to recreate or modify the model. Additionally, as stated above, we left out an important 22-page document that provides the thorough details on the model code.

2. Diet updates:

a. I suggest the authors are more specific in how the model was updated and can be interpreted. For example in Line 161: “We focused on data collected primarily during and after recent marine heatwaves (2014 – 2021) to more accurately reflect the current conditions within the NCC, as there is evidence that the ecosystem has entered a novel state [15,16]. We updated the representation of food web interactions with published and unpublished datasets and reports to reflect potential reshuffling of trophic links.” The model more accurately reflects the biomass and catch of the MHW period, but less so of “reshuffling of trophic links” since the diets do not specifically reflect that time period (more on this point below). It would be interesting to include diet data from this time period if available (which is challenging to have available data), especially for Chinook salmon since they are the focus of the model.

We agree that we demonstrate the reshuffling of trophic links “less so” than biomass and catch, although we still do demonstrate the reshuffling of some trophic links, as is, because we have incorporated many newer diet studies, including from a long time series of juvenile Chinook salmon, which does change the strength and ties of the trophic connections (even with some of the original connections still included). We have been more careful in the wording, per your comments above and below. For example, we have specified that we only had updated diet information for 26 functional groups, 19 of which were during MHW years, “We updated the representation of food web interactions for 26 functional groups, including diet data from the MHW period for 19 groups, with published and unpublished datasets and reports to reflect potential reshuffling of some trophic links.” (Lines 166-169), and have added additional caveats and explanation in the Methods:

“Thus, it is important to note that the diet matrix does not fully represent changes since the onset of marine heatwaves (MHW). We were able to update the diets of 26 functional groups, 19 of which included samples from the MHW years (2014 and later; see Table S1). The addition of diets collected during the MHW period will reflect some shifts in diet trophic dynamics, even if averaged with older data. However, it will be more conservative in how far the diets have shifted since we are averaging them with their original diets. We think this conservative approach is beneficial because diet studies are stochastic (depending completely on when and where individuals are sampled) and have high degrees of uncertainty. In an ideal world, this updated model would include solely diet data from 2014 and onwards, yet decisions were ultimately driven by the fact that so few diet studies have been made available since the onset of marine heatwaves.” (lines 567-577).

We agree that these details are important to include.

b. While diet data were updated, it is difficult to determine which groups were updated and what time period they represent from Table S1. Following the links to the online diet database (https://oceanview.pfeg.noaa.gov/cctd/ ; source of diet data for some groundfish), it lists the most current data as 2016 for Pacific herring and sardine, and 2011 for sablefish, as examples. It could be stated more clearly what time frame the diet data represent (newest surveys plus what older data were averaged together) to determine, for example, if the marine heatwave period is represented in the average or not.

Thanks for this comment. We agree that this would be a useful contribution, and have now included this as an additional column in the supplementary Table S1 in the latest revision.

c. Diet averaging methods: The authors describe their method of updating diet data as an average across all available diet data (including newer data and the original model data) to broaden the sources of information and reflect the diverse prey field (Line 549). I don’t disagree with this method but would suggest the authors highlight that ‘updating’ the diets does not reflect the MHW period specifically and would not reflect potential shifts in diet trophic dynamics.

Thank you for this point. We agree that we need to be more careful in how we describe this, and have edited the manuscript to make this more clear. See response above to point 2a for more details. In an ideal world with an unlimited supply of sampling effort, we would indeed only use MHW period diet data to parameterize the MHW period, but do not believe that to be feasible given the sparseness of diet studies. 

Validation

1. The authors followed published and well-regarded prebalance validation methods described in Link (2010). These are legitimate steps in validating a model’s stability and basic functioning but do not speak to how well the model captures the dynamics of the specific system (Northern California Current) and trophic dynamics it is intended to represent. It is helpful to see the comparison of primary production estimates (Fig 10) as one step of this validation but the rest of the model’s trophic dynamics/biomass trajectories are not validated in that way. Comparisons to biomass time series is a common practice for models starting at an earlier time period, but given the more recent time period the model, I can see how that would be challenging. Heymans et al. (2016) describe “Best Practices for validation of Ecosim models”, which could also be useful.

Thank you for your comments and for the additional reference, which is now included in the updated manuscript. We agree that the prebalance validation methods are legitimate but don’t demonstrate how well the model captures the dynamics of the ecosystem. As no earlier papers using Ecotran do any timeseries validation (see Ruzicka et al. 2012, 2016a, 2016b, 2018; Chiaverano et al. 2018, referenced in the initial submission), we intended to go beyond the basic PREBAL steps by including the primary production timeseries, which, in turn, drives the rest of the consumption in the system. We have improved the manuscript by including comparisons (model output to biomass time series) of 7 other available taxa (new Figures 11 and 12), and we agree that this has greatly improved the manuscript.

2. I question why the expanded model was not initiated in the time period of the original published model (Ruzinka et al. 2010) so that it could be (1) compared to the original model with reduced functional groups and spatial footprint to determine how the expansion of the model changed the trophic dynamics and interpretation, (2) allow for a comparison of the MHW ecosystem structure with that portrayed in the earlier model (a stated purpose of this update) and (3) run forward in time and fit to time series of biomass and catch. In this way, it could be determined if the MHW conditions were captured by the model, it could be validated using the historic trends in biomass, and then run into the future based on that understanding.

Thank you for your comments. If we understand your question, you question why we didn’t build an additional model that expanded on the functional groups and spatial footprint of the original model (without updating the biomass and diets). Put quite simply, that would be months of additional work that wouldn’t give us a model that is updated for current conditions. While this comparison may be a useful modeling exercise, it doesn’t get us closer to having a management-ready model for our ecosystem as it currently is. 

To point #1, expanding a few functional groups and increasing the spatial resolution shouldn’t be dramatically changing the trophic dynamics and interpretation of the model. Model currencies are expressed per unit area, so adding additional resolution doesn’t change the overall average trends and patterns, instead we are able to learn more about specific groups and specific locales. 

To point #2, your suggested changes are not all necessary to make the comparison that you are suggesting. The energy budget and network properties that emerge are directly comparable across models without having to change the spatial extent of the models, which don’t differ dramatically. To compare across functional groups, collapsing to common denominators (aggregated juvenile Chinook groups, for example), are easy ways to make these comparisons without the additional steps of re-parameterizing and re-balancing an additional food web model. In fact, this is what we are currently doing for a direct comparison between models. However, this comparison is an additional effort that will be a standalone publication and would make this already lengthy manuscript unwieldy. 

It is unclear what you mean by point #3. Our model cannot be “fit to time series of biomass and catch”. We can assess whether the time series of biomass and catch matches our model output over the same time period, which is what we have done with primary production in Fig 10 (and with timeseries of other taxa in new Figs. 11-12), but we cannot actually fit the model to data, in the same way a statistical model is fit to data. 

Additionally, running the old version of the model forward in time would be unlikely able to capture the MHW conditions (which is outside the scope of what we are doing in this manuscript), since such dramatic events occurred (the occurrence of a novel species, for example). That doesn’t mean the earlier model wasn’t useful for the time period in which it was parameterized. The MHW caused such dramatic changes to the ecosystem that we felt it was necessary to rebuild the model, as is presented in this manuscript.

3. A major focus of suggested analyses using this model will be on Chinook salmon. As such, it would be helpful to show some validation to determine how well salmon dynamics are represented in the model. At a minimum, highlighting the Chinook part of the network (e.g., Figures 5 and 6a) could help qualitatively/visually validate and interpret the model.

Thank you for these suggestions. While we agree it would be valuable to have a time series validation of the model for juvenile salmon, we do not currently have a survey with repeat sampling and high temporal resolution to validate the model against. We have now indicated salmon separately within Figs. 5-7 and added a new Fig 13 to make these groups stand out more to the reader, per your suggestions. We agree that these changes have improved the manuscript.

Availability

1. The main model file, code, and a large portion of the model development files are in a public Zenodo repository.

2. The model is more available than many research models that support ecosystem analyses published in peer-review journals. However, I do not believe it meets the criteria of ‘Available’ if it is being published independently as a tool to be used by other researchers and managers. The supporting examples below are the same as listed under Untility_1c:

a. The README.txt file states how there is no workflow to follow from start to finish, some data are not provided due to data ownership criteria, and some files may be missing.

b. The main model file (NCC2_08032022.csv) is available and could be run but if other users wanted to modify the model for different applications, it would be challenging to work through the underlying files and code.

c. The calculations span more than one programming software

d. The “FlowChart.pdf” on Zenodo describes the workflow at a high level but not enough to easily recreate/modify the model.

Thank you for stating that the model is more available than many research models supporting ecosystem analysis – this was certainly our goal. Indeed, it is a difficult process involving 1500 files in our repository, so it makes sense that so many researchers have omitted important details. 

While we believe that this manuscript is not just a new tool (see responses above), we disagree with your assessment of whether or not the model is Available. Firstly, there is nothing in the definition of ‘Availability’ from the webpage that isn’t met by our submission:

If the manuscript’s primary purpose is the description of new software or a new software package, this software must be open source, deposited in an appropriate archive, and conform to the Open Source Definition. If the manuscript mainly describes a database, this database must be open-access and hosted somewhere publicly accessible, and any software used to generate a database should also be open source. If relevant, databases should be open for appropriate deposition of additional data. Dependency on commercial software such as Mathematica and MATLAB does not preclude a paper from consideration, although complete open source solutions are preferred. In these cases, authors should provide a direct link to the deposited software or the database hosting site from within the paper. If the primary focus of a manuscript is the presentation of a new tool, such as a newly developed or modified questionnaire or scale, it should be openly available under a license no more restrictive than CC BY.

Indeed all of the data and code files to use the model are available, by direct link (within the manuscript) to a long-term repository with a CC BY license. There appears to be some confusion about the distinction between the data/code used to reproduce the work vs the data/code used to actually use the tool – only the latter of which needs to theoretically meet these criteria (although, again, this submission is not only a new tool). We have now clarified the distinction between the reproducibility and model code parts of the data and code submission within our repository. All of the reproducibility data and code we’ve provided, at the same link, is above and beyond what is necessary to make the tool available. We have provided this because we feel strongly about open science and wish to be as transparent as possible, but this is not outlined as a necessity for publication. See individual responses in text above (under Untility_1c) for duplicated points a-d.

Minor Comments

1. Trophic network figures 5&6 are difficult to interpret. A suggestion is to highlight (labels, colored lines, reduced diagrams) some key trophic pathways such as Chinook salmon, killer whales, or other species of commercial/ecological interest.

Thank you for this point. We have now labelled salmon groups distinctly, per your suggestion, for all 3 original network diagrams (Figs 5-7). Additionally, we have added a new Figure 13 that highlights Chinook salmon relationships in a simplified food web diagram. We have made all figure code publicly available such that readers can modify the plots further or if they wish to only see connections to and from other functional groups. We will note that there is something to be gained and something lost in every different view of a network diagram. Reducing the diagram to connections to and from Chinook will highlight more Chinook relationships (Figure 13), but will lose the overall connectivity of the entire network that is more representative of the ecosystem model (Figs 5-7).

2. DietDataSources.csv on Zenodo – many cells reference the “XXXX diet database”; suggest replacing “XXXX” with the name of the database

Thank you for catching this. We have fixed this per your suggestion.

3. Figure 8: Suggest removing the gray grid lines and rotate the axis labels on the x-axis

Thank you for this suggestion. We attempted to make the suggestions that you have proposed, however, we decided to keep the figure as is for a couple of reasons. While removing gray grid lines might look more aesthetically-pleasing, we believe the values of the bars (on both axes) will be much more difficult to decipher without the grid lines to aid the eye (we’re choosing function over aesthetics here). Additionally, rotating the x-axis labels will make most columns twice as wide (1 character width to 2), meaning that the font will have to be decreased to fit the figure onto one page, making the labels illegible. 

4. Figure 2 (Cross Shelf Physical Model) is extremely similar to Figure 2 published in Ruzinka et al., 2016. Check for copyright rules and appropriate acknowledgment.

Thank you for this comment. You are right that the figures are similar, but our Figure 2 here was created from scratch in an illustration software. We have now added “adapted from Ruzicka et al 2016” to the figure caption to provide acknowledgement.

---

## [Decision Letter · Decision Letter 1]

26 Sep 2023

PONE-D-22-35465R1

An updated end-to-end ecosystem model of the Northern California Current reflecting ecosystem changes due to recent marine heat waves

PLOS ONE

Dear Dr. Gomes,

Thank you for submitting your manuscript to PLOS ONE. After careful consideration, we feel that it has merit but does not fully meet PLOS ONE’s publication criteria as it currently stands. Therefore, we invite you to submit a revised version of the manuscript that addresses the points raised during the review process.

We look forward to receiving your revised manuscript.

Kind regards,

Abdul Azeez Pokkathappada, Ph.D.

Academic Editor

PLOS ONE

Journal Requirements:

2. We note that Figures 1 and 3 in your submission contain map images which may be copyrighted. All PLOS content is published under the Creative Commons Attribution License (CC BY 4.0), which means that the manuscript, images, and Supporting Information files will be freely available online, and any third party is permitted to access, download, copy, distribute, and use these materials in any way, even commercially, with proper attribution. For these reasons, we cannot publish previously copyrighted maps or satellite images created using proprietary data, such as Google software (Google Maps, Street View, and Earth). For more information, see our copyright guidelines: http://journals.plos.org/plosone/s/licenses-and-copyright.

    1. You may seek permission from the original copyright holder of Figures 1 and 3 to publish the content specifically under the CC BY 4.0 license.  

Additional Editor Comments (if provided):

Reviewers' comments:

Reviewer's Responses to Questions

**Comments to the Author**

1. If the authors have adequately addressed your comments raised in a previous round of review and you feel that this manuscript is now acceptable for publication, you may indicate that here to bypass the “Comments to the Author” section, enter your conflict of interest statement in the “Confidential to Editor” section, and submit your "Accept" recommendation.

Reviewer #3: All comments have been addressed

Reviewer #4: (No Response)

Reviewer #5: (No Response)

2. Is the manuscript technically sound, and do the data support the conclusions?

Reviewer #3: Yes

Reviewer #4: Partly

Reviewer #5: Yes

3. Has the statistical analysis been performed appropriately and rigorously? 

Reviewer #3: Yes

Reviewer #4: N/A

Reviewer #5: I Don't Know

4. Have the authors made all data underlying the findings in their manuscript fully available?

Reviewer #3: Yes

Reviewer #4: Yes

Reviewer #5: Yes

5. Is the manuscript presented in an intelligible fashion and written in standard English?

Reviewer #3: Yes

Reviewer #4: Yes

Reviewer #5: Yes

6. Review Comments to the Author

Reviewer #3: This manuscript presents what is essentially an updated version of an existing end-to-end model using the EcoTran platform by Ruzicka et al. (2012) and Ruzicka et al. (2016) in the Northern California Current ecosystem, with several new datasets during and after recent marine heatwaves (2014-2021). The present result showed that the most recent data update is the need to take into account the anomalous oceanographic conditions that occurred during the last decade to facilitate a better understanding of how these conditions can affect the ecosystem, including some vulnerable/threatened species such as the Pacific Salmon in this research. Moreover, with sufficient data, this proves that this model is capable of being used in other places.

So far, the manuscript has been successful, and I have not found any significant flaws or methods. In addition, the documentation of methods and input data is apparent, and all are easily accessible, according to the references in the text. The research result is significant, especially for scientists and managers interested in the NCC ecosystem. However, the authors must carefully examine some minor issues before this manuscript is accepted for publication. Some details in the manuscripts need to recheck to make a better understanding for the reader. Besides, using consistent terms in the publication is essential so the reader is not confused.

Further, I suggest the authors discuss in the discussion part if this model can be applied in the other's location.

Major issue

None

Minor issue

Line 59: is there any different between marine food web and marine food-web? If there is no different, please remove one of them. NCC change to full name: Northern California Current.

Line 133: marine heatwaves (MHW)

Line 133-134: change to "…..increased magnitude and frequency of marine heatwaves (MHW) (i.e., the 2014-2016 and 2019-2020 MHW) [8–10],"

Line 159: Ruzicka et al. (2012, 2016) [32,33] change to Ruzicka et al. [32,33].. please check again in the whole manuscript for the same format problem (example Line 219, 229 and many more).

Line 164: marine heatwaves (MHW) change to MHW

Line 178: Remove Northern California Current and keep the short form NCC. The authors already mentioned the short form before. Also, check the whole manuscript for the same problem (for example, Line 329, 484, etc.).

Line 260-262: Can the authors include the P/Q = Production efficiency and AE = Assimilation Efficiency value in Table 1?

Line 286: The Ecopath ecotrophic efficiency (EE) change to The Ecopath EE

Line 292-296: "…… underestimated since many animals can avoid sampling gear"..The term many animals here, including birds or non-aquatic animals? If only aquatic animals, please add "aquatic animals" for better understanding.

Line 309: "wasn't" changed to "was not"

Line 372: the unit for biomass density is not mt/km3, as mentioned in Line 342. Please check the whole manuscript for the unit.

Line 375: Which Appendix do the authors mean? Do you refer to Appendix: Newport Hydrographic Line? I suggest the authors mention more details (such as "see Appendix: Newport Hydrographic Line") because this manuscript has a lot of documents and information. Sometimes make, the reader misunderstands.

Line 401: metric tons / km3 change to mt/km3

Line 402: Does the "supplemental code" refer to the file in Ruzicka et al., 2016, or the present study? I can't find the supplemental code at the https://doi.org/10.5281/zenodo.7079777. Please check again.

Line 417: Please check the unit = mt km-2

Line 419-420: the full scientific name for E. pacifica and T. spinifera due to first-time mention

Line 422: Change to E. pacifica and T. spinifera

Line 427: Do you mean "Zooplankton, jellies, and pyrosomes" subheading?

Line 432: Again. Please check the subheading name

Line 439 and 457: change to mt/km3

Line 460 and 469: the unit for areal biomass densities is not mt/km2? check Line 342

Line 484: Northern California Current change to NCC

Line 492 and 494: Ecotrophic Efficiency change to EE.

Line 568 and 577: marine heatwaves change to MHW

Line 628: vertically generalized production model (VGPM) change to VGPM

Line 656: Link, 2010 [39,80]) change to Link [39]. And reference 80 is not published by Link. Please check again the reference.

Line 658: Similar to Line 656. Check the format for reference.

Line 660: biomass-specific production values (P/B) change to P/B

Line 661: biomass-specific consumption (C/B) change to C/B

Line 662: ecotrophic efficiency (EE) change to EE

Line 657-662: I suggest a change to "Food web evaluation criteria guidance from Link [39] states that i; biomass density values of all functional groups should span 5 – 7 orders of magnitude, ii; there should be a 5 – 10% decrease in biomass density (on the log scale) for every unit increase in trophic level, iii; P/B should never exceed C/B values, and iv; EE for each group should be below 1 [39].

Line 687: vertically generalized production model (VGPM) change to VGPM

Line 700-702: "Biomass-specific production values (P/B) never exceed biomass-specific consumption (C/B) values, and ecotrophic efficiency (EE) values are all below 1 (Table 1)" change to "P/B never exceed C/B values, and EE values are all below 1 (Table 1)"

Line 730: "from a vertically generalized production model (VGPM; 2002 - 2021; Fig 10) [49,50]. "change to "from a VGPM 2002 – 2021 (Fig 10) [49,50]."

Line 784: Do the authors mean "see calculated EE in Table 1"

Line 785: euphausiids mean by krills or E. pacifica and T. spinifera or better mentions by the function group in Table 1.

Line 856: marine heatwaves change to MHW

Line 864: Northern California Current change to NCC

Line 881: The data availability statement should mention "supplement or supplement code" because authors used many "see supplement" or "see supplement code" in the text.

Suggestion:

Ecosystem model files, scripts and supplement code, including the balanced and unbalanced diet matrices, biomass estimates, various cleaning scripts, readme files, and all files mentioned in the text can be found at https://doi.org/10.5281/zenodo.7079777.

Table and Figure

Check the unit for Figure 4a and 4b.

Reviewer #4: The present manuscript describes the EcoTran (extension of ECOPATH) ecosystem model parameterized for the Northern Californian Current system. It is an enhanced version of the existing application with the enhancement achieved through defining more functional groups, considering more detailed spatial structure of the model domain, introducing more fisheries, and adding more recent diet data to capture changes in the marine ecosystem with the recent MHW period. The objective of the study is to propose an efficient tool for ecosystem-based fisheries management. For 90 functional groups/species defined in this application, the authors collected and cleaned data on species diet, landings, biomass densities, and fine-tuned specific input parameters to satisfy the mass-balancing equation. Overall, the manuscript addresses complex and timely question, and the extensive effort on data digging, the model parametrization and validation undoubtedly merits being published. However, the outcomes of the study as they are presented now are weak, and the message that the model can serve as a tool suitable for ecosystem analyses and fisheries management is unconvincing. I believe that the current manuscript requires major revision before it could be considered suitable for publication. Several major problems related to the model construction, the outputs and their validation are listed below.

The model

No immigration and emigration are considered in the model. It is not explicitly stated, nor discussed in the present manuscript, but according to Ruzicka et al (2012), the modelling “assumed a steady-state system with no biomass accumulation and no migration in or out of the system during the model period”. In Table S3 all emigration parameters are set to 0 and advection accounts only for cross-shelf movements (upwelling and downwelling). The horizontal advection seems to exist only to model nutrient flux. How such assumption can be justified for those highly migratory predators, which migrate seasonally to the NCC domain (e.g., albacore and juvenile bluefin tunas and gray whales), or for smaller pelagic species performing offshore-nearshore movements (jack mackerel), or for the species known to undertake the latitudinal migrations with inter-annual variability (Pacific sardine)?

The MHW period is presented as the impetus for updating the model, but it remains unclear whether any notable changes could be captured by the updated model, especially since it has not been compared to its previous version. With respect to the MHW impact, is it at all possible to trace its impact and if yes how, if the key variable, i.e., water temperature, is not accounted for in the model?

The model outputs under historical time series of CUTI seem to be highly correlated, just with expected delays between trophic levels. For example, the same pattern is seen in the time series of Market squid (Fig. 11), sardine and anchovy (Fig. 12), while the stock assessments (driven by CPUE and length or age frequency) show very different dynamics. Interestingly, well-documented (see e.g., MacCall et al., 2016) collapse of anchovy is shown by the stock assessment on Fig. 12, but not captured by EcoTran. I’m not an Ecopath expert and familiar with the modeling approach only through the literature, so I’m not sure if this is the problem of the flawed model assumptions, or a particularity of the modeling approach or the ecosystem driven by seasonal upwelling, but the modelled temporal dynamics seems to be simply driven by the dynamics of nitrate and ammonium at the base of the food web. Thanks to the data availability, I could reproduce these plots and trace the pattern down to the base of the food web. Thus, ammonium and shrimp biomass are linearly correlated with Pearson r=0.91 with 2-monthly lag, then shrimp biomass is correlated with anchovy with monthly lag (r=0.88), sardine biomass is correlated with Chinook yearlings with one-year lag (r=0.95), which is correlated with Chinook group with monthly lag (r=0.99), Chinook group is correlated with tunas with 9-monthly lag (r=0.98) etc. Besides, for some species of similar trophic level, the time series dynamics are nearly identical, e.g., for herring, anchovy, mesopelagic fish aggregate, sardine, and squid. Also, surprising correlations exist between higher trophic level species with very different life history traits, e.g., between gray whales and skates&rays group (r=0.9), tuna and seabirds (0.91), or tunas and hake (r=0.94). Such perfect alignment between temporal dynamics of functional groups seems highly unrealistic, indicating an oversimplification, e.g., considering NCC as a closed ecosystem (see my comment above).

Validation

Regarding the phytoplankton validation, the similar periodicity between model predicted phytoplankton biomass density and the VGPM primary production is regarded as the model skill (lines 752-754). However, in many coastal systems, chl-a blooms are driven by upwelling. So, it is very likely that the “remarkably close cyclical resemblance” is primarily the effect of model forcing, i.e., CUTI variability.

Since all validation plots are shown on standardized y-axis, it is unclear how close the absolute biomass densities are to independent estimates.

Finally, if describing the adequate temporal dynamics of intermediate and high trophic level species, is beyond the model capacity, I encourage authors to demonstrate those model skills, for which it can be a helpful and reliable tool for fisheries management. The validation can be done, for example, by running retrospective analyses, results of which can be verified with independent data.

Minor comments:

Figures 5-7 are still difficult to read and interpret. Even though a color coding is added to distinguish functional groups, reading the connections between species, and interpreting the diet interactions is impossible. So, either these figures should deliver some qualitative information like footprint and reach in Ruzicka et al. (2012, Figure 6), or it would be better to make group aggregations to make the graph with less vertices, or even provide the diet matrix as in Ruzicka et al. (2012, Table A.3).

Fig 10. Is there any reason to include all 15 regions into the “time series” of observed phytoplankton, which results in vertical lines every monthly date? Comparing the biomass densities averaged over the model domain seems to be more appropriate. Note also, two months of data are missing in the VGPM PP data file.

Reviewer #5: a. The model is a useful tool and made available to other researchers to use, with potential to improve ecosystem-based management practices. The extensive data sources and compilation improves the existing model and improves utility for potential users. The authors have addressed previous reviewer comments. I suggest this paper is published and only have minor comments.

b. Line 234: The latitude of the Newport Hydrographic Line is farther south than 46.7*N. Is this an incorrect latitude, or are the authors referencing a different transect?

c. Line 772 – 778: In the jelly/gelatinous zooplankton paragraph, the authors could elaborate on the ecosystem and management implications of the rise or persistence of gelatinous organism, as discussed in discussed in Brodeur et al., 2019, not just sampling bias issues.

d. Line 792-794: Could the authors be more specific in how the six functional groups might improve management, e.g. various stocks versus seasonality?

e. General comment: In the introduction, the authors note recent changes in the NCC, such as marine heatwaves, ocean acidification, etc. Can you speak in the discussion on if, or how, this model might reflect a regime shift in the ecosystem? How different were the results of this model iteration to previous model iterations (e.g., Ruzicka et al., 2016) that did not include 2015-2019 data?

7. PLOS authors have the option to publish the peer review history of their article (what does this mean?). If published, this will include your full peer review and any attached files.

Reviewer #3: No

Reviewer #4: No

Reviewer #5: No

---

## [Author Response · Author response to Decision Letter 1]

27 Oct 2023

See attached word document for formatted version of this response to reviewers.

Editor's comments:

We confirm that we have conformed to PLOS ONE’s style requirements and uploaded figures according to the PLOS ONE figure naming convention. 

2. We note that Figures 1 and 3 in your submission contain map images which may be copyrighted. All PLOS content is published under the Creative Commons Attribution License (CC BY 4.0), which means that the manuscript, images, and Supporting Information files will be freely available online, and any third party is permitted to access, download, copy, distribute, and use these materials in any way, even commercially, with proper attribution. For these reasons, we cannot publish previously copyrighted maps or satellite images created using proprietary data, such as Google software (Google Maps, Street View, and Earth). For more information, see our copyright guidelines: http://journals.plos.org/plosone/s/licenses-and-copyright.

Figures 1 and 3 were custom made in R with the ggplot2 package. This package accesses boundaries in the public domain from the US Department of Commerce, Census Bureau, Cartographic Boundary Files (https://www.census.gov/programs-surveys/geography/technical-documentation/naming-convention/cartographic-boundary-file.html). Thus, no content within Figures 1 and 3 are copyrighted and the Creative Commons Attribution License (CC BY 4.0) is appropriate for this material. We have now added statements to Figures 1 and 3 noting where the state boundaries came from.

Reviewer's comments:

Reviewer #3: This manuscript presents what is essentially an updated version of an existing end-to-end model using the EcoTran platform by Ruzicka et al. (2012) and Ruzicka et al. (2016) in the Northern California Current ecosystem, with several new datasets during and after recent marine heatwaves (2014-2021). The present result showed that the most recent data update is the need to take into account the anomalous oceanographic conditions that occurred during the last decade to facilitate a better understanding of how these conditions can affect the ecosystem, including some vulnerable/threatened species such as the Pacific Salmon in this research. Moreover, with sufficient data, this proves that this model is capable of being used in other places.

So far, the manuscript has been successful, and I have not found any significant flaws or methods. In addition, the documentation of methods and input data is apparent, and all are easily accessible, according to the references in the text. The research result is significant, especially for scientists and managers interested in the NCC ecosystem. However, the authors must carefully examine some minor issues before this manuscript is accepted for publication. Some details in the manuscripts need to recheck to make a better understanding for the reader. Besides, using consistent terms in the publication is essential so the reader is not confused.

Further, I suggest the authors discuss in the discussion part if this model can be applied in the other's location.

Thank you for your positive feedback and for your thoughtful review. 

Major issue

None

Minor issue

Line 59: is there any different between marine food web and marine food-web? If there is no different, please remove one of them. NCC change to full name: Northern California Current.

Thank you for this comment. Indeed there is a difference between marine food web and marine food-web. If you search each in Google Scholar, different references are returned. There appears to be some differences in the way that search engines use the additional hyphen. To make our manuscript accessible to either searches, we would prefer to keep both keyword phrases. Keywords are designed to complement the title and abstract – that is, all three are indexed in many search engines. So only words that do not appear in the title or abstract should be added to keywords (keywords are often mis-used in this way). For this reason, “Northern California Current” would not be a useful keyword, since it already appears in the abstract. However, if individuals search “NCC” as a shorthand for “Northern California Current”, they will not find our article without including “NCC” as a keyword (as “NCC” is not included in the abstract). 

Line 133: marine heatwaves (MHW)

Thank you for this comment. The acronym “MHW” has been added in this location.

Line 133-134: change to "…..increased magnitude and frequency of marine heatwaves (MHW) (i.e., the 2014-2016 and 2019-2020 MHW) [8–10],"

Thank you for this comment. We have made these changes, per your suggestion.

Line 159: Ruzicka et al. (2012, 2016) [32,33] change to Ruzicka et al. [32,33].. please check again in the whole manuscript for the same format problem (example Line 219, 229 and many more).

Thank you for this comment. We have made these changes in the seven locations we found, per your suggestion.

Line 164: marine heatwaves (MHW) change to MHW

Thank you for this comment. We have made these changes, per your suggestion.

Line 178: Remove Northern California Current and keep the short form NCC. The authors already mentioned the short form before. Also, check the whole manuscript for the same problem (for example, Line 329, 484, etc.).

Thank you for this comment. Because readers do not always read an article sequentially from start to finish, we would prefer keeping the full length of the phrase at the beginning of each section so readers do not have to search for acronym definitions. For example, on line 176 the Methods start, and line 178 is the first time we mention, within that section, the Northern California Current. We believe that defining the acronym within each large section improves the readability of the manuscript. With this said, we replaced Northern California Current with NCC at lines 329, 484, and 865, per your suggestion.

Line 260-262: Can the authors include the P/Q = Production efficiency and AE = Assimilation Efficiency value in Table 1?

Thank you for this comment. The Production efficiency (P/Q) is simply the weight-specific production rate (P/B) divided by the weight-specific consumption rate (C/B), both of which are reported in Table 1. Having an additional column that is easily calculated from the two columns already in the Table, makes it unnecessarily large and cumbersome for the reader and the publication process. We have included this calculation in the Table caption as well as a reference to the csv version of Table 1, where readers can find this information already calculated. Assimilation Efficiency is 0.8 for all consumer groups. Instead of adding another column for this, which again is cumbersome, we have now included it in the Table caption. Readers can find both P/Q and 1 – AE (unassimilated) in the csv version of Table 1 in the supplementary data folder at https://doi.org/10.5281/zenodo.7079777.

Line 286: The Ecopath ecotrophic efficiency (EE) change to The Ecopath EE

Thank you for this comment. We have made these changes, per your suggestion.

Line 292-296: "…… underestimated since many animals can avoid sampling gear"..The term many animals here, including birds or non-aquatic animals? If only aquatic animals, please add "aquatic animals" for better understanding.

Thank you for this comment. Yes this statement includes any animals, including seabirds, which can avoid survey vessels.

Line 309: "wasn't" changed to "was not"

Thank you for this comment. We have made these changes, per your suggestion.

Line 372: the unit for biomass density is not mt/km3, as mentioned in Line 342. Please check the whole manuscript for the unit.

Thank you for this comment. We’ve corrected this to say “areal biomass density”, which is the final unit of biomass density that we use in the model. All volumetric biomass densities mt/km3 (a necessary step for some surveys that provide volume sampled) are converted to areal biomass densities mt/km2 in the model (which some surveys provide directly, see Appendix), so each unit is appropriate. We’ve clarified this throughout the manuscript.

Line 375: Which Appendix do the authors mean? Do you refer to Appendix: Newport Hydrographic Line? I suggest the authors mention more details (such as "see Appendix: Newport Hydrographic Line") because this manuscript has a lot of documents and information. Sometimes make, the reader misunderstands.

Thank you for this comment. We have made these changes throughout the manuscript, per your suggestion.

Line 401: metric tons / km3 change to mt/km3

Thank you for this comment. We have made these changes, per your suggestion.

Line 402: Does the "supplemental code" refer to the file in Ruzicka et al., 2016, or the present study? I can't find the supplemental code at the https://doi.org/10.5281/zenodo.7079777. Please check again.

Thank you for this comment. We have now clarified the DOI and directory in the manuscript.

Line 417: Please check the unit = mt km-2

Thank you for this comment. We’ve corrected the formatting to be consistent with other units in the manuscript. The unit itself is correct.

Line 419-420: the full scientific name for E. pacifica and T. spinifera due to first-time mention

Thank you for this comment. We have made these changes, per your suggestion.

Line 422: Change to E. pacifica and T. spinifera

Thank you for this comment. We have made these changes, per your suggestion.

Line 427: Do you mean "Zooplankton, jellies, and pyrosomes" subheading?

Thank you for this question. Yes, we did mean this subheading, and have clarified in the latest version of the manuscript.

Line 432: Again. Please check the subheading name

Thank you for this comment. We have clarified in the latest version of the manuscript.

Line 439 and 457: change to mt/km3

Thank you for this comment. For line 439, mt/km2 is actually the correct unit here. We have now clarified that these biomass densities are by area (areal). We have corrected line 457 to read mt/km3.

Line 460 and 469: the unit for areal biomass densities is not mt/km2? check Line 342

Thank you for this comment. We’ve corrected this throughout the manuscript to say “areal biomass density”, which is the final unit of biomass density that we use in the model (mt/km2). All volumetric biomass densities mt/km3 (a necessary step for some surveys that provide volume sampled) are converted to areal biomass densities mt/km2 in the model (which some surveys provide directly, see Appendix), so each unit is appropriate. We’ve clarified this throughout the manuscript.

Line 484: Northern California Current change to NCC

Thank you for this comment. We have made these changes, per your suggestion.

Line 492 and 494: Ecotrophic Efficiency change to EE.

Thank you for this comment. We have made these changes, per your suggestion.

Line 568 and 577: marine heatwaves change to MHW

Thank you for this comment. We have made these changes, per your suggestion.

Line 628: vertically generalized production model (VGPM) change to VGPM

Thank you for this comment. We have made these changes, per your suggestion.

Line 656: Link, 2010 [39,80]) change to Link [39]. And reference 80 is not published by Link. Please check again the reference.

Thank you for this comment. We have now added “Heymans et al.” to this line to reflect reference 80 (now 81) more accurately.

Line 658: Similar to Line 656. Check the format for reference.

Thank you for this comment. We have made these changes, per your suggestion.

Line 660: biomass-specific production values (P/B) change to P/B

Thank you for this comment. Where we are defining the PREBAL criteria, we would prefer to spell out these names as not everyone is intimately familiar with the acronyms (but keeping the acronyms in parentheses links these terms to the tables and equations in the text).

Line 661: biomass-specific consumption (C/B) change to C/B

Thank you for this comment. Where we are defining the PREBAL criteria, we would prefer to spell out these names as not everyone is intimately familiar with the acronyms (but keeping the acronyms in parentheses links these terms to the tables and equations in the text).

Line 662: ecotrophic efficiency (EE) change to EE

Thank you for this comment. Where we are defining the PREBAL criteria, we would prefer to spell out these names as not everyone is intimately familiar with the acronyms (but keeping the acronyms in parentheses links these terms to the tables and equations in the text).

Line 657-662: I suggest a change to "Food web evaluation criteria guidance from Link [39] states that i; biomass density values of all functional groups should span 5 – 7 orders of magnitude, ii; there should be a 5 – 10% decrease in biomass density (on the log scale) for every unit increase in trophic level, iii; P/B should never exceed C/B values, and iv; EE for each group should be below 1 [39].

Thank you for this comment. It appears that you are suggesting replacing i), ii), iii), and iv) with i;, ii;, iii;, and iv;. We believe using the semicolon in this way is less common and more confusing than the parenthetical as the semicolon often separates clauses in a sentence or objects in a list, whereas the i, ii, iii, and iv here correspond to (and should not be separated from) the following text. For this reason, we are opting to keep the formatting as is.

Line 687: vertically generalized production model (VGPM) change to VGPM

Thank you for this comment. We have made these changes, per your suggestion.

Line 700-702: "Biomass-specific production values (P/B) never exceed biomass-specific consumption (C/B) values, and ecotrophic efficiency (EE) values are all below 1 (Table 1)" change to "P/B never exceed C/B values, and EE values are all below 1 (Table 1)"

Thank you for this comment. Because readers do not always read an article sequentially from start to finish, we would prefer keeping the full length of the phrase at the beginning of each section so readers do not have to search for acronym definitions when starting at any particular section (Results, in this case).

Line 730: "from a vertically generalized production model (VGPM; 2002 - 2021; Fig 10) [49,50]. "change to "from a VGPM 2002 – 2021 (Fig 10) [49,50]."

Thank you for this comment. We have made these changes, per your suggestion.

Line 784: Do the authors mean "see calculated EE in Table 1"

Thank you for this comment. We have made these changes, per your suggestion.

Line 785: euphausiids mean by krills or E. pacifica and T. spinifera or better mentions by the function group in Table 1.

Thank you for this comment. We have now clarified this in the manuscript.

Line 856: marine heatwaves change to MHW

Thank you for this comment. We have made these changes, per your suggestion.

Line 864: Northern California Current change to NCC

Thank you for this comment. We have made these changes, per your suggestion.

Line 881: The data availability statement should mention "supplement or supplement code" because authors used many "see supplement" or "see supplement code" in the text.

Suggestion:

Ecosystem model files, scripts and supplement code, including the balanced and unbalanced diet matrices, biomass estimates, various cleaning scripts, readme files, and all files mentioned in the text can be found at https://doi.org/10.5281/zenodo.7079777.

Thank you for this comment. We have made these changes, per your suggestion.

Table and Figure

Check the unit for Figure 4a and 4b.

Thank you for this comment. We can confirm that the unit is correct in Figures 4a and 4b.

Reviewer #4: The present manuscript describes the EcoTran (extension of ECOPATH) ecosystem model parameterized for the Northern Californian Current system. It is an enhanced version of the existing application with the enhancement achieved through defining more functional groups, considering more detailed spatial structure of the model domain, introducing more fisheries, and adding more recent diet data to capture changes in the marine ecosystem with the recent MHW period. The objective of the study is to propose an efficient tool for ecosystem-based fisheries management. For 90 functional groups/species defined in this application, the authors collected and cleaned data on species diet, landings, biomass densities, and fine-tuned specific input parameters to satisfy the mass-balancing equation. Overall, the manuscript addresses complex and timely question, and the extensive effort on data digging, the model parametrization and validation undoubtedly merits being published. However, the outcomes of the study as they are presented now are weak, and the message that the model can serve as a tool suitable for ecosystem analyses and fisheries management is unconvincing. I believe that the current manuscript requires major revision before it could be considered suitable for publication. Several major problems related to the model construction, the outputs and their validation are listed below.

Thank you for your positive feedback, your constructive criticisms, and for your thoughtful review. 

The model

No immigration and emigration are considered in the model. It is not explicitly stated, nor discussed in the present manuscript, but according to Ruzicka et al (2012), the modelling “assumed a steady-state system with no biomass accumulation and no migration in or out of the system during the model period”. In Table S3 all emigration parameters are set to 0 and advection accounts only for cross-shelf movements (upwelling and downwelling). The horizontal advection seems to exist only to model nutrient flux. How such assumption can be justified for those highly migratory predators, which migrate seasonally to the NCC domain (e.g., albacore and juvenile bluefin tunas and gray whales), or for smaller pelagic species performing offshore-nearshore movements (jack mackerel), or for the species known to undertake the latitudinal migrations with inter-annual variability (Pacific sardine)?

Thank you for this thoughtful comment. You are correct that immigration and emigration are not included in the model. The original (and continuing) impetus for the NCC model was to study juvenile salmon ecology of threatened and endangered populations of Columbia River fish, and the model domain was thus restricted to the northern section of the California Current. Model development also took advantage of surveys that focused on the Northern California Current. We did not consider latitudinal migration across domain boundaries for two reasons. First, we are unable to directly model the processes that affect migrator groups when they are outside of the model domain during winter months. Inclusion of migration dynamics would require rates and timing of migration to be imposed upon model simulations and assumptions to be made about interannual variability in those processes. We made the simplifying assumption that the trophic processes affecting the migrator groups during the winter season in the south would resemble what those processes would be in the north. This would lead to an overestimate of the impact of migrator groups upon the lower trophic food web during winter months. However, the carry-over bias of over-exploitation of lower-trophic groups by migrators during winter months into the summer is minimized by the high intrinsic productivity and the rapid response of the major lower-trophic groups (i.e., copepods and euphausiids) to the seasonal onset of upwelling in the spring. Second, algorithms to simulate latitudinal migration and intra-domain movement processes are a major effort in and of themselves. They are currently being developed as part of the longer-term improvement of the EcoTran platform, but they are beyond the scope of the present manuscript.

The MHW period is presented as the impetus for updating the model, but it remains unclear whether any notable changes could be captured by the updated model, especially since it has not been compared to its previous version. With respect to the MHW impact, is it at all possible to trace its impact and if yes how, if the key variable, i.e., water temperature, is not accounted for in the model?

Thank you for this thoughtful comment. Water temperature is not necessary to compare ecosystem models between two different states. Models are often compared across years or locations, and a comparison across pre-MHW and post-MHW states would be similar. A subset of us have a paper in review doing exactly this with this model (see: https://www.biorxiv.org/content/10.1101/2023.08.11.553012v1.full.pdf, which has now been highlighted in the present manuscript). Additionally, processes are being included into EcoTran through step-by-step improvement – inclusion of water temperature, and how this affects the physiology of various functional groups, is in active development, but beyond the scope of this work. 

The model outputs under historical time series of CUTI seem to be highly correlated, just with expected delays between trophic levels. For example, the same pattern is seen in the time series of Market squid (Fig. 11), sardine and anchovy (Fig. 12), while the stock assessments (driven by CPUE and length or age frequency) show very different dynamics. Interestingly, well-documented (see e.g., MacCall et al., 2016) collapse of anchovy is shown by the stock assessment on Fig. 12, but not captured by EcoTran. I’m not an Ecopath expert and familiar with the modeling approach only through the literature, so I’m not sure if this is the problem of the flawed model assumptions, or a particularity of the modeling approach or the ecosystem driven by seasonal upwelling, but the modelled temporal dynamics seems to be simply driven by the dynamics of nitrate and ammonium at the base of the food web. Thanks to the data availability, I could reproduce these plots and trace the pattern down to the base of the food web. Thus, ammonium and shrimp biomass are linearly correlated with Pearson r=0.91 with 2-monthly lag, then shrimp biomass is correlated with anchovy with monthly lag (r=0.88), sardine biomass is correlated with Chinook yearlings with one-year lag (r=0.95), which is correlated with Chinook group with monthly lag (r=0.99), Chinook group is correlated with tunas with 9-monthly lag (r=0.98) etc. Besides, for some species of similar trophic level, the time series dynamics are nearly identical, e.g., for herring, anchovy, mesopelagic fish aggregate, sardine, and squid. Also, surprising correlations exist between higher trophic level species with very different life history traits, e.g., between gray whales and skates&rays group (r=0.9), tuna and seabirds (0.91), or tunas and hake (r=0.94). Such perfect alignment between temporal dynamics of functional groups seems highly unrealistic, indicating an oversimplification, e.g., considering NCC as a closed ecosystem (see my comment above).

“…the modelled temporal dynamics seems to be simply driven by the dynamics of nitrate and ammonium at the base of the food web” – as you say, yes, this is true, but this is what we would expect because it is a bottom-up (upwelling) driven ecosystem. However, the change of local upwelling characteristics (upwelling strength, seasonal timing, upwelling/downwelling event duration) drives local interannual variability, and the effects of these local environmental dynamics are what we are trying to capture in this model. The diets, biomasses, and physiology parameters scale the magnitude and timing (partially dictate the lags), while the upwelling (or other) physical timeseries is what dictates the seasonal patterns and variation to the simulation runs. There are many ecosystem and behavioral complexities that we have certainly not captured. However, we’d argue that our model is still useful as it allows us to understand food web dynamics (e.g., how nutrients/energy stimulated by upwelling propagates up the food web under different conditions). 

The fact that complex anchovy dynamics, including widespread population collapse, are not captured by an ecosystem model like this one is not surprising, as predicting such stochastic events are notoriously difficult to capture, especially if we do not fully understand the mechanisms behind such a collapse. 

It is not entirely true that the NCC model is a closed system. We are accounting for the daily export losses of phytoplankton and zooplankton production from the NCC domain, and these losses do have a substantial effect on higher trophic levels as has been noted by others, e.g., Botsford et al. 2003 (Fish. Oceanogr. 12:245-259), Rupp et al. 2012 (Fish. Oceanogr. 21:1-19), and Garcia-Reyes et al. 2014 (Prog. Oceanogr. 120:177-188).

Validation

Regarding the phytoplankton validation, the similar periodicity between model predicted phytoplankton biomass density and the VGPM primary production is regarded as the model skill (lines 752-754). However, in many coastal systems, chl-a blooms are driven by upwelling. So, it is very likely that the “remarkably close cyclical resemblance” is primarily the effect of model forcing, i.e., CUTI variability.

Yes, ultimately phytoplankton is driven by the variability in CUTI in the model – as are, ultimately, all functional groups (see detailed response above). However, the remarkably close cyclical resemblance means that we have captured that linkage (CUTI – primary productivity) well enough to reproduce independently observed dynamics. Otherwise, modelled phytoplankton could still be “forced” by CUTI, or any other timeseries, but not have the same timing (e.g., lags, slower or faster periodicity) as the independently observed VGPM timeseries. This alignment is not trivial. 

Since all validation plots are shown on standardized y-axis, it is unclear how close the absolute biomass densities are to independent estimates.

Thank you for this point. Yes, we are not intending to be able to predict absolute biomass densities due to the many missing complexities of the ecosystem (see above). There are many hundreds of parameters in this ecosystem model, and we do not expect that they are all perfectly accurate such as to be able to match absolute biomass over time. We believe this to be an unrealistic goal at this moment in the development of ecosystem models. Instead, the goal here is to be able to understand how the food web (or individual functional groups) are affected in a relative way by perturbations to the system (e.g., via CUTI, individual functional group changes, etc.). We do not wish to show absolute biomass densities as we do not think the model should be used in this way; demonstrating such visualizations will lead the reader to believe that it can (or should) be used in this way. Instead, these validation plots are intended to visualize how well the model does (or does not) match temporal trends in surveyed biomass [i.e., how well (or not) does upwelling and our food web parameterization capture real trends].

Finally, if describing the adequate temporal dynamics of intermediate and high trophic level species, is beyond the model capacity, I encourage authors to demonstrate those model skills, for which it can be a helpful and reliable tool for fisheries management. The validation can be done, for example, by running retrospective analyses, results of which can be verified with independent data.

Thank you for this comment. If we understand it correctly, market squid, sardine, anchovy, Pacific mackerel, and jack mackerel are visualized in Figures 11 and 12 as the reviewer describes, and are themselves intermediate trophic level species (Trophic levels = 3.63, 3.11, 3.19, 3.49 and 3.64, respectively). However, as the reviewer points out we do not have any high trophic level species represented in these plots. We agree that this is beyond the model capacity and appreciate the point that it is worth displaying these model skills (or lack thereof). We have now included four more species of relatively high trophic levels (common murres TL = 4.37, sooty shearwaters TL = 4.31, baleen whales TL = 3.69, and southern resident killer whales TL = 5.13) as Figure 13 and we have added language about these plots within the methods and discussion.

Minor comments:

Figures 5-7 are still difficult to read and interpret. Even though a color coding is added to distinguish functional groups, reading the connections between species, and interpreting the diet interactions is impossible. So, either these figures should deliver some qualitative information like footprint and reach in Ruzicka et al. (2012, Figure 6), or it would be better to make group aggregations to make the graph with less vertices, or even provide the diet matrix as in Ruzicka et al. (2012, Table A.3).

Thank you for this comment. We disagree that these figures are impossible to interpret as is. The arrows fade away when interactions are less important and become broader and darker as the interactions are more important. Taking Figure 7 as an example, it is clear that more energy is flowing to functional group 43 than functional group 91. There is something to be gained and something lost in every different view of a network diagram, and we do not intend to be able to capture every angle of it. We believe that reducing the diagram to connections to and from more aggregated groups will lose the overall connectivity of the entire network that we are trying to demonstrate here. We believe this more holistic visualization is more representative of the ecosystem model (Figs 5-7) that we’ve built. We have made all figure code publicly available such that readers can modify the plots further if they wish to only see connections to and from particular functional groups. Changing the line thickness to represent footprint and reach (as in Ruzicka et al. 2012 Fig 6) instead of energy flux won’t actually change the complexity of these figures, it will only change which continuous metric is visualized.

Additionally, as you’ve suggested, we have provided the full diet matrix in the supplementary data (https://doi.org/10.5281/zenodo.7079777) in both the file “NCC2_09032022.csv” (and .xlsx, ‘Diets’ tab) in the root directory and in “Final_DietMatrix_Balanced_Unbalanced_Difference.csv” (and .xlsx) in directory: Reproducibility\\DietWork\\.

Fig 10. Is there any reason to include all 15 regions into the “time series” of observed phytoplankton, which results in vertical lines every monthly date? Comparing the biomass densities averaged over the model domain seems to be more appropriate. Note also, two months of data are missing in the VGPM PP data file.

Thank you for catching the 2 months of missing data in the VGPM data file. The missing data (August and Sept. 2020) was during active model development and we assume that one iteration of downloading this data included empty values for those months. We have revisited the VGPM data and this has now been fixed in the latest version of the data and code repository. Also, thank you for catching the fact that all 15 regions were plotted unnecessarily. We have now replaced Figure 10 with a similar one with all 15 regions averaged, per your suggestion (which now includes the missing months of data). We agree that this has improved the readability of the figure. 

Reviewer #5: a. The model is a useful tool and made available to other researchers to use, with potential to improve ecosystem-based management practices. The extensive data sources and compilation improves the existing model and improves utility for potential users. The authors have addressed previous reviewer comments. I suggest this paper is published and only have minor comments.

Thank you for your positive feedback and for your thoughtful review. 

b. Line 234: The latitude of the Newport Hydrographic Line is farther south than 46.7*N. Is this an incorrect latitude, or are the authors referencing a different transect?

Thank you for catching this. This latitude is one of the subregional boundaries and was accidentally copied into the incorrect place. This has now been fixed in the most recent version of the manuscript.

c. Line 772 – 778: In the jelly/gelatinous zooplankton paragraph, the authors could elaborate on the ecosystem and management implications of the rise or persistence of gelatinous organism, as discussed in discussed in Brodeur et al., 2019, not just sampling bias issues.

Thank you for this suggestion. We have now expanded on this discussion, per your suggestion, and we agree that this improves the quality of the manuscript.

d. Line 792-794: Could the authors be more specific in how the six functional groups might improve management, e.g. various stocks versus seasonality?

Thank you for this suggestion. We have now expanded on this discussion, per your suggestion, and we agree that this improves the quality of the manuscript.

e. General comment: In the introduction, the authors note recent changes in the NCC, such as marine heatwaves, ocean acidification, etc. Can you speak in the discussion on if, or how, this model might reflect a regime shift in the ecosystem? How different were the results of this model iteration to previous model iterations (e.g., Ruzicka et al., 2016) that did not include 2015-2019 data?

Thank you for this thoughtful question. We have added language about this to the 6th and to the last paragraph of the discussion. A subset of us have a paper in review that further explores this topic in much more detail (https://www.biorxiv.org/content/10.1101/2023.08.11.553012v1.full.pdf). The present manuscript is already quite long and cumbersome as is, but we hope that the sentences that we’ve added here set up the next manuscript in a satisfying way for the reader.

---

## [Decision Letter · Decision Letter 2]

21 Dec 2023

An updated end-to-end ecosystem model of the Northern California Current reflecting ecosystem changes due to recent marine heatwaves

PONE-D-22-35465R2

Dear Dr. Gomes,

We’re pleased to inform you that your manuscript has been judged scientifically suitable for publication and will be formally accepted for publication once it meets all outstanding technical requirements.

Kind regards,

Abdul Azeez Pokkathappada, Ph.D.

Academic Editor

PLOS ONE

Additional Editor Comments (optional):

Reviewers' comments:

Reviewer's Responses to Questions

**Comments to the Author**

1. If the authors have adequately addressed your comments raised in a previous round of review and you feel that this manuscript is now acceptable for publication, you may indicate that here to bypass the “Comments to the Author” section, enter your conflict of interest statement in the “Confidential to Editor” section, and submit your "Accept" recommendation.

Reviewer #4: All comments have been addressed

Reviewer #5: All comments have been addressed

2. Is the manuscript technically sound, and do the data support the conclusions?

Reviewer #4: Yes

Reviewer #5: Yes

3. Has the statistical analysis been performed appropriately and rigorously? 

Reviewer #4: N/A

Reviewer #5: I Don't Know

4. Have the authors made all data underlying the findings in their manuscript fully available?

Reviewer #4: Yes

Reviewer #5: Yes

5. Is the manuscript presented in an intelligible fashion and written in standard English?

Reviewer #4: Yes

Reviewer #5: Yes

6. Review Comments to the Author

Reviewer #4: (No Response)

Reviewer #5: I have previously reviewed this manuscript and in this revision, I found that the authors have addressed my minor suggestions. In reading other reviews, I believe the authors have addressed their comments, but I cannot speak to the statistical analysis or model assumptions. I represent the end-user audience of this model. This research represents an advancement of established ECOTRAN modeling in an ecologically important upwelling system, with consideration for recent environmental change, added parameters, and more functional groups.

7. PLOS authors have the option to publish the peer review history of their article (what does this mean?). If published, this will include your full peer review and any attached files.

Reviewer #4: No

Reviewer #5: **Yes: **M. Kelsey Lane

---

## [Editor Report · Acceptance letter]

10 Jan 2024

PONE-D-22-35465R2 

PLOS ONE

Dear Dr. Gomes, 

I'm pleased to inform you that your manuscript has been deemed suitable for publication in PLOS ONE. Congratulations! Your manuscript is now being handed over to our production team.

Kind regards, 

on behalf of

Dr. Abdul Azeez Pokkathappada 

Academic Editor

PLOS ONE